# Smooth Calibration Error: Uniform Convergence and Functional Gradient Analysis

**Futoshi Futami**[1,2,*]**, Atsushi Nitanda**[3,4,5,†]

[1]The University of Osaka, Japan
[2]RIKEN AIP, Japan
[3]Institute of High Performance Computing, Agency for Science, Technology and Research, Singapore
[4]Centre for Frontier AI Research, Agency for Science, Technology and Research, Singapore
[5]College of Computing and Data Science, Nanyang Technological University, Singapore

[*]`futami.futoshi.es@osaka-u.ac.jp`
[†]`Atsushi_Nitanda@a-star.edu.sg`

## Abstract

Calibration is a critical requirement for reliable probabilistic prediction, especially in high-risk applications. However, the theoretical understanding of which learning algorithms can simultaneously achieve high accuracy and good calibration remains limited, and many existing studies provide empirical validation or a theoretical guarantee in restrictive settings. To address this issue, in this work, we focus on the smooth calibration error (CE) and provide a uniform convergence bound, showing that the smooth CE is bounded by the sum of the smooth CE over the training dataset and a generalization gap. We further prove that the functional gradient of the loss function can effectively control the training smooth CE. Based on this framework, we analyze three representative algorithms: gradient boosting trees, kernel boosting, and two-layer neural networks. For each, we derive conditions under which both classification and calibration performances are simultaneously guaranteed. Our results offer new theoretical insights and practical guidance for designing reliable probabilistic models with provable calibration guarantees.

## 1 Introduction

Probabilistic prediction plays a central role in many real-world applications, such as healthcare (Jiang et al., 2012), weather forecasting (Murphy & Winkler, 1984), and language modeling (Nguyen & O'Connor, 2015), where uncertainty estimates are often critical. Since ensuring the reliability of such predictions has become a key challenge in modern machine learning, *calibration* Dawid (1982); Foster & Vohra (1998)—which requires that predicted probabilities align with the actual frequency of the true label—has attracted increasing attention. Calibration is a relatively weak condition that can be satisfied by simple models. However, it is frequently violated in practice. Notably, Guo et al. (2017) demonstrated that many modern deep learning models can be significantly miscalibrated, which has since drawn increasing attention to calibration within the machine learning community. Although various regularization techniques (Kumar et al., 2018; Karandikar et al., 2021; Popordanoska et al., 2022; Marx et al., 2023) and recalibration methods (Zadrozny & Elkan, 2001; Guo et al., 2017; Gupta & Ramdas, 2021; Kull et al., 2019) have been proposed to improve calibration, most of them are either empirically evaluated without a theoretical guarantee or lead to a trade-off between calibration and sharpness (Kuleshov & Liang, 2015), degrading predictive accuracy. As a result, it remains unclear which learning algorithms can train well-calibrated predictors without sacrificing accuracy.

One classical approach to this problem is the minimization of *proper loss functions*, such as the squared loss or the cross-entropy loss (Schervish, 1989; Buja et al., 2005). These losses are minimized in expectation by the true conditional probability and thus have a natural connection to calibration. However, in practice, constraints on model classes and suboptimal optimization may prevent proper losses from yielding calibrated predictions. To address this issue, in a recent theoretical work, the notion of a *post-processing gap* (Błasiok et al., 2023) has been introduced, which quantifies the

potential improvement of the loss function achievable by applying Lipschitz-continuous transformations to the model outputs. This framework provides an optimization-theoretic perspective on calibration. Nevertheless, analyses based on this concept typically assume access to infinite data (i.e., population-level risk) (Błasiok et al., 2023; Globus-Harris et al., 2023), which limits their applicability to real-world algorithms based on finite training samples. Furthermore, only a few theoretical results explicitly connect calibration error with concrete learning algorithms (Hansen et al., 2024; Błasiok et al., 2023). As a result, a key question remains unresolved: *What types of learning algorithms can achieve both high predictive accuracy and strong calibration guarantees in practice?*

To address this gap, we focus on the *smooth calibration error* (CE) (Błasiok et al., 2023; Foster & Hart, 2018; Kakade & Foster, 2008), a recently proposed calibration metric for binary classification that has favorable theoretical properties. Using smooth CE, we contribute to answering the above question through two main theoretical advances. First, we derive a uniform convergence bound for smooth CE, showing that the population-level smooth CE can be bounded by the smooth CE over the training dataset plus a generalization gap (Section 3). Second, we prove that such training smooth CE can be bounded by the norm of the functional gradient (or its approximation) of the loss evaluated on training data, providing a principled criterion for optimizing calibration (Section 4). Taken together, these results show that algorithms that achieve small functional gradients while regularizing model complexity can obtain small smooth CE.

We apply our theoretical framework to three representative algorithms for the binary classification closely tied to functional gradients: gradient boosting trees (GBTs), kernel boosting, and two-layer neural networks (NNs). For GBTs, we provide the theoretical guarantee for their calibration performance, showing that the smooth CE decreases with iteration (Section 4.1). For kernel boosting, we analyze the trade-off between optimization and model complexity induced by the reproducing kernel Hilbert space (Section 4.2). For two-layer NNs, we leverage their connection to the neural tangent kernel to study calibration behavior on the basis of function space optimization (Section 4.3). Furthermore, for all these algorithms, by introducing a margin-based assumption, we derive sufficient conditions on the sample size and the number of iterations required to simultaneously achieve $\epsilon$-level smooth CE and misclassification rate under appropriately chosen hyperparameters.

In summary, in this work, we develop a unified theoretical framework for the smooth CE that integrates a uniform convergence analysis with a functional-gradient-based characterization of training dynamics. Our results establish new theoretical foundations for understanding and improving calibration in modern learning algorithms.

## 2 Preliminaries

In this section, we introduce proper losses and the smooth CE.

### 2.1 Problem setting of binary classification

We denote random variables by capital letters (e.g., $X$) and deterministic values by lowercase letters (e.g., $x$). The Euclidean inner product and norm are denoted by $\cdot$ and $\|\cdot\|$, respectively.

We consider a binary classification problem in the supervised setting. Let $\mathcal{Z} = \mathcal{X} \times \mathcal{Y}$ denote the data domain, where $\mathcal{X} \subset \mathbb{R}^d$ is the input space and $\mathcal{Y} = \{0, 1\}$ is the label space. We assume that data points are sampled from an unknown data-generating distribution $\mathcal{D}$ over $\mathcal{X} \times \mathcal{Y}$. Let $\mathcal{F}$ be a class of predictors $f : \mathcal{X} \to [0, 1]$ approximating the ground-truth conditional probability $f^*(x) = \mathbb{E}[Y|x]$.

To evaluate the performance of the predictor, we use a loss function. Since the task is binary classification, the label $Y$ follows a Bernoulli distribution, i.e., $Y \sim \text{Ber}(v)$ for some $v \in [0, 1]$, where $\Pr(Y = 1) = v$. A loss function is defined as $\ell : [0, 1] \times \mathcal{Y} \to \mathbb{R}$, with evaluation given by $\ell(v, y)$. Our goal is not only to achieve high classification accuracy but also to assess the quality of the predicted probabilities. In this context, proper loss functions play a central role (Gneiting & Raftery, 2007). The loss $\ell$ is called proper if, for any $v, v' \in [0, 1]$, the following holds: $\mathbb{E}_{Y \sim \text{Ber}(v)}[\ell(v, Y)] \leq \mathbb{E}_{Y \sim \text{Ber}(v)}[\ell(v', Y)]$.

For clarity, we focus on two representative proper losses: the squared loss $\ell_{\text{sq}}(v, y) := (y - v)^2$ and the cross-entropy loss $\ell_{\text{ent}}(v, y) := -y \log v - (1 - y) \log(1 - v)$. Given $x$ and $f$, we substitute $v = f(x)$ in these expressions. For other proper losses, see Appendix A and Błasiok et al. (2023).

## 2.2 CALIBRATION METRICS AND LOSS FUNCTIONS

Given a distribution $\mathcal{D}$ and a predictor $f$, we say the model is *perfectly calibrated* if $\mathbb{E}[Y \mid f(X)] = f(X)$ holds almost surely (Błasiok et al., 2023). Various metrics have been proposed to quantify the deviation from perfect calibration. The most widely used metric is the *expected calibration error (ECE)*: $\mathrm{ECE}(f) \coloneqq \mathbb{E}\left[\left|\mathbb{E}[Y \mid f(X)] - f(X)\right|\right]$. Despite its popularity, ECE is difficult to estimate efficiently (Arrieta-Ibarra et al., 2022; Lee et al., 2023) and is known to be discontinuous (Foster & Hart, 2018; Kakade & Foster, 2008). As a more tractable alternative, we focus on the *smooth CE*:

**Definition 1** (Smooth CE (Błasiok et al., 2023)). *Let* $\mathrm{Lip}_L([0,1],[-1,1])$ *be the set of $L$-Lipschitz functions from* $[0,1]$ *to* $[-1,1]$. *Given $f$ and $\mathcal{D}$, the smooth CE is defined as*

$$\mathrm{smCE}(f,\mathcal{D}) \coloneqq \sup_{h \in \mathrm{Lip}_{L=1}([0,1],[-1,1])} \mathbb{E}\left[h(f(X)) \cdot (Y - f(X))\right].$$

We remark that the concept of the smooth CE is also introduced in Foster & Hart (2018); Kakade & Foster (2008). We also express $\mathrm{smCE}(f,\mathcal{D}) = \sup_{h \in \mathrm{Lip}_1([0,1],[-1,1])} \langle h(f(X)), Y - f(X) \rangle_{L_2(\mathcal{D})}$. The smooth CE offers favorable properties such as continuity and computational tractability (Błasiok et al., 2023; Hu et al., 2024). Moreover, the smooth CE provides both upper and lower bounds on the binning ECE, making it a principled surrogate for analyzing calibration. See Section 5 for details.

Błasiok et al. (2023) established a connection between the smooth CE and the optimization based on the concept of the possible improvement of the function under the squared loss. Given $f$ and $\mathcal{D}$, we define the post-processing gap as

$$\mathrm{pGap}(f,\mathcal{D}) \coloneqq \mathbb{E}[\ell_{\mathrm{sq}}(f(X),Y)] - \inf_{h \in \mathrm{Lip}_1([0,1],[-1,1])} \mathbb{E}[\ell_{\mathrm{sq}}(f(X) + h(f(X)),Y)].$$

Then, the following inequality holds (Theorem 2.4 in Błasiok et al. (2023)):

$$\mathrm{smCE}(f,\mathcal{D})^2 \leq \mathrm{pGap}(f,\mathcal{D}) \leq 2\mathrm{smCE}(f,\mathcal{D}). \tag{1}$$

Błasiok et al. (2023) also provided analogous results for the general proper scoring rule, which indicates that the potential improvement in the population risk is closely tied to calibration quality.

We now consider the setting of the cross-entropy loss. In practice, the predicted probability $f$ is often obtained by applying the sigmoid function $\sigma$ to a logit function $g : \mathcal{X} \to \mathbb{R}$, i.e., $f(x) = \sigma(g(x)) = 1/(1 + e^{-g(x)})$. Therefore, it is more natural to consider post-processing over the logit $g$ rather than the predicted probability $f$, as many models explicitly learn logits and apply the sigmoid function only at the final prediction stage. We define a corresponding post-processing gap over the logit function with the cross-entropy loss as

$$\mathrm{pGap}^\sigma(g,\mathcal{D}) \coloneqq \mathbb{E}[\ell_{\mathrm{ent}}(\sigma(g(X)),Y)] - \inf_{h \in \mathrm{Lip}_1(\mathbb{R},[-4,4])} \mathbb{E}[\ell_{\mathrm{ent}}(\sigma(g(X) + h(g(X))),Y)],$$

where $\mathrm{Lip}_1(\mathbb{R},[-4,4])$ defines a class of 1-Lipschitz functions from $\mathbb{R}$ to $[-4,4]$ and the choice of 4 reflects the fact that the sigmoid function is $1/4$-Lipschitz. Then, this post-processing leads to defining a smooth CE directly over the logit function:

**Definition 2** (Dual smooth CE (Błasiok et al., 2023)). *Given a logit function $g : \mathcal{X} \to \mathbb{R}$ and $f(x) = \sigma(g(x))$, the dual smooth CE is defined as*

$$\mathrm{smCE}^\sigma(g,\mathcal{D}) \coloneqq \sup_{h \in \mathrm{Lip}_{1/4}(\mathbb{R},[-1,1])} \langle h(g(X)), Y - f(X) \rangle_{L_2(\mathcal{D})}.$$

Similarly to Eq. (1), the following relation holds:

$$2\mathrm{smCE}^\sigma(g,\mathcal{D})^2 \leq \mathrm{pGap}^\sigma(g,\mathcal{D}) \leq 4\mathrm{smCE}^\sigma(g,\mathcal{D}). \tag{2}$$

Finally, we note that the dual smooth CE always upper bounds the smooth CE: $\mathrm{smCE}(f,\mathcal{D}) \leq \mathrm{smCE}^\sigma(g,\mathcal{D})$. Hence, minimizing the dual smooth CE guarantees a small value of the smooth CE.

## 3 UNIFORM CONVERGENCE OF THE SMOOTH CE

As shown in Eqs. (1) and (2), (dual) smooth CE is related to improvements in the population risk, and Błasiok et al. (2023) utilized these relationships to analyze when algorithms can achieve a small

smooth CE. However, their results are defined over population-level quantities, where the expectation by $\mathcal{D}$ is taken, making them inapplicable to practical algorithms trained on finite data points.

Since our goal is to analyze the behavior of the smooth CE under standard learning algorithms trained on finite data, it is natural to consider its empirical counterpart. Given a dataset $S_n = \{(X_i, Y_i)\}_{i=1}^n$ consisting of independently and identically distributed (i.i.d.) data points sampled from the data distribution $\mathcal{D}$, we define the **empirical smooth CE** (Błasiok et al., 2023) given $S_n$ as

$$\mathrm{smCE}(f, S_n) := \sup_{h \in \mathrm{Lip}_1([0,1],[-1,1])} \frac{1}{n} \sum_{i=1}^n h(f(X_i)) \cdot (Y_i - f(X_i)).$$

We also express this as $\mathrm{smCE}(f, S_n) = \sup_{h \in \mathrm{Lip}_1([0,1],[-1,1])} \langle h(f(X)), Y - f(X) \rangle_{L_2(S_n)}$. We similarly define $\mathrm{smCE}^\sigma(g, S_n)$ as empirical counterparts of $\mathrm{smCE}^\sigma(g, \mathcal{D})$.

We consider two datasets: a training set $S_{\mathrm{tr}} = \{Z_i\}_{i=1}^n \sim \mathcal{D}^n$, used to learn the predictor $f$, and an independent test set $S_{\mathrm{te}} = \{Z_i'\}_{i=1}^n \sim \mathcal{D}^n$. We call $\mathrm{smCE}(f, S_{\mathrm{tr}})$ as the **training smooth CE**, $\mathrm{smCE}(f, S_{\mathrm{te}})$ as the **test smooth CE**, and $\mathrm{smCE}(f, \mathcal{D})$ as the **population smooth CE**. Given $S_{\mathrm{tr}}$, we are interested in evaluating $|\mathrm{smCE}(f, \mathcal{D}) - \mathrm{smCE}(f, S_{\mathrm{tr}})|$. Błasiok et al. (2023) has shown that

$$|\mathrm{smCE}(f, \mathcal{D}) - \mathrm{smCE}(f, S_{\mathrm{te}})| = \mathcal{O}_p(1/\sqrt{n}). \tag{3}$$

Therefore, we need to evaluate $|\mathrm{smCE}(f, S_{\mathrm{te}}) - \mathrm{smCE}(f, S_{\mathrm{tr}})|$, which we refer to as the **smooth CE generalization gap**. Combining this gap with Eq. (3), we obtain the desired bound. To evaluate the smooth CE generalization gap, we use the covering number bound (Wainwright, 2019). Suppose that $\mathcal{F}$ is equipped with the metric $\|\cdot\|_\infty$. Let $\mathcal{N}(\epsilon, \mathcal{F}, \|\cdot\|_\infty)$ be an $\epsilon$-cover with metric $\|\cdot\|_\infty$ of $\mathcal{F}$, with the cardinality $N(\epsilon, \mathcal{F}, \|\cdot\|_\infty)$. Then, for any $f \in \mathcal{F}$, there exists $\tilde{f} \in \mathcal{N}(\epsilon, \mathcal{F}, \|\cdot\|_\infty)$ such that $\|f - \tilde{f}\|_\infty \leq \epsilon$.

We now present our first main result: (All proofs for this section are provided in Appendix C.)

**Theorem 1.** *Let $S_{\mathrm{te}} \sim \mathcal{D}^n$ and $S_{\mathrm{tr}} \sim \mathcal{D}^n$ be independent test and training datasets. Then, for any $\delta > 0$, with probability at least $1 - \delta$ over the draw of $S_{\mathrm{te}}$ and $S_{\mathrm{tr}}$, we have:*

$$\sup_{f \in \mathcal{F}} |\mathrm{smCE}(f, S_{\mathrm{te}}) - \mathrm{smCE}(f, S_{\mathrm{tr}})| \leq \inf_{\epsilon \geq 0} 8\epsilon + 24 \int_{\epsilon'}^1 \sqrt{\frac{\ln N(\epsilon', \mathcal{F}, \|\cdot\|_\infty)}{n}} d\epsilon' + 2\sqrt{\frac{\log \delta^{-1}}{n}}.$$

Since the smooth CE involves evaluating the composite function $h(f(X))$, standard chaining bounds would introduce complexity over the composite class $\mathrm{Lip}_1([0,1],[-1,1]) \circ \mathcal{F}$. However, by leveraging the smoothness property of smooth CE, the above bound does not include the complexity of Lipschitz functions. Combining Theorem 1 with Eq. (3), we obtain the bound for the **population smooth CE**.

**Corollary 1.** *Under the same assumptions as in Theorem 1, there exist a universal constant $C_1$ such that with probability at least $1 - \delta$ over the draw of $S_{\mathrm{tr}}$, the following holds for all $f \in \mathcal{F}$:*

$$|\mathrm{smCE}(f, \mathcal{D}) - \mathrm{smCE}(f, S_{\mathrm{tr}})| \leq \frac{C_1}{\sqrt{n}} + \inf_{\epsilon \geq 0} 8\epsilon + 24 \int_{\epsilon'}^1 \sqrt{\frac{\ln N(\epsilon', \mathcal{F}, \|\cdot\|_\infty)}{n}} d\epsilon' + 3\sqrt{\frac{\log \frac{3}{\delta}}{n}}.$$

The first term of the right-hand side represents the complexity of the Lipschitz function class.

In certain cases, the Rademacher complexity offers a more interpretable characterization (Mohri et al., 2018). The empirical and expected Rademacher complexities of a function class $\mathcal{F}$ are defined as $\hat{\mathfrak{R}}_S(\mathcal{F}) := \mathbb{E}_\sigma \sup_{f \in \mathcal{F}} \frac{1}{n} \sum_{i=1}^n \sigma_i f(x_i)$, and $\mathfrak{R}_{\mathcal{D},n}(\mathcal{F}) := \mathbb{E}_{\mathcal{D}^n}[\hat{\mathfrak{R}}_S(\mathcal{F})]$, respectively.

**Theorem 2.** *Under the same assumptions as in Theorem 1, there exist a universal constant $C_2$ such that with probability at least $1 - \delta$ over the draw of $S_{\mathrm{tr}}$, the following holds:*

$$\sup_{f \in \mathcal{F}} |\mathrm{smCE}(f, \mathcal{D}) - \mathrm{smCE}(f, S_{\mathrm{tr}})| \leq \frac{C_2}{\sqrt{n}} + 4\mathfrak{R}_{\mathcal{D},n}(\mathcal{F}) + 2\sqrt{\frac{\log \frac{2}{\delta}}{n}}.$$

The first term on the right-hand side reflects the complexity of $\mathrm{Lip}_1([0,1],[-1,1])$, while the second corresponds to $\mathcal{F}$. A direct application of uniform convergence theory would yield a bound in terms of $\mathfrak{R}_{\mathcal{D},n}(\mathrm{Lip}_1([0,1],[-1,1]) \circ \mathcal{F})$, which cannot be reduced to $\mathfrak{R}_{\mathcal{D},n}(\mathcal{F})$ using the standard contraction

lemma. Our analysis circumvents this difficulty by employing a covering argument for Lipschitz functions. Similarly, Rademacher complexity can be used to bound $|\text{smCE}(f, S_{\text{te}}) - \text{smCE}(f, S_{\text{tr}})|$. However, such a bound depends on the complexity of the Lipschitz function, making it suboptimal relative to Theorem 1. See Appendix D for a detailed comparison between Corollary 1 and Theorem 2.

In conclusion, by jointly controlling the complexity of the hypothesis class and reducing the training smooth CE, we guarantee a small population smooth CE. In the next section, we show how to achieve a small training smooth CE via functional gradients.

## 4 CONTROLLING THE SMOOTH CE VIA FUNCTIONAL GRADIENT

In this section, we discuss how we can minimize the training smooth CE. From Eqs. (1) and (2), (dual) smooth CE can be characterized via the (dual) post-processing gap, which describes the potential improvement over *functions*. Building on this perspective, we show that both smooth and dual smooth CEs can be further characterized using *functional gradients*. Since we focus on the functional gradients over the training dataset, they are just finite-dimensional vectors. We provide the precise connection between functional gradients and the population smooth CE in Appendix I.

For the squared loss, the gradient for the predicted probability is $\nabla_f \ell_{\text{sq}}(f, y) = f - y$, and for the cross-entropy loss, the gradient for the logit is $\nabla_g \ell_{\text{ent}}(\sigma(g), y) = \sigma(g) - y$. Using these, we can write the smooth and dual smooth CEs on the training dataset $S_{\text{tr}}$ as

$$\text{smCE}(f, S_{\text{tr}}) = \sup_{h \in \text{Lip}_1([0,1],[-1,1])} \langle h(f(X)), -\nabla_f \ell_{\text{sq}}(f(X), Y) \rangle_{L_2(S_n)},$$

$$\text{smCE}^\sigma(g, S_{\text{tr}}) = \sup_{h \in \text{Lip}_{1/4}(\mathbb{R},[-1,1])} \langle h(g(X)), -\nabla_g \ell_{\text{ent}}(\sigma(g(X)), Y) \rangle_{L_2(S_n)}.$$

The connection between functional gradients and the smooth CE can be extended to general proper scoring rules using the post-processing gaps, see Appendix I for the details.

Therefore, to effectively control the training smooth CE, it is natural to consider algorithms that focus on functional gradients. From this perspective, in the following subsections, we investigate how the smooth CE interacts with the training dynamics of three widely used models: gradient boosting tree, kernel boosting in a reproducing kernel Hilbert space (RKHS), and two-layer neural networks. These models can all be interpreted on the basis of functional gradients, and our theoretical framework enables a unified understanding of their calibration behavior under different forms of regularization.

In this section, we express the empirical risk as $L_n(g) = \frac{1}{n} \sum_{i=1}^n \ell_{\text{ent}}(\sigma(g(X_i)), Y_i)$ and the functional gradient over the training dataset as $\nabla_g L_n(g) = (\nabla_g \ell_{\text{ent}}(\sigma(g(X_1)), Y_1)), \dots, (\nabla_g \ell_{\text{ent}}(\sigma(g(X_n)), Y_n)) \in \mathbb{R}^n$.

### 4.1 CASE STUDY I: GRADIENT BOOSTING TREE

Gradient boosting (Friedman, 2001) is a widely used method for constructing predictive models by iteratively adding base learners to minimize a loss function. It generates a sequence of functions $\{g^{(t)}\}_{t=0}^T$ via the updates, for $t = 0, \dots, T-1$:

$$g^{(t+1)}(x) = g^{(t)}(x) - w_t \psi_t(x), \tag{4}$$

where $w_t > 0$ is the stepsize and $\psi_t$ approximates the functional gradient of the empirical loss (Mason et al., 1999). Since using the exact functional gradient often leads to overfitting, $\psi_t$ is restricted to a predefined function class $\Psi$, which provides implicit regularization. We impose the following mild assumption on $\Psi$, which is readily satisfied by common choices such as regression trees.

**Assumption 1.** *The set of real-valued functions $\Psi$ satisfies: for every $\psi \in \Psi$, the negation $-\psi$ also belongs to $\Psi$; $\sup_{\psi \in \Psi} \|\psi\|_\infty \leq B$ for some $B \geq 1$; and the constant function $0$ is included in $\Psi$.*

To obtain the update direction, we iteratively solve $\psi^{(t)} = \arg\min_{\psi \in \Psi} L_n(g^{(t)} - w_t \psi(x))$. Many boosting methods approximate this by a quadratic upper bound, leading to the squared loss

$$\psi_t = \arg\min_{\psi \in \Psi} \|M w_t \psi - \nabla_g L_n(g^{(t)})\|_{L_2(S_n)}^2 = \frac{1}{n} \sum_{i=1}^n |M w_t \psi(X_i) - \nabla_g \ell_{\text{ent}}(\sigma(g(X_i)), Y_i)|^2.$$

where $M$ is the smoothness parameter of the loss (e.g., $M = 1/4$ for cross-entropy).

Although gradient boosting is often observed to be well calibrated in practice (Niculescu-Mizil & Caruana, 2005; Wenger et al., 2020), its theoretical guarantees remain less understood. Our framework provides a natural way to analyze this. Following prior analysis on boosting, we impose a standard margin assumption:

**Assumption 2** (Telgarsky (2013)). *Let $\Delta_n = \{q \in \mathbb{R}_+^n \mid \sum_{i=1}^n q_i = 1\}$ denote the $n$-dimensional probability simplex. Given a dataset $S_{\mathrm{tr}} = \{(X_i, Y_i)\}_{i=1}^n$, there exists $\gamma > 0$ such that for every $q \in \Delta_n$, there exists $\psi \in \Psi$ satisfying $\sum_{i=1}^n q_i(2Y_i - 1)\psi(X_i) \geq \gamma B$.*

Here, $2Y_i - 1$ maps the binary label $Y_i \in \{0, 1\}$ to $\pm 1$. This condition guarantees that for any weighted sample distribution $q \in \Delta_n$, one can choose a base learner with nontrivial correlation to the labels. Such margin assumptions are also standard in classification theory (Wei et al., 2018). Under this setup, we present our main result for the **training dual smooth CE**:

**Theorem 3.** *Given $S_{\mathrm{tr}}$, under Assumptions 1 and 2, if $T \geq 2$ with constant stepsize $w_t = w$, the averaged predictor $\bar{g}^{(T)} = \frac{1}{T} \sum_{t=0}^{T-1} g^{(t)}$ satisfies*

$$\mathrm{smCE}^\sigma\big(\bar{g}^{(T)}, S_n\big) \leq \frac{L_n(g^{(0)})}{\gamma BwT} + \frac{wB}{8\gamma}.$$

*Proof outline.* By definition, the smooth CE is bounded by the $L_1$ norm of the functional gradient:

$$\mathrm{smCE}^\sigma(g, S_{\mathrm{tr}}) \leq \frac{1}{n} \sum_{i=1}^n |\nabla_g \ell_{\mathrm{ent}}(g(X_i), Y_i)| = \|\nabla_g \ell_{\mathrm{ent}}(g(X), Y)\|_{L_1(S_n)}. \tag{5}$$

Under Assumption 2, this norm can be further bounded as

$$\|\nabla_g \ell_{\mathrm{ent}}(g(X), Y)\|_{L_1(S_n)} \leq \frac{1}{\gamma B} \langle \psi_t, \nabla_t \rangle_{L_2(S_n)} + \frac{w}{8\gamma} B$$

as shown in Lemma 4 in Appendix F. The inner product $\langle \psi_t, \nabla_t \rangle_{L_2(S_n)}$ is then bounded based on the standard convex optimization techniques. See Appendix F.3 for the complete proof. $\square$

Thus, choosing $w = \mathcal{O}(1/\sqrt{T})$ ensures that the training smooth CE converges to 0 at rate $\mathcal{O}(1/\sqrt{T})$.

Next, we establish a guarantee for the **population smooth CE** by applying Theorem 2. This requires specifying $\Psi$ to evaluate the complexity of $\mathcal{F}$. We consider gradient boosting trees (Friedman, 2002; Hastie et al., 2005), where $\Psi$ is the class of binary regression trees. A binary regression tree of depth $m$ partitions the input space $\mathbb{R}^d$ into at most $J \leq 2^m$ disjoint regions: $\mathbb{R}^d = \bigcup_{j=1}^J R_j$, with $R_j \cap R_k = \emptyset$ for $j \neq k$. Each region $R_j$ is assigned a constant $c_j \in \mathbb{R}$, yielding a piecewise constant function: $\psi_\theta(x) = \sum_{j=1}^J c_j \cdot \mathbb{1}_{\{x \in R_j\}}$ with parameters $\theta = \{c_j, R_j\}_{j=1}^J$. The complete GBT algorithm is provided in Appendix F. Under these settings, we obtain the following result:

**Corollary 2.** *Under the same assumptions as Theorem 3, there exist universal constants $\{C_i\}$s such that, with probability at least $1 - \delta$ over the draw of $S_{\mathrm{tr}}$, we have*

$$\mathrm{smCE}(\sigma(\bar{g}^{(T)}), \mathcal{D}) \leq \frac{L_n(g^{(0)})}{\gamma BwT} + \frac{wB}{8\gamma} + \frac{C_2}{\sqrt{n}} + C_3 wT\sqrt{\frac{2^m \log nd}{n}} + 2\sqrt{\frac{\log(2/\delta)}{n}}.$$

We find that increasing the number of steps $T$ reduces the training smooth CE, but also enlarges the function class, as the Rademacher complexity grows at rate $\mathcal{O}(wT)$. This highlights a trade-off between lowering training smooth CE and controlling model complexity. Corollary 3 shows that with appropriate hyperparameters, one can attain any target precision $\epsilon$ for the smooth CE.

In addition, bounding the norm of the functional gradient provides a generalization guarantee for test accuracy (Nitanda & Suzuki, 2018). For completeness, we include in Appendix F a formal upper bound on the misclassification rate, $P_{(X,Y)\sim\mathcal{D}}[(2Y - 1)\bar{g}^{(T)}(X) \leq 0]$.

Combining this with Theorem 2, we conclude that with suitable choices of $T$ and $n$, both the smooth CE and the misclassification rate can be controlled to within any target precision $\epsilon$.

**Corollary 3.** *Under assumptions in Corollary 2, for any $\epsilon > 0$, if the hyperparameters satisfy:*

$$T = \Omega(\gamma^{-2}\epsilon^{-2}), \quad w = \Theta(\gamma^{-1}\epsilon^{-1}T^{-1}), \quad n = \tilde{\Omega}(\gamma^{-2}\epsilon^{-4}),$$

*then, with probability at least $1 - \delta$, the averaged predictor $\bar{g}^{(T)}$ satisfies $\mathrm{smCE}(\sigma(\bar{g}^{(T)}), \mathcal{D}) \leq \epsilon$ and $P_{(X,Y)\sim\mathcal{D}}[(2Y-1)\bar{g}^{(T)}(X) \leq 0] \leq \epsilon$.*

Here, $\tilde{\Omega}(\cdot)$ hides logarithmic factors in the big-$\Omega$ notation. Corollary 3 shows that $\epsilon$-smooth CE and $\epsilon$-classification error can be achieved with $\mathcal{O}(1/\epsilon^2)$ iterations of GBT. To the best of our knowledge, this is the first theoretical analysis of GBTs that jointly accounts for both test accuracy and smooth CE. We complement our theory with numerical experiments in Appendix K, which examine how the number of iterations and training sample size affect smooth CE and prediction accuracy.

## 4.2 CASE STUDY II: KERNEL BOOSTING

We next consider *kernel boosting* (Wei et al., 2017), where the functional gradient is approximated by functions in an RKHS. Let $(\mathcal{H}, \langle \cdot, \cdot \rangle_{\mathcal{H}})$ be an RKHS associated with the kernel $k : \mathcal{X} \times \mathcal{X} \to \mathbb{R}$. The functional gradient over the training dataset is approximated as

$$\underset{\phi \in \mathcal{H}, \|\phi\|_{\mathcal{H}} \leq 1}{\arg\max} \left\langle \nabla_g L_n(g^{(t)}), \phi \right\rangle_{L_2(S_n)} = \frac{\mathcal{T}_k \nabla_g L_n(g)}{\|\mathcal{T}_k \nabla_g L_n(g)\|_{\mathcal{H}}},$$

where $\mathcal{T}_k \nabla_g L_n(g) := \frac{1}{n} \sum_{i=1}^n k(X_i, \cdot) \nabla_g \ell_{\mathrm{ent}}(\sigma(g(X_i)), Y_i)$ is the empirical kernel operator. In kernel boosting, we set $h_t = -\mathcal{T}_k \nabla_g L_n(g^{(t)})$ in Eq. (4) and use the update rule:

$$g^{(t+1)} = g^{(t)} - w_t \mathcal{T}_k \nabla_g L_n(g^{(t)}).$$

To analyze this, we introduce the following normalized margin assumption:

**Assumption 3.** *Given a dataset $S_{\mathrm{tr}}$, there exists a function $\phi \in \mathcal{H}$ and a constant $\gamma > 0$ such that for all $(X_i, Y_i) \in S_{\mathrm{tr}}$, $(2Y_i - 1)\phi(X_i) \geq \gamma$.*

This assumption is essentially equivalent to Assumption 2 in the GBT analysis. Setting $q$ in Assumption 2 as a unit vector yields Assumption 3. Conversely, taking convex combinations of $(2Y_i - 1)\phi(X_i) \geq \gamma$ in Assumption 3 recovers Assumption 2.

Under Assumption 3, the $L_1$ norm of the functional gradient is bounded as follows:

$$\|\nabla_g \ell_{\mathrm{ent}}(g(X), Y)\|_{L_1(S_n)} = \frac{1}{n} \sum_{i=1}^n |\nabla_g \ell_{\mathrm{ent}}(g(X_i), Y_i)| \leq \frac{1}{\gamma} \|\mathcal{T}_k \nabla_g L_n(g)\|_{\mathcal{H}}.$$

All proofs for this section are provided in Appendix G. Combining this with Eq. (5), we can control the smooth CE via $\|\mathcal{T}_k \nabla_g L_n(g)\|_{\mathcal{H}}$. Using this relation, we obtain the following result.

**Theorem 4.** *Suppose Assumption 3, $\sup_{x,x'\in\mathcal{X}} k(x, x') \leq \Lambda$, and $\|g^{(0)}\|_{\mathcal{H}} \leq \Lambda'$ hold. When using the constant stepsize $w_t = w$ which satisfies $w < 4/\Lambda$, the average of function $\bar{g}^{(T)} = \frac{1}{T} \sum_{t=0}^{T-1} g^{(t)}$ satisfies*

$$\mathrm{smCE}^\sigma(\bar{g}^{(T)}, S_{\mathrm{tr}}) \leq \frac{1}{\gamma} \sqrt{\frac{L_n(g^{(0)})}{wT}}.$$

*Additionally, there exist universal constants $\{C_i\}$s such that with probability at least $1 - \delta$ over the draw of $S_{\mathrm{tr}}$, we have:*

$$\mathrm{smCE}(\sigma(\bar{g}^{(T)}), \mathcal{D}) \leq \frac{1}{\gamma} \sqrt{\frac{L_n(g^{(0)})}{wT}} + \frac{C_2}{\sqrt{n}} + C_4(\Lambda' + \sqrt{2wTL_n(g^{(0)})})\sqrt{\frac{\Lambda}{n}} + 2\sqrt{\frac{\log \frac{2}{\delta}}{n}}.$$

Although covering number bounds require assumptions on kernel eigenvalue decay, here we use the Rademacher complexity bound from Theorem 2, which yields a simpler result.

We observe that increasing the number of steps $T$ decreases the training smooth CE but simultaneously enlarges the function class, since the Rademacher complexity of predictors (the third term on the

right-hand side) grows with the norm of $\bar{g}^{(t)}$ at rate $\mathcal{O}(\sqrt{wT})$. This highlights a trade-off between reducing training smooth CE and controlling model complexity.

Since the norm of the functional gradient also upper bounds the misclassification rate, we show—analogous to Corollary 3—that suitable hyperparameter choices guarantee any target precision $\epsilon$ for both the smooth CE and the misclassification rate.

**Corollary 4.** *Suppose assumptions in Theorem 4 hold. For any $\epsilon > 0$, if the hyperparameters satisfy:*

$$T = \Omega(\gamma^{-2}\epsilon^{-2}), \quad w = \Theta(\gamma^{-2}\epsilon^{-2}T^{-1}), \quad n = \tilde{\Omega}(\gamma^{-2}\epsilon^{-4}),$$

*then, with probability at least $1 - \delta$, the average of function $\bar{g}^{(T)}$ satisfies $\mathrm{smCE}(\sigma(\bar{g}^{(T)}), \mathcal{D}) \leq \epsilon$ and $P_{(X,Y)\sim\mathcal{D}}[(2Y-1)\bar{g}^{(T)}(X) \leq 0] \leq \epsilon$.*

Corollary 4 establishes that both $\epsilon$-smooth CE and $\epsilon$-classification error can be achieved using $\mathcal{O}(1/\epsilon^2)$ iterations of kernel boosting. This result is grounded in the observation that bounding the functional gradient norm leads to a small smooth CE and a misclassification rate.

### 4.3 CASE STUDY III: TWO-LAYER NEURAL NETWORK

The next example is the two-layer neural network (NN). While NNs are typically trained by gradient descent on their parameters, recent work (Nitanda et al., 2019) shows that under certain hyperparameter settings—such as output scaling and a sufficiently large number of neurons—two-layer NNs behave similarly to kernel boosting with the neural tangent kernel (NTK).

We adopt the setting of Nitanda et al. (2019). Define the logit function $g_\theta : \mathcal{X} \to \mathbb{R}$ by a two-layer NN $g_\theta(x) = \frac{1}{m^\beta}\sum_{r=1}^m a_r\phi(\theta_r \cdot x)$, where $m > 0$ is the number of hidden units, $\beta \in [0,1]$ is a scaling exponent, and $\phi : \mathbb{R} \to \mathbb{R}$ is a smooth activation function (e.g., sigmoid, tanh). The weights $\{a_r\}_{r=1}^m \in \{-1,1\}^m$ are fixed, while the parameters $\theta = \{\theta_r\}_{r=1}^m$ with $\theta_r \in \mathbb{R}^d$ are updated via gradient descent: $\theta^{(t+1)} = \theta^{(t)} - w\nabla_\theta L_n(g_{\theta^{(t)}})$ with constant stepsize $w > 0$.

We now give an informal statement of Theorem 2 from Nitanda et al. (2019), which provides guarantees for the functional gradient. Assume: (1) the activation function is smooth; (2) the initial parameters are sampled from a sub-Gaussian distribution; (3) the NTK-transformed data are separable with margin $\gamma$; (4) the stepsize is sufficiently small; and (5) the number of hidden units $m$ is large enough. Then, for $T \leq \frac{m\gamma^2 K_3}{w}$, there exist constants $K_1$, $K_2$, and $K_3$ such that, for all $\beta \in [0,1]$, with probability at least $1 - \delta$ over the random initialization, the following bound holds:

$$\frac{1}{T}\sum_{t=0}^{T-1} \|\nabla_g \ell_{\mathrm{ent}}(g_{\theta^{(t)}}(X), Y)\|_{L_1(S_n)}^2 \leq \frac{K_1}{\gamma^2 T}\left(\frac{m^{2\beta-1}}{w} + K_2\right). \tag{6}$$

See Appendix H for the formal statement. Then, similar to Theorem 4, we have

$$\min_{t\in\{0,\ldots,T-1\}} \mathrm{smCE}^\sigma(g_{\theta^{(t)}}, S_{\mathrm{tr}}) \leq \frac{1}{T}\sum_{t=0}^{T-1} \mathrm{smCE}^\sigma(g_{\theta^{(t)}}, S_{\mathrm{tr}}) \leq \sqrt{\frac{K_1}{\gamma^2 T}\left(\frac{m^{2\beta-1}}{w} + K_2\right)}.$$

Combining the Rademacher complexity estimate of Nitanda et al. (2019) with Theorem 2, we derive upper bounds on the population smooth CE and misclassification rate, analogous to the kernel boosting setting (see Appendix H for details). As in Theorem 4, the bound reveals a trade-off in $T$: increasing $T$ reduces the training smooth CE but enlarges the complexity term, which grows at rate $\mathcal{O}(\sqrt{wT})$. Similar to Corollary 4, we further show that with suitable hyperparameters, both $\epsilon$-smooth CE and $\epsilon$-classification error can be guaranteed.

**Corollary 5** (Informal)**.** *Suppose the same assumptions as those for Eq. (6) hold. If for any $\epsilon > 0$, the hyperparameters satisfy one of the following:*

*(i) $\beta \in [0,1)$, $m = \Omega(\gamma^{\frac{-2}{1-\beta}}\epsilon^{\frac{-1}{1-\beta}})$, $T = \Omega(\gamma^{-2}\epsilon^{-2})$, $w = \Theta(\gamma^{-2}\epsilon^{-2}T^{-1}m^{2\beta-1})$, $n = \tilde{\Omega}(\gamma^{-2}\epsilon^{-4})$,*
*(ii) $\beta = 0$, $m = \Theta\left(\gamma^{-2}\epsilon^{-3/2}\log(1/\epsilon)\right)$, $T = \Theta\left(\gamma^{-2}\epsilon^{-1}\log^2(1/\epsilon)\right)$, $w = \Theta(m^{-1})$, $n = \tilde{\Omega}(\epsilon^{-2})$,*
*then with probability at least $1 - \delta$, gradient descent with the stepsize $w$ finds a parameter $\theta^{(t)}$ satisfying $\mathrm{smCE}(\sigma(g_{\theta^{(t)}}), S_{\mathrm{tr}}) \leq \epsilon$ and $\mathbb{P}_{(X,Y)\sim\mathcal{D}}[(2Y-1)g_{\theta^{(t)}}(X) \leq 0] \leq \epsilon$ within $T$ iterations.*

See Appendix H for the formal statement with explicit constants. This result shows that both $\epsilon$-smooth CE and $\epsilon$-classification error can be achieved within $\mathcal{O}(1/\epsilon^2)$ or $\mathcal{O}(1/\epsilon)$ iterations, assuming the

NTK-transformed data are separable with margin $\gamma$. Between the two hyperparameter settings, (ii) attains the same error target with fewer iterations $T$ by increasing the number of hidden units $m$, and further yields improved complexity compared with Corollary 4 for kernel boosting. Appendix K provides empirical results illustrating how smooth CE and accuracy vary with $T$ and $n$.

## 5 RELATED WORK

Various metrics have been proposed to quantify deviations from perfect calibration, including ECE (Guo et al., 2017; Rahaman et al., 2021; Minderer et al., 2021). The binning ECE has been widely used to estimate the conditional expectation $\mathbb{E}[Y|f(X)]$ in the ECE, which partitions the probability interval into discrete bins. Despite its popularity, binning ECE is sensitive to hyperparameters such as the number of bins, and it lacks both consistency and smoothness as a distance metric (Kumar et al., 2019; Nixon et al., 2019; Minderer et al., 2021). A broader comparison of ECE variants is given by Gruber & Buettner (2022). Błasiok et al. (2023) systematically studied calibration distances and formalized the *true distance from calibration* as a rigorous notion of deviation from perfect calibration. As an efficiently estimable surrogate, they proposed the smooth CE, building on Foster & Hart (2018); Kakade & Foster (2008). More recently, Hu et al. (2024) showed that smooth CE can be efficiently evaluated in practice. Importantly, Błasiok et al. (2023) proved that smooth CE provides both upper and lower bounds on binning ECE, making it a theoretically sound proxy. Hence, the results established here for smooth CE also yield implications for binning ECE; see Appendix B for details.

To achieve a well-calibrated prediction with a theoretical guarantee, many prior works focus on recalibration, such as binning-based recalibration methods (Gupta et al., 2020; Gupta & Ramdas, 2021; Kumar et al., 2019; Sun et al., 2023; Futami & Fujisawa, 2024). However, it has been reported that such post-processing loses the sharpness of the prediction, which sometimes leads to accuracy degradation (Kumar et al., 2018; Karandikar et al., 2021; Popordanoska et al., 2022; Marx et al., 2023; Kuleshov & Liang, 2015). To theoretically guarantee the calibration, our analysis takes a different perspective. We provide guarantees for the smooth CE without relying on such a post-processing approach similar to recalibration. This is achieved by combining a uniform convergence bound and a functional gradient characterization of the training smooth CE. Then, our analysis simultaneously guarantees both the smooth CE and accuracy for several practical algorithms.

Prior work has connected the post-processing gap to the population smooth CE (Błasiok et al., 2023; Błasiok et al., 2023), making it less applicable to algorithms trained on finite datasets. Similarly, although Gruber & Buettner (2022) discussed post-processing, their approach faces the same limitation. In contrast, our analysis directly targets the smooth CE from finite data. From a generalization perspective, Futami & Fujisawa (2024) developed algorithm-dependent bounds for binning ECE using information-theoretic techniques. By contrast, we derive a uniform convergence bound for smooth CE, which applies more broadly beyond binning. Moreover, their generalization bound for ECE converges at a rate $\mathcal{O}(\log n/n^{1/3})$, which is slower than our rate. We conjecture that this slower convergence arises from the nonparametric estimation of conditional probabilities via binning, whereas smooth CE avoids such estimation and hyperparameter dependence. Finally, their results focus only on the generalization gap and do not address when the training ECE itself becomes small.

Although in recent studies, new boosting algorithms have been proposed to improve various notions of calibration (Hebert-Johnson et al., 2018; Globus-Harris et al., 2023), our analysis provides a theoretical explanation for why existing gradient boosting already yields strong calibration performance. Unlike prior works on boosting, which primarily focus on accuracy on the basis of functional gradients (Zhang & Yu, 2005; Nitanda & Suzuki, 2018; Nitanda et al., 2019), we leverage functional gradients to analyze calibration, offering a novel perspective on the behavior of these algorithms in calibration. While our analysis assumes a constant stepsize, the choice of stepsize in boosting is crucial for achieving better performance (Telgarsky, 2013). Extending our calibration analysis beyond the constant stepsize setting is an important direction for future work.

In several works, the post-processing has been highlighted in achieving advanced calibration objectives, such as multicalibration (Hebert-Johnson et al., 2018; Hansen et al., 2024; Błasiok et al., 2023; Globus-Harris et al., 2023). Whereas our analysis focuses on smooth CE, this metric is closely related to these variants of CE from the viewpoint of low-degree calibration (Gopalan et al., 2022). We believe our findings may offer new insights into these metrics, which we leave for future exploration.

## 6 CONCLUSION AND LIMITATION

This work presented the first rigorous guarantees on the smooth CE for several widely used learning algorithms. Our analysis proceeds in two stages: we first derive a uniform convergence bound for the smooth CE, and then upper-bound the training smooth CE via functional gradients. We demonstrate that algorithms that can control the functional gradient simultaneously achieve a small smooth CE and misclassification rate. Despite these contributions, several limitations remain. First, our bounds are derived using a uniform bound over the post-processing function $h$, not the Lipschitz function class, which may result in loose estimates. More refined analyses that better align with practical performance would be valuable. Moreover, our analysis relies on a strong margin assumption, which requires a well-separated data distribution. Since calibration is a rather weak notion compared with accuracy, extending the analysis to more realistic and weaker conditions is an important direction for future work. Finally, our analysis is limited to binary classification as the smooth CE is designed for this setting. Extending our analysis to the multiclass setting should be explored in future work.

## REPRODUCIBILITY STATEMENT

This paper is primarily theoretical, and all technical details and complete proofs of our results are provided in the Appendix. In addition, for the supplementary experimental analyses presented in the Appendix, we provide the corresponding source code in the Supplementary Material to facilitate reproducibility.

## ACKNOWLEDGEMENTS

This work was supported by Japan Science and Technology Agency (JST) as part of Adopting Sustainable Partnerships for Innovative Research Ecosystem (ASPIRE), Grant Number JPMJAP25B1. FF was supported by JST, PRESTO Grant Number JPMJPR22C8, Japan. This research is supported by the National Research Foundation, Singapore, Infocomm Media Development Authority under its Trust Tech Funding Initiative, and the Ministry of Digital Development and Information under the AI Visiting Professorship Programme (award number AIVP-2024-004). Any opinions, findings and conclusions or recommendations expressed in this material are those of the author(s) and do not reflect the views of National Research Foundation, Singapore, Infocomm Media Development Authority, and the Ministry of Digital Development and Information.

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

## A    ADDITIONAL FACTS ABOUT PROPER LOSSES

Here, we describe the general form of a proper loss function and its relation to the post-processing gap. Given a proper loss function $\ell$, it can always be represented in terms of a convex function as follows:

$$\ell(p, y) = -\phi(p) - \nabla\phi(p) \cdot (y - p),$$

where $\phi : [0, 1] \to \mathbb{R}$ is a convex function and $\nabla\phi(p)$ denotes a subgradient at $p$. This representation is known as the *Savage representation* (Gneiting & Raftery, 2007). Błasiok et al. (2023) further showed that, in the binary case, the subgradient satisfies $\nabla\phi(p) = \ell(p, 0) - \ell(p, 1)$, and referred to $\nabla\phi(p)$ as the *dual* of $p$ (dual($p$)). Hereinafter, we assume that $\phi$ is differentiable, since the convex functions corresponding to commonly used losses such as the squared loss and the log loss are differentiable.

Table 1: Savage representations for the cross-entropy Loss and squared loss

|  | cross-entropy Loss | Squared Loss |
|---|---|---|
| $\ell(p, y)$ | $-y \ln p - (1 - y) \ln(1 - p)$ | $(y - p)^2$ |
| $\phi(v)$ | $p \ln p + (1 - p) \ln(1 - p)$ | $p(p - 1)$ |
| dual$(p)$ | $\ln\left(\frac{p}{1-p}\right)$ | $2p - 1$ |
| $\psi(s)$ | $\ln(1 + e^s)$ | $\begin{cases} 0, & s < -1 \\ \frac{(s+1)^2}{4}, & -1 \le s \le 1 \\ s, & s > 1 \end{cases}$ |
| $\ell^\psi(s, y)$ | $\ln(1 + e^s) - y \cdot s$ | $\begin{cases} -ys, & s < -1 \\ (y - (s+1)/2)^2, & -1 \le s \le 1 \\ (1 - y)s, & t > 1 \end{cases}$ |
| $\text{pred}_\psi(s)(= \nabla\psi(s))$ | $\frac{e^s}{1+e^s}$ | $\begin{cases} 0, & s < -1 \\ \frac{s+1}{2}, & -1 \le s \le 1 \\ 1, & s > 1 \end{cases}$ |

We define the convex conjugate of a function $\phi(p)$ as follows: for all $s \in \mathbb{R}$,

$$\psi(s) = \sup_{p \in [0,1]} \{s \cdot p - \phi(p)\},$$

where $\psi$ is a convex function. We refer to $s$ as a *score*, as will be explained later. Using this notation, we define the *dual loss* $\ell^\psi : \mathbb{R} \times \mathcal{Y} \to \mathbb{R}$ as

$$\ell^\psi(s, y) := \psi(s) - s \cdot y.$$

The score $s$ is linked to the probability $p$ via the dual mapping dual$(p) = \nabla\phi(p) = \ell(p, 0) - \ell(p, 1)$. By Fenchel–Young duality, this relationship is inverted as $p = \nabla\psi(s)$, which we also denote as $p = \text{pred}_\psi(s)$.

With these definitions, the proper loss can be equivalently written as

$$\ell(p, y) = \ell^\psi(\text{dual}(p), y) = \psi(\text{dual}(p)) - \text{dual}(p) \cdot y.$$

For details and proofs, see Błasiok et al. (2023). We summarize the key properties of these expressions in Table 1.

Similarly, we can define the dual post-processing gap for the proper scoring rule (Błasiok et al., 2023):

**Definition 3** (Dual-post processing gap)**.** *Assume that $\Psi$ is a differentiable and convex function with derivative $\nabla\psi(t) \in [0, 1]$ for all $t \in \mathbb{R}$. Assume that $\psi$ is $\lambda$ smooth function. Given $\psi, \ell^\psi$, $g : \mathcal{X} \to \mathbb{R}$, and $\mathcal{D}$, we define the dual post-processing gap as*

$$\text{pGap}^{(\psi,\lambda)}(g, \mathcal{D}) := \mathbb{E}[\ell^\psi(g(X), Y)] - \inf_{h \in \text{Lip}_1(\mathbb{R}, [-1/\lambda, 1/\lambda])} \mathbb{E}[\ell^\psi(g(X) + h(g(X)), Y)]$$

When considering the cross-entropy loss, this dual post-processing gap corresponds to improving the logit function.

**Definition 4** (Dual smooth calibration). *Consider the same setting as the definition of the dual-post processing gap. Given $\psi$ and $g$, we define $f(\cdot) = \nabla\psi(g(\cdot))$. We then define the dual calibration error of $g$ as*

$$\text{smCE}^{(\psi,\lambda)}(g,\mathcal{D}) := \sup_{h \in \text{Lip}_{L=\lambda}(\mathbb{R},[-1,1])} \mathbb{E}\eta(g(X)) \cdot (Y - f(X))$$

Then, similarly to the relationship between the smooth ECE and the post-processing gap, the following holds: if $\psi$ is a $\lambda$-smooth function, then we have

$$\text{smCE}^{(\psi,\lambda)}(g,\mathcal{D})^2/2 \leq \lambda\text{pGap}^{(\psi,\lambda)}(g,\mathcal{D})^2/2 \leq \text{smCE}^{(\psi,\lambda)}(g,\mathcal{D})$$

and

$$\text{smCE}(f,\mathcal{D}) \leq \text{smCE}^{(\psi,\lambda)}(g,\mathcal{D})$$

holds.

In the case of the cross-entropy loss, we have $\psi(s) = \log(1 + e^s)$, which is $1/4$-smooth. For the squared loss, $\psi$ is $1/2$-smooth. Therefore, the dual smooth calibration and the smooth calibration are essentially equivalent in both cases. We remark that the optimality of the constants appearing in smooth CE in Eq. (1) and dual smooth CE in Eq. (2) is discussed in Appendix D of Błasiok et al. (2023).

## B    DISCUSSION ABOUT CALIBRATION METRICS

Here, we introduce several definitions of calibration metrics and their relationship to the smooth CE, which is the primary focus of our paper.

### B.1    TRUE DISTANCE CALIBRATION AND SMOOTH CE

Given a distribution $\mathcal{D}$ and predictor $f$, we consider the distribution induced by $f$, which is the joint distribution of prediction-label pairs $(f(x), y) \in [0,1] \times \{0,1\}$, denoted by $\mathcal{D}_f$.

**Definition 5** (Perfect calibration (Błasiok et al., 2023)). *We say that a prediction-label distribution $\Gamma$ over $[0,1] \times \{0,1\}$ is perfectly calibrated if $\mathbb{E}_{(v,y)\sim\Gamma}[y|v] = v$. Moreover, given $\mathcal{D}$ over $\mathcal{X} \times \{0,1\}$, we say that $f$ is perfectly calibrated with respect to $\mathcal{D}$ if $\mathcal{D}_f$ is perfectly calibrated.*

We denote the set of perfectly calibrated predictors with respect to $\mathcal{D}$ by $\text{cal}(\mathcal{D})$.

Next, we introduce the true calibration distance, defined as the distance to the closest calibrated predictor:

**Definition 6** (True calibration distance). *The true distance of a predictor $f$ from calibration is defined as*

$$\text{dCE}_{\mathcal{D}}(f) := \inf_{g \in \text{cal}(\mathcal{D})} \mathbb{E}_{\mathcal{D}}|f(x) - g(x)|.$$

This is the $l_1$ metric, which possesses many desirable properties for measuring calibration; see Błasiok et al. (2023) for details. However, it is not practically usable because $\text{cal}(\mathcal{D})$ may be non-convex, and the metric depends on the domain $\mathcal{X}$, whereas calibration metrics typically depend only on $\mu(\mathcal{D}, f)$.

Błasiok et al. (2023) proposed that any calibration metric should satisfy the following three criteria:

**(i) Prediction-only access**    A calibration metric should depend only on the distribution over $(f(x), y) \sim \mathcal{D}_f$, and not on the distribution over $\mathcal{X}$. If it meets this requirement, we say that a calibration metric $\mu$ satisfies the **Prediction-only access** (PA) property. Note that the true calibration distance depends on $(x, f(x), y)$ and is therefore called the sample access (SA) model.

**(ii) Consistency**

**Definition 7.** *For $c > 0$, a calibration metric $\mu$ is said to satisfy $c$-robust completeness if there exists a constant $a \geq 0$ such that for every distribution $D$ on $\mathcal{X} \times \{0, 1\}$, and predictor $f \in \mathcal{F}$*

$$\mu(\mathcal{D}, f) \leq a \left(\mathrm{dCE}_{\mathcal{D}}(f)\right)^c.$$

**Definition 8.** *For $s > 0$, a calibration metric $\mu$ satisfies $s$-robust soundness if there exists a constant $b \geq 0$ such that for every distribution $D$ on $\mathcal{X} \times \{0, 1\}$, and predictor $f \in \mathcal{F}$,*

$$\mu(\mathcal{D}, f) \geq b \left(\mathrm{dCE}_{\mathcal{D}}(f)\right)^s.$$

**Definition 9.** *A calibration metric $\mu$ is said to be $(c, s)$-consistent if it satisfies both $c$-robust completeness and $s$-robust soundness.*

Thus, consistent calibration metrics are polynomially related to the true distance from calibration. Błasiok et al. (2023) showed that any PA model must satisfy $s/c \geq 2$.

**(iii) Efficiency**   A calibration metric $\mu$ is said to be efficient if it can be computed to accuracy $\epsilon$ in time $\mathrm{poly}(1/\epsilon)$ using $\mathrm{poly}(1/\epsilon)$ random samples from $\mathcal{D}_f$.

Błasiok et al. (2023) showed that the smooth CE satisfies all of these properties. In particular, with respect to consistency, the smooth CE satisfies

$$\mathrm{smCE}(f, \mathcal{D}) \leq \mathrm{dCE}_{\mathcal{D}}(f) \leq 4\sqrt{2\mathrm{smCE}(f, \mathcal{D})}$$
$$\Leftrightarrow \frac{1}{32}\mathrm{dCE}_{\mathcal{D}}(f)^2 \leq \mathrm{smCE}(f, \mathcal{D}) \leq \mathrm{dCE}_{\mathcal{D}}(f)$$

which means that the smooth CE is $(c = 1, s = 2)$-consistent and achieves $s/c = 2$.

## B.2   ECE AND BINNING ESTIMATOR

Recall the definition of the expected calibration error (ECE):

$$\mathrm{ECE}_{\mathcal{D}}(f) := \mathbb{E}\left[|\mathbb{E}[Y|f(X)] - f(X)|\right].$$

The binning estimator is commonly used to approximate this. Given a partition $\mathcal{I} = \{I_1, \ldots, I_m\}$ of $[0, 1]$ into intervals, we define:

$$\mathrm{binnedECE}_{\mathcal{D}}(\Gamma, \mathcal{I}) = \sum_{j \in [m]} |\mathbb{E}_{(V, Y \sim \Gamma)}[(V - Y)\mathbb{1}(V \in I_j)]|.$$

It has been shown in Lemma 4.7 of Błasiok et al. (2023) that

$$\mathrm{dCE}_{\mathcal{D}}(f) \leq \mathrm{ECE}_{\mathcal{D}}(f)$$

demonstrating robust soundness. However, ECE does not satisfy robust completeness; Błasiok et al. (2023) proved (Lemma 4.8) that for any $\epsilon \in \mathbb{R}^+$, there exists a distribution $\mathcal{D}$ such that $\mathrm{dCE}_{\mathcal{D}}(f) \leq \epsilon$ but $\mathrm{ECE}_{\mathcal{D}}(f) \geq 1/2 - \epsilon$. This also highlights the discontinuity of ECE, which hinders its estimation from finite samples.

Błasiok et al. (2023) showed that by accounting for bin widths, the binning ECE satisfies consistency:

$$\mathrm{intCE}(f) := \min_{\mathcal{I}}(\mathrm{binnedECE}_{\mathcal{D}}(\Gamma, \mathcal{I}) + w_{\Gamma}(\mathcal{I}))$$

where

$$w_{\Gamma}(\mathcal{I}) := \sum_{j \in [m]} |\mathbb{E}_{(V, Y \sim \Gamma)}w(I_j)\mathbb{1}(V \in I_j)|$$

and $w(I)$ denotes the width of interval $I$.

Then, the following holds (Theorem 6.3 in Błasiok et al. (2023)):

$$\mathrm{dCE}_{\mathcal{D}}(f) \leq \mathrm{intCE}(f) \leq 4\sqrt{\mathrm{dCE}_{\mathcal{D}}(f)}$$

Thus, intCE satisfies $(1/2, 1)$-consistency. Błasiok et al. (2023) also provided an estimator for intCE$(\Gamma)$.

As we have seen, bounding the smooth CE leads to a bound on $\mathrm{dCE}_{\mathcal{D}}(f)$, which in turn bounds intCE$(f)$—an optimized estimator for the binned ECE.

Finally, we remark that since $\mathbb{E}[Y|f = v]$ is continuous, we can relate the ECE and the smooth CE as follows:

**Theorem 5.** *Suppose that $\mathbb{E}[Y|f = v]$ satisfies L-Lipschitz continuity. Then the following relation holds:*

$$\frac{1}{2}\mathrm{smCE}(f, \mathcal{D}) \leq \mathrm{ECE}_{\mathcal{D}}(f) \leq (1 + 2\sqrt{2(1 + L)})\sqrt{2\mathrm{smCE}(f, \mathcal{D})}.$$

Thus, controlling the smooth CE also leads to controlling the ECE when the underlying conditional probability function is continuous.

*Proof.* We first prove the following inequality:

$$\mathrm{ECE}_{\mathcal{D}}(f) \leq \mathrm{binnedECE}_{\mathcal{D}}(\Gamma, \mathcal{I}) + (1 + L)w_{\Gamma}(\mathcal{I}). \tag{7}$$

This can be derived using Lemma 4 in Futami & Fujisawa (2024). For completeness, we outline the key steps of the analysis below:

$$
\begin{aligned}
\mathrm{ECE}_{\mathcal{D}}(f) = \mathbb{E}\left[|\mathbb{E}[Y|f = V] - V|\right] &= \sum_{i=1}^{m} P(V \in I_i)\mathbb{E}[|\mathbb{E}[Y|f = V] - V| \,|\, V \in I_i] \\
&= \sum_{i=1}^{m} P(V \in I_i)\mathbb{E}[|\mathbb{E}[Y|f = V] - \mathbb{E}[V|V \in I_i] \\
&\quad + \mathbb{E}[V|V \in I_i] - V| \,|\, V \in I_i] \\
&\leq \sum_{i=1}^{m} P(V \in I_i)\mathbb{E}[|\mathbb{E}[Y|V] - \mathbb{E}[Y|V \in I_i]| \,|\, V \in I_i] \\
&\quad + \sum_{i=1}^{m} P(V \in I_i)\mathbb{E}[|\mathbb{E}[Y|V \in I_i] - \mathbb{E}[V|V \in I_i]| \,|\, V \in I_i] \\
&\quad + \sum_{i=1}^{m} P(V \in I_i)\mathbb{E}[|\mathbb{E}[V|V \in I_i] - V| \,|\, V \in I_i].
\end{aligned}
$$

As for the second term, it is evident that

$$\sum_{i=1}^{m} P(V \in I_i)\mathbb{E}|\mathbb{E}[Y|V \in I_i] - \mathbb{E}[V|V \in I_i]| = \mathrm{binnedECE}_{\mathcal{D}}(\Gamma, \mathcal{I})$$

and as for the third term,

$$\sum_{i=1}^{m} P(V \in I_i)\mathbb{E}[|\mathbb{E}[V|V \in I_i] - V| \,|\, V \in I_i] \leq \sum_{i=1}^{m} \mathbb{E}[w(I_i)\mathbb{1}_{V \in I_i}] = w(\Gamma).$$

Finally, the first term can be bounded by using the Lipschitz continuity,

$$
\begin{aligned}
\sum_{i=1}^{m} &P(V \in I_i)\mathbb{E}[|\mathbb{E}[Y|V] - \mathbb{E}[Y|V \in I_i]| \,|\, V \in I_i] \\
&\leq \sum_{i=1}^{m} P(V \in I_i)\mathbb{E}[|\mathbb{E}[Y|V] - \mathbb{E}[Y|V_{\mathcal{I}}]| \,|\, V \in I_i] \\
&\quad + \sum_{i=1}^{m} P(V \in I_i)\mathbb{E}[|\mathbb{E}[Y|V_{\mathcal{I}}] - \mathbb{E}[Y|V \in I_i]| \,|\, V \in I_i] \\
&\leq L \sum_{i=1}^{m} P(V \in I_i)\mathbb{E}[|V - \mathbb{E}[V|V \in I_i]| \,|\, V \in I_i] \leq Lw(\Gamma)
\end{aligned}
$$

where

$$V_{\mathcal{I}} \coloneqq \sum_{i=1}^{m} V_{I_i} \cdot \mathbb{1}_{V \in I_i} = \sum_{i=1}^{m} \mathbb{E}[V|V \in I_i] \cdot \mathbb{1}_{V \in I_i}.$$

The term $\mathbb{E}[|\mathbb{E}[Y|V_{\mathcal{I}}] - \mathbb{E}[Y|V \in I_i]||V \in I_i] = 0$ holds by the definition of conditional expectation. Thus, we have established Eq. (7).

Next, we consider taking the infimum over all partitions $\mathcal{I}$. By combining Claim 6.6 and Lemma 6.7 in Błasiok et al. (2023) and Eq. (7), we obtain:

$$\mathrm{ECE}_{\mathcal{D}}(f) \leq (1 + 2/\epsilon)\underline{\mathrm{dCE}_{\mathcal{D}}(f)} + (1 + L)\epsilon$$

where $\underline{\mathrm{dCE}_{\mathcal{D}}(f)}$ denotes the lower calibration distance (see Błasiok et al. (2023) for the formal definition), and $\epsilon > 0$ is an arbitrary width parameter. By setting $\epsilon = \sqrt{\frac{2}{1+L}\underline{\mathrm{dCE}_{\mathcal{D}}(f)}}$ and using the fact that $\underline{\mathrm{dCE}_{\mathcal{D}}(f)} \leq 2\mathrm{smCE}(f, \mathcal{D})$ from Theorem 7.3 in Błasiok et al. (2023), we obtain the desired result. $\qquad\square$

We remark that if the underlying conditional distribution satisfies a continuity condition, then the ECE becomes a consistent calibration metric.

### B.3 OTHER METRICS

As discussed in Błasiok et al. (2023) and Gopalan et al. (2022), the smooth CE is a special case of the *weighted calibration error* introduced by Gopalan et al. (2022).

**Definition 10** (Weighted Calibration Error). *Let $\mathcal{M}$ be a class of functions $h : [0, 1] \to \mathbb{R}$. The calibration error relative to $\mathcal{M}$ is defined as*

$$\mathrm{CE}_{\mathcal{D}}(f, \mathcal{M}) \coloneqq \sup_{h \in \mathcal{M}} \mathbb{E}\left[h(f(X)) \cdot (Y - f(X))\right] = \sup_{h \in \mathcal{M}} \langle h(f(X)), Y - f(X)\rangle_{L_2(\mathcal{D})}.$$

In this view, the smooth CE corresponds to $\mathrm{smCE}(f, \mathcal{D}) = \mathrm{CE}_{\mathcal{D}}(f, \mathrm{Lip}_1([0, 1], [-1, 1]))$.

For the dual smooth CE, the inverse of the sigmoid function is $\sigma^{-1}(x) = \log(x/(1-x))$. When $g(x) = \sigma^{-1}(f(x))$, we have:

$$\mathrm{smCE}^{\sigma}(g, \mathcal{D}) = \mathrm{CE}_{\mathcal{D}}(f, \mathrm{Lip}_{1/4}(\mathbb{R}, [-1, 1]) \circ \sigma^{-1})$$

Another important function class is the RKHS $\mathcal{H}$ associated with a positive definite kernel $k$. This space is equipped with the feature map $\phi : \mathbb{R} \to \mathcal{H}$ satisfying $\langle h, \phi(v)\rangle_{\mathcal{H}} = h(v)$. The associated kernel $k : \mathbb{R} \times \mathbb{R} \to \mathbb{R}$ is defined by $k(u, v) = \langle \phi(u), \phi(v)\rangle_{\mathcal{H}}$.

Let $\mathcal{H}_1 \coloneqq \{h \in \mathcal{H} | \|h\|_{\mathcal{H}} \leq 1\}$. We define the kernel calibration error as $\mathrm{CE}_{\mathcal{D}}(f, \mathcal{H}_1)$.

Given samples $\{(v_1, y_1), \ldots, (v_n, y_n))\}$ where $v_i = f(x_i)$, the kernel CE under finite samples is defined as

$$\hat{\mathrm{CE}}_{\mathcal{D}}(f, \mathcal{W}_H)^2 \coloneqq \frac{1}{n^2}\sum_{i,j}(y_i - v_i)k(v_i, v_j)(y_j - v_j).$$

This was first proposed by Kumar et al. (2018) as the maximum mean calibration error (MMCE).

A key kernel is the Laplace (exponential) kernel $k_{\mathrm{Lap}}(u, v) = \exp(-|u - v|)$, for which it has been shown that

$$\frac{1}{3}\mathrm{smCE}(f, \mathcal{D}) \leq \mathrm{CE}_{\mathcal{D}}(f, \mathcal{H}_1).$$

On the other hand, the Gaussian kernel does not upper bound the smooth CE and has several limitations; see Błasiok et al. (2023) for details.

We numerically evaluate the behavior of MMCE with the Laplace kernel and smooth CE in Appendix K.

# C   PROOF OF SECTION 3

## C.1   PROOF OF THEOREM 1

Before the formal proof, we provide a proof sketch to highlight the key differences from standard generalization bounds as in Mohri et al. (2018); Wainwright (2019).

First, we reformulate the smooth CE as a linear convex optimization (Hu et al., 2024): let $v_i = f(x_i)$ and $\omega = (\omega_1, \ldots, \omega_n) \in \mathbb{R}^n$. Then $\mathrm{smCE}(f, S) = \max_\omega \sum_{i=1}^n (y_i - v_i)\omega_i/n$ subject to the constraints $|\omega_i| \leq 1, \forall i$, and $|\omega_i - \omega_j| \leq |v_i - v_j|, \forall i, j$. Let $\omega^*$ denote one such solution, and we define $\phi(f, z_i) := (y_i - v_i)\omega_i^*$. Then, we express $\phi(f, S) = \frac{1}{n}\sum_{i=1}^n \phi(f, z_i)$.

Although our analysis follows the structure of classical generalization bounds, the dependence among $\{\phi(f, z_i)\}_{i=1}^n$ induced by $\omega^*$ through the optimization precludes the use of standard concentration inequalities based on independence. Instead of Hoeffding's inequality, we use the bounded difference inequality to show $\sup_{f \in \mathcal{F}} |\phi(f, S_{\mathrm{te}}) - \phi(f, S_{\mathrm{tr}})| \leq \mathbb{E}_{S_{\mathrm{te}}, S_{\mathrm{tr}} \sim D^{2n}} \sup_{f \in \mathcal{F}} |\phi(f, S_{\mathrm{te}}) - \phi(f, S_{\mathrm{tr}})| + \sqrt{\log \delta^{-1}/n}$. This requires studying the stability of the above convex problem.

We would like to apply a symmetrization argument using i.i.d. Rademacher variables $\sigma_i \in \pm 1$ and evaluate $\mathbb{E}_{S_{\mathrm{te}}, S_{\mathrm{tr}} \sim D^{2n}} \mathbb{E}_\sigma \sup_{f \in \mathcal{F}} \frac{1}{n} \sum_{i=1}^n \sigma_i [\phi(f, Z_i') - \phi(f, Z_i)]$. However, this technique is not directly applicable in our setting due to dependencies among the terms $\{\phi(f, z_i)\}_{i=1}^n$.

To address this, we introduce $\{\sigma_i\}$s in a way that preserves the structure of the linear convex formulation of smooth CE. Since the resulting bound does not take the form of standard Rademacher complexity, we discretize the hypothesis class $\mathcal{F}$ and derive a covering number bound.

*Proof.* We begin by leveraging the reformulation of the smooth CE as a linear program introduced by Błasiok et al. (2023) (Theorem 7.14). Given a dataset $S = \{(x_i, y_i)\}_{i=1}^n$ and defining $v_i = f(x_i)$, it is known that the smooth CE on $S$ corresponds to the optimal value of the following optimization problem:

$$\mathrm{smCE}(f, S) = \max_{\{\omega_i\}} \frac{1}{n} \sum_{i=1}^n (y_i - v_i)\omega_i \tag{8}$$

subject to the constraints:

$$-1 \leq \omega_i \leq 1, \quad \forall i, \quad |\omega_i - \omega_j| \leq |v_i - v_j|, \quad \forall i, j.$$

Note that this optimization problem is a linear optimization problem in $n$ variables, and its feasible region is bounded (and closed) in $\mathbb{R}^n$. Therefore the existence of an optimal solution follows directly from standard results in analysis, that a continuous function on a compact set attains its maximum and minimum. Note that the optimal solution is not necessarily unique. Nevertheless, given a function $f$ and dataset $S$, the value $\mathrm{smCE}(f, S)$ is uniquely determined. Let $\omega_i^*$ denote one such optimal solution. Then, we can write:

$$\mathrm{smCE}(f, S) = \frac{1}{n} \sum_{i=1}^n (y_i - v_i)\omega_i^* = \frac{1}{n} \sum_{i=1}^n \phi(f, z_i)$$

where $z_i = (x_i, y_i)$. We denote $\mathrm{smCE}(f, S_{\mathrm{tr}}) = \phi(f, S_{\mathrm{tr}})$ and $\mathrm{smCE}(f, S_{\mathrm{te}}) = \phi(f, S_{\mathrm{te}})$.

As discussed above, although the optimizer $\omega^*$ may not be unique, the value $\mathrm{smCE}(f, S)$ is uniquely determined for a fixed $f$ and dataset $S$. Therefore, the uniform upper bound on the smooth CE can be controlled via the supremum over $f$ when the training and test datasets are fixed. We thus focus on the following inequality:

$$|\mathrm{smCE}(f, S_{\mathrm{te}}) - \mathrm{smCE}(f, S_{\mathrm{tr}})| \leq \sup_{f \in \mathcal{F}} |\phi(f, S_{\mathrm{te}}) - \phi(f, S_{\mathrm{tr}})|$$

Following the standard approach in uniform convergence analysis, we derive the convergence in expectation as follows:

**Lemma 1.** *Under the same setting as Theorem 1, with probability $1 - \delta$, we have*

$$\sup_{f \in \mathcal{F}} |\phi(f, S_{\text{te}}) - \phi(f, S_{\text{tr}})| \leq \mathbb{E}_{S_{\text{te}}, S_{\text{tr}} \sim D^{2n}} \sup_{f \in \mathcal{F}} |\phi(f, S_{\text{te}}) - \phi(f, S_{\text{tr}})| + 2\sqrt{\frac{\log \frac{1}{\delta}}{n}}.$$

The proof of this lemma is provided in Appedix C.2.

Note that this is not a consequence of Hoeffding's inequality, since

$$\phi(f, S_{\text{tr}}) = \frac{1}{n} \sum_{i=1}^{n} (y_i - v_i)\omega_i^* = \frac{1}{n} \sum_{i=1}^{n} \phi(f, z_i)$$

and thus the terms $\phi(f, z_i)$ are dependent through $\omega_i^*$, which is a solution of the linear program in Eq. (8). Since there are $n$ dependent variables $\phi(f, z_i)$, Hoeffding's inequality, which requires independence, is not applicable. To establish Lemma 1, we instead employ McDiarmid's inequality combined with a bounded difference argument; see the proof in Appendix C.2.

Our proof proceeds in three steps: (1) we introduce Rademacher random variables, (2) we evaluate the exponential moment to control the empirical process induced by these variables, and (3) we refine the exponential moment bound via a chaining argument.

We introduce Rademacher random variables to control the empirical process in Lemma 1. Following standard generalization analysis, we aim to apply the symmetrization technique. However, the standard formulation

$$\mathbb{E}_{(S_n, S'_n) \sim D^{2n}} \mathbb{E}_\sigma \sup_{f \in \mathcal{F}} \frac{1}{n} \sum_{i=1}^{n} [\sigma_i \phi(f, Z'_i) - \sigma_i \phi(f, Z_i)] \tag{9}$$

is not suitable for Step (2), as it complicates the evaluation of the exponential moment.

To understand this, let us explicitly express the uniform generalization gap as follows:

$$\mathbb{E}_{S_{\text{te}}, S_{\text{tr}} \sim D^{2n}} \sup_{f \in \mathcal{F}} \phi(f, S_{\text{te}}) - \phi(f, S_{\text{tr}}) \tag{10}$$

$$= \mathbb{E}_{(S_n, S'_n) \sim D^{2n}} \sup_{f \in \mathcal{F}} \left( \max_\omega \frac{1}{n} \sum_{i=1}^{n} (y_i - f(x_i))\omega_i - \max_{\omega'} \frac{1}{n} \sum_{i=1}^{n} (y'_i - f(x'_i))\omega'_i \right)$$

$$= \mathbb{E}_{(S_n, S'_n) \sim D^{2n}} \sup_{f \in \mathcal{F}} \max_\omega \min_{\omega'} \frac{1}{n} \sum_{i=1}^{n} [(y_i - f(x_i))\omega_i - (y'_i - f(x'_i))\omega'_i] \tag{11}$$

Here, we used the reformulation of the smooth CE over the training dataset via a Lipschitz function. For simplicity, we omit the constraints of the linear program. We now introduce Rademacher variables and demonstrate the difficulty in evaluating the exponential moment.

To simplify the discussion, let us consider the case $n = 2$. For a fixed dataset $S$ and function $f$, Eq. (9) under the expression of Eq. (10) is written as:

$$\mathbb{E}_\sigma \max_\omega \min_{\omega'} \frac{\sigma_1}{2} [(y_1 - f(x_1))\omega_1 - (y'_1 - f(x'_1))\omega'_1] + \frac{\sigma_2}{2} [(y_2 - f(x_2))\omega_2 - (y'_2 - f(x'_2))\omega'_2]$$

$$= \frac{1}{8} \max_\omega \min_{\omega'} (y_1 - f(x_1))\omega_1 + (y_2 - f(x_2))\omega_2 - (y'_1 - f(x'_1))\omega'_1 - (y'_2 - f(x'_2))\omega'_2$$

$$+ \frac{1}{8} \max_\omega \min_{\omega'} (y_1 - f(x_1))\omega_1 - (y_2 - f(x_2))\omega_2 - (y'_1 - f(x'_1))\omega'_1 + (y'_2 - f(x'_2))\omega'_2$$

$$+ \frac{1}{8} \max_\omega \min_{\omega'} -(y_1 - f(x_1))\omega_1 + (y_2 - f(x_2))\omega_2 + (y'_1 - f(x'_1))\omega'_1 - (y'_2 - f(x'_2))\omega'_2$$

$$+ \frac{1}{8} \max_\omega \min_{\omega'} -(y_1 - f(x_1))\omega_1 - (y_2 - f(x_2))\omega_2 + (y'_1 - f(x'_1))\omega'_1 + (y'_2 - f(x'_2))\omega'_2$$

$$\neq 0$$

This non-zero result indicates the challenge in applying exponential moment analysis for Step (2). In standard analysis, this type of expectation is typically zero, which enables the use of tools such as Massart's lemma. Hence, the standard symmetrization technique fails in this setting.

Therefore, we must develop an alternative symmetrization strategy. To illustrate the idea, we begin by introducing only $\sigma_1$ and consider the following expression:

Eq. (11)

$$
= \mathbb{E}_{(S_n, S'_n) \sim D^{2n}} \mathbb{E}_{\sigma_1} \sup_{f \in \mathcal{F}} \max_{\omega} \min_{\omega'} \Bigg[
$$

$$
\frac{\sigma_1}{n} \left( (y_1 - f(x_1)) \left( \frac{1 + \sigma_1}{2} \omega_1 + \frac{1 - \sigma_1}{2} \omega'_1 \right) - (y'_1 - f(x'_1)) \left( \frac{1 + \sigma_1}{2} \omega'_1 + \frac{1 - \sigma_1}{2} \omega_1 \right) \right)
$$

$$
+ \frac{1}{n} \sum_{i=2}^{n} \left( (y_i - f(x_i)) \omega_i - (y'_i - f(x'_i)) \omega'_i \right) \Bigg]
$$

Here, $\sigma_1$ is a Rademacher random variable. This equality holds because, when $\sigma_1 = 1$, the maximization and minimization are unchanged; when $\sigma_1 = -1$, the roles of $\omega_1$ and $\omega'_1$ are exchanged. However, since we are averaging over $S$ and $S'$, and these datasets are i.i.d., such swapping does not affect the distribution of the expectation. Therefore, the equality holds. This method of introducing Rademacher variables differs fundamentally from the standard symmetrization approach.

We then introduce i.i.d. Rademacher random variables $\{\sigma_i\}_{i=1}^{n}$ and consider the following expression:

Eq. (11)

$$
= \mathbb{E}_{(S_n, S'_n) \sim D^{2n}} \mathbb{E}_{\sigma} \sup_{f \in \mathcal{F}} \max_{\omega} \min_{\omega'}
$$

$$
\frac{1}{n} \sum_{i=1}^{n} \sigma_i \left[ (y_i - f(x_i)) \left( \frac{1 + \sigma_i}{2} \omega_i + \frac{1 - \sigma_i}{2} \omega'_i \right) - (y'_i - f(x'_i)) \left( \frac{1 + \sigma_i}{2} \omega'_i + \frac{1 - \sigma_i}{2} \omega_i \right) \right]
$$

An important property is that for fixed $f$ and dataset $S$, we have:

$$
\mathbb{E}_{\sigma} \max_{\omega} \min_{\omega'} \frac{1}{n} \sum_{i=1}^{n} \sigma_i \left[ (y_i - f(x_i)) \left( \frac{1 + \sigma_i}{2} \omega_i + \frac{1 - \sigma_i}{2} \omega'_i \right) \right. \tag{12}
$$

$$
\left. - (y'_i - f(x'_i)) \left( \frac{1 + \sigma_i}{2} \omega'_i + \frac{1 - \sigma_i}{2} \omega_i \right) \right] = 0.
$$

This cancellation becomes evident in the case $n = 2$: by expanding the left-hand side above, we obtain

$$
\frac{1}{4} \max_{\omega} \min_{\omega'} (y_1 - f(x_1)) \omega_1 + (y_2 - f(x_2)) \omega_2 - (y'_1 - f(x'_1)) \omega'_1 - (y'_2 - f(x'_2)) \omega'_2
$$

$$
+ \frac{1}{4} \max_{\omega} \min_{\omega'} (y_1 - f(x_1)) \omega_1 + (y'_2 - f(x'_2)) \omega_2 - (y'_1 - f(x'_1)) \omega'_1 - (y_2 - f(x_2)) \omega'_2
$$

$$
+ \frac{1}{4} \max_{\omega} \min_{\omega'} (y'_1 - f(x'_1)) \omega_1 + (y_2 - f(x_2)) \omega_2 - (y_1 - f(x_1)) \omega'_1 - (y'_2 - f(x'_2)) \omega'_2
$$

$$
+ \frac{1}{4} \max_{\omega} \min_{\omega'} (y'_1 - f(x'_1)) \omega_1 + (y'_2 - f(x'_2)) \omega_2 - (y_1 - f(x_1)) \omega'_1 - (y_2 - f(x_2)) \omega'_2
$$

$$
= 0.
$$

This is because the structure of the linear convex problem ensures symmetric cancellation when Rademacher variables are introduced. Therefore, we can proceed the analysis of the exponential moment of Step (2).

We define the integrated empirical process as

$$
R(\mathcal{F}, S_n, S'_n) := \mathbb{E}_{\sigma} \sup_{f \in \mathcal{F}} \max_{\omega} \min_{\omega'} \frac{1}{n} \sum_{i=1}^{n} \sigma_i \phi(f, Z_i, Z'_i),
$$

where

$$
\phi(f, Z_i, Z'_i) := (y_i - f(x_i)) \left( \frac{1 + \sigma_i}{2} \omega_i + \frac{1 - \sigma_i}{2} \omega'_i \right) - (y'_i - f(x'_i)) \left( \frac{1 + \sigma_i}{2} \omega'_i + \frac{1 - \sigma_i}{2} \omega_i \right).
$$

We now derive a bound using a variant of the Massart lemma (Mohri et al., 2018). For simplicity, assume the function class $\mathcal{F}$ has finite cardinality $|\mathcal{F}|$. This will later be replaced by a covering number. Then for all $\lambda > 0$, we have:

$$
\begin{aligned}
R(\mathcal{F}, S_n, S_n') &= \mathbb{E}_\sigma \sup_{f \in \mathcal{F}} \max_\omega \min_{\omega'} \frac{1}{n} \sum_{i=1}^n \sigma_i \phi(f, Z_i, Z_i') \\
&\leq \mathbb{E}_\sigma \frac{1}{\lambda} \log \sum_{f \in \mathcal{F}} \exp\left( \max_\omega \min_{\omega'} \frac{\lambda}{n} \sum_{i=1}^n \sigma_i \phi(f, Z_i, Z_i') \right) \\
&\leq \frac{1}{\lambda} \log \sum_{f \in \mathcal{F}} \mathbb{E}_\sigma \exp\left( \max_\omega \min_{\omega'} \frac{\lambda}{n} \sum_{i=1}^n \sigma_i \phi(f, Z_i, Z_i') \right) \\
&\leq \frac{1}{\lambda} \log \left( |\mathcal{F}| \cdot \mathbb{E}_\sigma \exp\left( \max_\omega \min_{\omega'} \frac{\lambda}{n} \sum_{i=1}^n \sigma_i \phi(f, Z_i, Z_i') \right) \right).
\end{aligned}
$$

Since the expectation of the exponential term is 0 from Eq. (12), we apply McDiarmid's inequality:

**Lemma 2.** *Under the above setting, we have*

$$
\mathbb{E}_\sigma \exp\left( \max_\omega \min_{\omega'} \frac{\lambda}{n} \sum_{i=1}^n \sigma_i \phi(f, Z_i, Z_i') \right) \leq \exp\left( \frac{\lambda^2}{2n} \right).
$$

The proof of this theorem is provided in Appendix C.3. Using this, we obtain

$$
R(\mathcal{F}, S_n, S_n') \leq \frac{\log |\mathcal{F}|}{\lambda} + \frac{\lambda}{2n}.
$$

Optimizing over $\lambda$, we find

$$
R(\mathcal{F}, S_n, S_n') \leq \sqrt{\frac{2 \log |\mathcal{F}|}{n}}.
$$

This is a modified version of Massart's lemma since the left-hand side does not represent the classical Rademacher complexity.

Since $\mathcal{F}$ can be uncountable, we apply a discretization argument using the covering number. By the definition of the supremum, for any $\epsilon' \in \mathbb{R}^+$, there exists $f^* \in \mathcal{F}$ such that

$$
\mathbb{E}_\sigma \sup_{f \in \mathcal{F}} \frac{1}{n} \sum_{i=1}^n \sigma_i \phi(f, Z_i, Z_i') = \mathbb{E}_\sigma \frac{1}{n} \sum_{i=1}^n \sigma_i \phi(f, Z_i, Z_i') + \epsilon'.
$$

Here we present a stronger result compared to the theorem presented in the main paper, based on the $L_2(S_n)$ pseudometric, which is defined as:

$$
\|f - f'\|_{L_2(S_n)} := \sqrt{\frac{1}{n} \sum_{i=1}^n |f(X_i) - f'(X_i)|^2}.
$$

By the definition of the $\epsilon$-covering, for a given $f^* \in \mathcal{F}$, there exists $\tilde{f} \in \mathcal{N}(\epsilon, \mathcal{F}, L_2(S_{2n}))$ such that

$$
\|f^* - \tilde{f}\|_{L_2(S_{2n})} \leq \epsilon.
$$

We first derive the generalization bound using the $L_2(S_n)$ pseudometric, and subsequently upper bound it in terms of the $\| \cdot \|_\infty$ norm. By definition, we have

$$
\mathbb{E}_\sigma \max_\omega \min_{\omega'} \frac{1}{n} \sum_{i=1}^n \sigma_i \phi(f^*, Z_i, Z_i') \leq \mathbb{E}_\sigma \max_\omega \min_{\omega'} \frac{1}{n} \sum_{i=1}^n \sigma_i \phi(\tilde{f}, Z_i, Z_i') + 2\epsilon.
$$

To prove this, we fix $\sigma$ and expand the difference:

$$\frac{1}{n}\sum_{i=1}^{n}\sigma_i\left[(y_i-f^*(x_i))\left(\frac{1+\sigma_i}{2}\omega_i+\frac{1-\sigma_i}{2}\omega_i'\right)-(y_i'-f^*(x_i'))\left(\frac{1+\sigma_i}{2}\omega_i'+\frac{1-\sigma_i}{2}\omega_i\right)\right]$$

$$=\frac{1}{n}\sum_{i=1}^{n}\sigma_i\left[(y_i-\tilde{f}(x_i))\left(\frac{1+\sigma_i}{2}\omega_i+\frac{1-\sigma_i}{2}\omega_i'\right)-(y_i'-\tilde{f}(x_i'))\left(\frac{1+\sigma_i}{2}\omega_i'+\frac{1-\sigma_i}{2}\omega_i\right)\right]$$

$$+\frac{1}{n}\sum_{i=1}^{n}\sigma_i\left[(\tilde{f}(x_i)-f^*(x_i))\left(\frac{1+\sigma_i}{2}\omega_i+\frac{1-\sigma_i}{2}\omega_i'\right)-(\tilde{f}(x_i')-f^*(x_i'))\left(\frac{1+\sigma_i}{2}\omega_i'+\frac{1-\sigma_i}{2}\omega_i\right)\right]$$

$$\leq\text{(first term)}+\sqrt{\frac{1}{n}\sum_{i=1}^{n}|\tilde{f}(x_i)-f^*(x_i)|^2}+\sqrt{\frac{1}{n}\sum_{i=1}^{n}|\tilde{f}(x_i')-f^*(x_i')|^2}$$

$$\leq\text{(first term)}+2\epsilon,$$

where we used the Cauchy–Schwarz inequality and the bounds $|\omega_i|,|\omega_i'|\leq 1$. Taking expectations over $\sigma$ and maximizing/minimizing over $\omega,\omega'$, the result follows.

Thus, we obtain:

$$R(\mathcal{F},S_n,S_n')\leq\sqrt{\frac{2\log N(\epsilon,\mathcal{F},L_2(S_{2n}))}{n}}+2\epsilon+\epsilon'. \tag{13}$$

Letting $\epsilon'\to 0$, we have

$$R(\mathcal{F},S_n,S_n')\leq 2\epsilon+\sqrt{\frac{2\log N(\epsilon,\mathcal{F},L_2(S_n))}{n}}. \tag{14}$$

We can use Eq. (14) for the uniform convergence bound, however, to get the refined dependency, we use the chaining technique (Wainwright, 2019; Zhang, 2023); Let $\epsilon_\ell=2^{-\ell}$ for $\ell=0,1,2,\ldots$. Let $\mathcal{F}_\ell$ be an $\epsilon_\ell$-cover of $\mathcal{F}$ with metric $L_2(S_{2n})$, and define $N_\ell=|\mathcal{F}_\ell|=N(\epsilon_\ell,\mathcal{F},L_2(S_{2n}))$. We may set $\mathcal{F}_0=\{0\}$ at scale $\epsilon_0=1$.

For each $f\in\mathcal{F}$, we consider $f_\ell(f)\in\mathcal{F}_\ell$ so that

$$\|f-f_\ell(f)\|_{L_2(S_{2n})}\leq\epsilon_\ell.$$

Based on the standard chaining technique, which uses $f\in\mathcal{F}$, we consider the following multi-scale decomposition:

$$f=(f-f_L(f))+\sum_{\ell=1}^{L}(f_\ell(f)-g_{\ell-1}(f)).$$

By the triangle inequality, we have

$$\|f_\ell(f)-f_{\ell-1}(f)\|_{L_2(S_n)}\leq\|f_\ell(f)-f\|_{L_2(S_n)}+\|f_{\ell-1}(f)-f\|_{L_2(S_n)}\leq 3\epsilon_\ell.$$

Note that the number of distinct $f_\ell(f)-f_{\ell-1}(f)$ is at most $N_\ell N_{\ell-1}$.

Similar to the derivation of Eqs. (13) and (14), regarding $\phi(f,Z_i,Z_i')$ as $f(Z_i)$, we need to care that $2\epsilon$ appears. Therefore, given $\epsilon$-cover for $\mathcal{F}$,

$$\|f-f_\ell(f)\|_{L_2(S_{2n})}\leq 2\epsilon_\ell.$$

and

$$\|f_\ell(f)-f_{\ell-1}(f)\|_{L_2(S_n)}\leq\|f_\ell(f)-f\|_{L_2(S_n)}+\|f_{\ell-1}(f)-f\|_{L_2(S_n)}\leq(2+4)\epsilon_\ell.$$

holds if regarding $\phi(f,Z_i,Z_i')$ as $f(Z_i)$ in the above.

Then we can consider the following decomposition;

$$R(\mathcal{F}, S_n, S_n') = \mathbb{E}_\sigma \sup_{f \in \mathcal{F}} \frac{1}{n} \sum_{i=1}^n \sigma_i \left[ (f - f_L(f))(Z_i) + \sum_{\ell=1}^L (f_\ell(f) - f_{\ell-1}(f))(Z_i) \right]$$

$$\leq \mathbb{E}_\sigma \sup_{f \in \mathcal{F}} \frac{1}{n} \sum_{i=1}^n \sigma_i (f - f_L(f))(Z_i) + \sum_{\ell=1}^L \mathbb{E}_\sigma \sup_{f \in \mathcal{F}} \frac{1}{n} \sum_{i=1}^n \sigma_i (f_\ell(f) - f_{\ell-1}(f))(Z_i)$$

$$\leq 2\epsilon_L + \sum_{\ell=1}^L \sup_{f \in \mathcal{F}} \| f_\ell(f) - f_{\ell-1}(f) \|_{L_2(S_n)} \sqrt{\frac{2 \ln(N_\ell N_{\ell-1})}{n}}$$

$$\leq 2\epsilon_L + 6 \sum_{\ell=1}^L \epsilon_\ell \sqrt{\frac{2 \ln(N_\ell N_{\ell-1})}{n}}$$

$$\leq 2\epsilon_L + 24 \sum_{\ell=1}^L (\epsilon_\ell - \epsilon_{\ell+1}) \sqrt{\frac{\ln N_\ell}{n}}$$

$$\leq 2\epsilon_L + 24 \int_{\epsilon_{L+1}}^{\epsilon_0} \frac{\sqrt{\ln N(\epsilon', \mathcal{F}, L_2(S_n))}}{\sqrt{n}} d\epsilon'.$$

Then for any $\epsilon > 0$, we pick $L = \sup\{j : \epsilon_j > 2\epsilon\}$. This simple $\epsilon_{L+1} \leq 2\epsilon$, thus $\epsilon_L \leq 4\epsilon$ holds. By definition of $L$, $\epsilon_L > 2\epsilon$ and this implies $\epsilon_{L+1} > \epsilon$

Therefore, we have

$$R(\mathcal{F}, S_n, S_n') \leq \inf_{\epsilon \geq 0} \left[ 8\epsilon + 24 \int_\epsilon^1 \frac{\sqrt{\ln N(\epsilon', \mathcal{F}, L_2(S_n))}}{\sqrt{n}} d\epsilon' \right].$$

By definition, we only need to take $\epsilon \leq 1$.

Finally using the fact that $\ln N(\epsilon', \mathcal{F}, L_2(S_n)) \leq \ln N(\epsilon', \mathcal{F}, \| \cdot \|_\infty)$ (Wainwright, 2019), we obtain the result.

$\square$

Finally, we remark on the case of $\text{smCE}^\sigma$. The dual smooth CE under the dataset $S$ is equivalent to the optimal value of the following optimization problem:

$$\text{smCE}^\sigma(g, S) = \max_{\{\omega_i\}} \frac{1}{n} \sum_{i=1}^n (y_i - v_i)\omega_i$$

subject to the constraints:

$$-1 \leq \omega_i \leq 1, \quad \forall i, \quad |\omega_i - \omega_j| \leq \frac{1}{4} |g(X_i) - g(X_j)|, \quad \forall i, j,$$

where $v_i = \sigma(g(X_i))$, and $\sigma$ denotes the sigmoid function. These constraints can also be rewritten as:

$$-1 \leq \omega_i \leq 1, \quad \forall i, \quad |\omega_i - \omega_j| \leq \frac{1}{4} |\sigma^{-1}(v_i) - \sigma^{-1}(v_j)|, \quad \forall i, j.$$

By definition, the only difference from the standard smooth CE formulation lies in the constraint of the linear program. Since the resulting problem remains a convex optimization, the same proof techniques developed above can be applied to the dual formulation as well. To carry out a similar analysis, it is necessary to bound the range of the logit function $g$.

## C.2 PROOF OF LEMMA 1

*Proof.* Since $\phi(f, S) = \frac{1}{n} \sum_{i=1}^n \phi(f, z_i) = \frac{1}{n} \sum_{i=1}^n (y_i - f(x_i))\omega_i^*$, where $S = \{z_i\}_{i=1}^n$ consists of i.i.d. samples and $|(y_i - f(x_i))\omega_i^*| \leq 1$, one may wish to apply Hoeffding's inequality. However,

the coefficients $\omega_i^*$ are obtained as the solution to a linear program that depends on the dataset $S_{\text{te}}$. Consequently, the terms $(y_i - f(x_i))\omega_i^*$ are not independent, and Hoeffding's inequality cannot be applied.

Instead, we employ McDiarmid's inequality, which requires only the bounded difference property of $\phi(f, S_{\text{te}})$. For completeness, we state McDiarmid's inequality below:

**Lemma 3.** *(Boucheron et al., 2013) We say that a function $f : \mathcal{X} \to \mathbb{R}$ has the bounded difference property if for some nonnegative constants $c_1, \ldots, c_n$,*

$$\sup_{x_1,\ldots,x_n,x_i' \in \mathcal{X}} |f(x_1,\ldots,x_n) - f(x_1,\ldots,x_{i-1},x_i',x_{i+1},\ldots,x_n)| \le c_i, \quad 1 \le i \le n. \quad (15)$$

*If $X_1, \ldots, X_n$ are independent random variables taking values in $\mathcal{X}$ and $f$ has the bounded difference property with constants $c_1, \ldots, c_n$, then for any $t \in \mathbb{R}$, we have*

$$\mathbb{E}\left[e^{t(f(X_1,\ldots,X_n)-\mathbb{E}[f(X_1,\ldots,X_n)])}\right] \le e^{\frac{t^2}{8}\sum_{i=1}^n c_i^2}.$$

*Moreover*

$$\Pr\left(f(X_1,\ldots,X_n) - \mathbb{E}[f(X_1,\ldots,X_n)] \ge \epsilon\right) \le e^{\frac{-2\epsilon^2}{\sum_{i=1}^n c_i^2}} \quad (16)$$

$$\Pr\left(f(X_1,\ldots,X_n) - \mathbb{E}[f(X_1,\ldots,X_n)] \le -\epsilon\right) \le e^{\frac{-2\epsilon^2}{\sum_{i=1}^n c_i^2}}$$

Therefore, we are required to estimate the constants $c_i$ in Eq. (15). To this end, consider replacing the $i$-th data point $z_i$ with $z_i'$, and let the resulting dataset be denoted by $S_n' = (z_1, \ldots, z_{i-1}, z_i', z_{i+1}, \ldots, z_n)$. We define the following notation:

$$\text{smCE}(f, S_n) = \frac{1}{n}\sum_{j=1}^n (y_j - f(x_j))\omega_j^*,$$

where $\omega_j^*$ is the solution of Eq. (8) given $f$ and $S_n$. Similarly, define

$$\text{smCE}(f, S_n') = \frac{1}{n}\sum_{j \ne i}^n (y_j - f(x_j))\omega_j^{'*} + \frac{1}{n}(y_i' - f(x_i'))\omega_i^{'*},$$

where $\omega_j^{'*}$ is the solution of Eq. (8) given $f$ and $S_n'$.

We now evaluate the change in the smooth CE under this replacement:

$$\text{smCE}(f, S_n) - \text{smCE}(f, S_n') = \frac{1}{n}\sum_{j=1}^n (y_j - f(x_j))\omega_j^* - \left[\frac{1}{n}\sum_{j \ne i}^n (y_j - f(x_j))\omega_j^{'*} + \frac{1}{n}(y_i' - f(x_i'))\omega_i^{'*}\right]$$

$$\le \frac{1}{n}\sum_{j=1}^n (y_j - f(x_j))\omega_j^* - \left[\frac{1}{n}\sum_{j \ne i}^n (y_j - f(x_j))\omega_j^* + \frac{1}{n}(y_i' - f(x_i'))\omega_i^*\right]$$

$$= \frac{1}{n}(y_i - f(x_i))\omega_i^* - \frac{1}{n}(y_i' - f(x_i'))\omega_i^*$$

$$= \frac{1}{n}(y_i - y_i')\omega_i^* + \frac{1}{n}(f(x_i') - f(x_i))\omega_i^* \le \frac{2}{n},$$

where the first inequality uses the definition of $\omega_j^*$. By the optimality condition, $smCE(f, S_n') = \frac{1}{n}\sum_{j \ne i}(y_j - f(x_j))\omega_j^{'*} + \frac{1}{n}(y_i' - f(x_i'))\omega_i^{'*} \ge \frac{1}{n}\sum_{j \ne i}(y_j - f(x_j))\omega_j^* + \frac{1}{n}(y_i' - f(x_i'))\omega_i^*$ holds because $\{\omega_j^*\}$ is just another feasible choice for the optimization, and the optimal value must be at least as large as the value attained at any feasible point. The second inequality follows from the fact that $|\omega_i^*| \le 1$.

Similarly, we have:

$$\text{smCE}(f, S_n) - \text{smCE}(f, S_n') = \frac{1}{n}\sum_{j=1}^{n}(y_j - f(x_j))\omega_j^* - \left[\frac{1}{n}\sum_{j\neq i}^{n}(y_j - f(x_j))\omega_j'^* + \frac{1}{n}(y_i' - f(x_i'))\omega_i'^*\right]$$

$$\geq \frac{1}{n}\sum_{j=1}^{n}(y_j - f(x_j))\omega_j'^* - \left[\frac{1}{n}\sum_{j\neq i}^{n}(y_j - f(x_j))\omega_j'^* + \frac{1}{n}(y_i' - f(x_i'))\omega_i'^*\right]$$

$$= \frac{1}{n}(y_i - f(x_i))\omega_i'^* - \frac{1}{n}(y_i' - f(x_i'))\omega_i'^*$$

$$= \frac{1}{n}(y_i' - y_i)\omega_i^* + \frac{1}{n}(f(x_i') - f(x_i))\omega_i'^* \geq -\frac{2}{n}.$$

The first inequality follows from $smCE(f, S_n) = \max_{\{\omega_i\}} \frac{1}{n}\sum_{i=1}^{n}(y_i - v_i)\omega_i = \frac{1}{n}\sum_{j=1}^{n}(y_j - f(x_j))\omega_j^* \geq \frac{1}{n}\sum_{j=1}^{n}(y_j - f(x_j))\omega_j'^*$. This inequality comes from using a value different from the optimal solution $\omega_j^*$.

Combining the two results, we obtain:

$$|\text{smCE}(f, S_n) - \text{smCE}(f, S_n')| \leq \frac{2}{n}.$$

Thus, the bounded difference constant $c_i$ in McDiarmid's inequality satisfies:

$$\sup_{\{z_j\}_{j=1}^{n}, \tilde{z}_i'} |\text{smCE}(f, \mathcal{D}, S_n) - \text{smCE}(f, \mathcal{D}, S_n')| \leq \frac{1}{n}. \tag{17}$$

Our goal is now to study the uniform stability of the quantity $\sup_{f\in\mathcal{F}} |\phi(f, S_{\text{te}}) - \phi(f, S_{\text{tr}})|$. By definition, for any $\epsilon \in \mathbb{R}^+$, there exists $g \in \mathcal{F}$ such that

$$\sup_{f\in\mathcal{F}} |\phi(f, S_{\text{te}}) - \phi(f, S_{\text{tr}})| \leq |\phi(g, S_{\text{te}}) - \phi(g, S_{\text{tr}})| + \epsilon.$$

Therefore, it suffices to study the stability coefficient $c_i$ for $|\phi(g, S_{\text{te}}) - \phi(g, S_{\text{tr}})|$. Consider the combined dataset $S = S_{\text{te}} \cup S_{\text{tr}} \sim \mathcal{D}^{2n}$, and analyze the effect of replacing a single data point in $S$, which consists of $2n$ i.i.d. samples.

We first consider the case where the replaced data point is from the test set $S_{\text{te}} = \{\tilde{z}_i\}_{i=1}^{n}$. Let the perturbed dataset be

$$S_{\text{te}}' = (\tilde{z}_1, \ldots, \tilde{z}_{i-1}, z_i', \tilde{z}_{i+1}, \ldots, \tilde{z}_n).$$

Then, using the triangle inequality and Eq. (17), we have:

$$|\phi(g, S_{\text{te}}) - \phi(g, S_{\text{tr}})| - |\phi(g, S_{\text{te}}') - \phi(g, S_{\text{tr}})| \leq |\phi(g, S_{\text{te}}) - \phi(g, S_{\text{te}}')| \leq \frac{2}{n},$$

$$|\phi(g, S_{\text{te}}) - \phi(g, S_{\text{tr}})| - |\phi(g, S_{\text{te}}') - \phi(g, S_{\text{tr}})| \geq -|\phi(g, S_{\text{te}}) - \phi(g, S_{\text{te}}')| \geq -\frac{2}{n}.$$

Combining these, we obtain:

$$||\phi(g, S_{\text{te}}) - \phi(g, S_{\text{tr}})| - |\phi(g, S_{\text{te}}') - \phi(g, S_{\text{tr}})|| \leq \frac{2}{n}.$$

A similar analysis applies when the replaced data point is from the training set $S_{\text{tr}} = \{z_i\}_{i=1}^{n}$. Let the perturbed test set be

$$S_{\text{tr}}' = (z_1, \ldots, z_{i-1}, z_i', z_{i+1}, \ldots, z_n).$$

Then,

$$||\phi(g, S_{\text{te}}) - \phi(g, S_{\text{tr}})| - |\phi(g, S_{\text{te}}) - \phi(g, S_{\text{tr}}')|| \leq \frac{2}{n}.$$

Therefore, the bounded difference coefficient for each data point in the combined dataset is

$$c_i = \sup_{\{z_j\}_{j=1}^{2n}, z_i'} |\phi(g, S_{\text{te}}) - \phi(g, S_{\text{tr}}) - (\phi(g, S_{\text{te}}') - \phi(g, S_{\text{tr}}))| \leq \frac{2}{n},$$

for all $i = 1, \ldots, 2n$.

Applying McDiarmid's inequality (Eq. (16)) with these coefficients yields:

$$\Pr\left(\sup_{f \in \mathcal{F}} |\phi(f, S_{\text{te}}) - \phi(f, S_{\text{tr}})| - \mathbb{E}_{S_{\text{te}}, S_{\text{tr}} \sim \mathcal{D}^{2n}} \sup_{f \in \mathcal{F}} |\phi(f, S_{\text{te}}) - \phi(f, S_{\text{tr}})| \geq \epsilon\right) \leq \exp(-n\epsilon^2/4).$$

$\square$

## C.3 Proof of Lemma 2

Similar to the proof of Lemma 1 in Appendix C.2, we apply McDiarmid's inequality to evaluate the exponential moment. To this end, we first compute the bounded difference constants in Eq. (15). Recall the definition:

$$\max_{\omega} \min_{\omega'} \frac{\lambda}{n} \sum_{i=1}^{n} \sigma_i \phi(f, Z_i, Z_i'),$$

where

$$\phi(f, Z_i, Z_i') \coloneqq \left[(y_i - f(x_i))\left(\tfrac{1+\sigma_i}{2}\omega_i + \tfrac{1-\sigma_i}{2}\omega_i'\right) - (y_i' - f(x_i'))\left(\tfrac{1+\sigma_i}{2}\omega_i' + \tfrac{1-\sigma_i}{2}\omega_i\right)\right].$$

For clarity, we first analyze the special case where $\sigma_i = 1$ for all $i \in [n]$; the general case is handled later. Under this assumption, the expression reduces to the difference of smooth CEs:

$$\max_{\omega} \min_{\omega'} \frac{\lambda}{n} \sum_{i=1}^{n} \phi(f, Z_i, Z_i') = \lambda\big(\text{smCE}(f, S_n) - \text{smCE}(f, S_n')\big). \tag{18}$$

Now fix the dataset and flip a single coordinate $\sigma_j$ from 1 to $-1$. Only the $j$-th term is affected. For $\sigma_j = 1$, the term is

$$(y_j - f(x_j))\omega_j - (y_j' - f(x_j'))\omega_j',$$

while for $\sigma_j = -1$, it becomes

$$(y_j' - f(x_j'))\omega_j - (y_j - f(x_j))\omega_j'.$$

This corresponds to swapping the training and test inputs in the smooth CE optimization.

Formally, define the datasets:

$$\tilde{S}_n = ((x_1, y_1), \ldots, (x_j', y_j'), \ldots, (x_n, y_n)), \quad \tilde{S}_n' = ((x_1', y_1'), \ldots, (x_j, y_j), \ldots, (x_n', y_n')).$$

Then the exponent, with $\sigma_i = 1$ for $i \neq j$ and $\sigma_j = -1$, can be written as

$$\max_{\omega} \min_{\omega'} \frac{\lambda}{n} \sum_{i \neq j} \phi(f, Z_i, Z_i') - \frac{\lambda}{n}\phi(f, Z_j, Z_j') = \lambda\big(\text{smCE}(f, \tilde{S}_n) - \text{smCE}(f, \tilde{S}_n')\big).$$

Using Eq. (17), which bounds the stability of smCE, and combining with Eq. (18), we obtain:

$$\left| \max_{\omega} \min_{\omega'} \frac{\lambda}{n} \sum_{i=1}^{n} \phi(f, Z_i, Z_i') - \left( \max_{\omega} \min_{\omega'} \frac{\lambda}{n} \sum_{i \neq j} \phi(f, Z_i, Z_i') - \frac{\lambda}{n}\phi(f, Z_j, Z_j') \right) \right|$$

$$= \lambda\big|\text{smCE}(f, S_n) - \text{smCE}(f, \tilde{S}_n)\big| + \lambda\big|\text{smCE}(f, S_n') - \text{smCE}(f, \tilde{S}_n')\big|$$

$$\leq \frac{2\lambda}{n}.$$

Since this bound holds for arbitrary $j \in [n]$, we have:

$$\sup_{j \in [n]} \left| \max_{\omega} \min_{\omega'} \frac{\lambda}{n} \sum_{i=1}^{n} \phi(f, Z_i, Z_i') - \left( \max_{\omega} \min_{\omega'} \frac{\lambda}{n} \sum_{i \neq j} \phi(f, Z_i, Z_i') - \frac{\lambda}{n} \phi(f, Z_j, Z_j') \right) \right| \leq \frac{2\lambda}{n}. \quad (19)$$

The same reasoning applies to any fixed realization $\sigma = \sigma|_{\pm}$ of the Rademacher variables (note that $\sigma|_{\pm}$ is a fixed realization and not a random variable). For each such realization, we construct the datasets $S_{\sigma|_{\pm}}$ and $S'_{\sigma|_{\pm}}$ used in the smooth CE optimization. Then,

$$\max_{\omega} \min_{\omega'} \frac{\lambda}{n} \sum_{i=1}^{n} \sigma|_{\pm} \phi(f, Z_i, Z_i') = \lambda \left( \mathrm{smCE}(f, S_{\sigma|_{\pm}}) - \mathrm{smCE}(f, S'_{\sigma|_{\pm}}) \right),$$

and changing any single component of $\sigma$ alters the value by at most $\frac{2\lambda}{n}$, as in Eq. (19).

Therefore, by the argument in Appendix C.2, the sensitivity coefficients satisfy $c_i = \frac{2\lambda}{n}$ for all $i \in [n]$. Substituting this into Lemma 4 in Appendix C.2 yields the desired exponential moment bound.

### C.4 PROOF OF COROLLARY 1

*Proof.* As stated in the main paper, Błasiok et al. (2023) proved in Theorem 9.5 that, with probability at least $1 - \delta$ over the draw of the test dataset,

$$|\mathrm{smCE}(f, \mathcal{D}) - \mathrm{smCE}(f, S_{\mathrm{te}})| \leq 2\mathfrak{R}_{\mathcal{D}, n}(\mathrm{Lip}_1([0, 1], [-1, 1])) + \sqrt{\frac{\log \frac{2}{\delta}}{2n}}. \quad (20)$$

They further showed that there exists a universal constant $C$ such that

$$\mathfrak{R}_{\mathcal{D}, n}(\mathrm{Lip}_1([0, 1], [-1, 1])) \leq \frac{C}{\sqrt{n}},$$

based on the result of Ambroladze & Shawe-Taylor (2004). See also Luxburg & Bousquet (2004) for the derivation of Rademacher complexity of lipshictz functions.

Combining this with Theorem 1 and Corollary 6, and applying the triangle inequality, we obtain:

$$\sup_{f} |\mathrm{smCE}(f, \mathcal{D}) - \mathrm{smCE}(f, S_{\mathrm{tr}})|$$
$$\leq \sup_{f} |\mathrm{smCE}(f, \mathcal{D}) - \mathrm{smCE}(f, S_{\mathrm{te}})| + \sup_{f} |\mathrm{smCE}(f, S_{\mathrm{te}}) - \mathrm{smCE}(f, S_{\mathrm{tr}})|.$$

We now allocate the total failure probability $\delta$ across the two terms using the union bound. Specifically, we set $\delta \to \frac{2}{3}\delta$ in Eq. (20), and $\delta \to \frac{1}{3}\delta$ in Theorem 1 or Corollary 6. Then, the sum of the confidence terms becomes:

$$\sqrt{\frac{\log \frac{2}{\frac{2}{3}\delta}}{2n}} + 2\sqrt{\frac{\log \frac{1}{\frac{1}{3}\delta}}{n}} = \left( 2 + \frac{1}{\sqrt{2}} \right) \sqrt{\frac{\log \frac{3}{\delta}}{n}} \leq 3\sqrt{\frac{\log \frac{3}{\delta}}{n}}.$$

Thus, we obtain the desired high-probability bound. $\qquad\square$

## D ANALYSIS BASED ON RADEMACHER COMPLEXITY

### D.1 PROOF OF THEOREM 2

To simplify the notation, we define

$$\phi(f, h, x, y) = h(f(x)) \cdot (y - f(x)).$$

Then

$$\text{smCE}(f, \mathcal{D}) - \text{smCE}(f, S_{\text{tr}})$$
$$\leq \sup_{f \in \mathcal{F}} \text{smCE}(f, \mathcal{D}) - \text{smCE}(f, S_{\text{tr}})$$
$$\leq \sup_{f \in \mathcal{F}} \sup_{h \in \text{Lip}_{L=1}([0,1],[-1,1])} \mathbb{E}\phi(f, h, X, Y) - \sup_{h \in \text{Lip}_{L=1}([0,1],[-1,1])} \frac{1}{n}\sum_{i=1}^{n} \phi(f, h, X_i, Y_i)$$
$$\leq \sup_{f \in \mathcal{F}} \sup_{h \in \text{Lip}_{L=1}([0,1],[-1,1])} \left\{ \mathbb{E}\phi(f, h, X, Y) - \frac{1}{n}\sum_{i=1}^{n} \phi(f, h, X_i, Y_i) \right\}$$
$$\leq \sup_{h \in \text{Lip}_{L=1}([0,1],[-1,1])} \sup_{f \in \mathcal{F}} \left\{ \mathbb{E}\phi(f, h, X, Y) - \frac{1}{n}\sum_{i=1}^{n} \phi(f, h, X_i, Y_i) \right\}.$$

In the last line, we simply used the swapping of the supremum, which can be shown using the definition of the supremum. (In general, for any $f \in \mathcal{F}$, $\phi(f, h) \leq \sup_{f'} \phi(f', h)$, which leads to $\sup_h \phi(f, h) \leq \sup_h \sup_{f'} \phi(f', h)$. This further leads to $\sup_f \sup_h \phi(f, h) \leq \sup_h \sup'_f \phi(f', h)$.)

Then, we use Mcdiramid's inequality. The bounded difference for the training dataset is $2/n$, we have the following result; for any $\delta > 0$ with probability $1 - \delta/2$, we have

$$\sup_{h \in \text{Lip}_{L=1}([0,1],[-1,1])} \sup_{f \in \mathcal{F}} \mathbb{E}\phi(f, h, X, Y) - \frac{1}{n}\sum_{i=1}^{n} \phi(f, h, X_i, Y_i)$$

$$\leq \mathbb{E}_{S_{\text{tr}} \sim D^n} \sup_{h \in \text{Lip}_{L=1}([0,1],[-1,1])} \sup_{f \in \mathcal{F}} \mathbb{E}\phi(f, h, X, Y) - \frac{1}{n}\sum_{i=1}^{n} \phi(f, h, X_i, Y_i) + 2\sqrt{\frac{\log\frac{2}{\delta}}{n}}.$$

Then by considering the standard symmetrization property, we have the following upper bound

$$\sup_{h \in \text{Lip}_{L=1}([0,1],[-1,1])} \sup_{f \in \mathcal{F}} \mathbb{E}\phi(f, h, X, Y) - \frac{1}{n}\sum_{i=1}^{n} \phi(f, h, X_i, Y_i)$$

$$\leq 2\mathbb{E}_S\mathbb{E}_\sigma\left[ \sup_{h \in \text{Lip}_{L=1}([0,1],[-1,1])} \sup_{f \in \mathcal{F}} \frac{1}{n}\sum_{i=1}^{n} \sigma_i\phi(f, h, X_i, Y_i) \right] + 2\sqrt{\frac{\log\frac{2}{\delta}}{n}}.$$

Finally, by considering the union bound, for any $\delta > 0$ with probability $1 - \delta$, we have

$$\sup_{h \in \text{Lip}_{L=1}([0,1],[-1,1])} \sup_{f \in \mathcal{F}} \left| \mathbb{E}\phi(f, h, X, Y) - \frac{1}{n}\sum_{i=1}^{n} \phi(f, h, X_i, Y_i) \right|$$

$$\leq 2\mathbb{E}_S\mathbb{E}_\sigma\left[ \sup_{h \in \text{Lip}_{L=1}([0,1],[-1,1])} \sup_{f \in \mathcal{F}} \frac{1}{n}\sum_{i=1}^{n} \sigma_i\phi(f, h, X_i, Y_i) \right] + 2\sqrt{\frac{\log\frac{2}{\delta}}{n}}.$$

Next, we evaluate the Rademacher complexity term.

For the notational convenience, we define

$$T := \mathbb{E}_S\mathbb{E}_\sigma\left[ \sup_{h \in \text{Lip}_{L=1}([0,1],[-1,1])} \sup_{f \in \mathcal{F}} \frac{1}{n}\sum_{i=1}^{n} \sigma_i\phi(f, h, X_i, Y_i) \right],$$
$$\eta_0 := \{\eta : [0,1] \to \mathbb{R} \mid -1 \leq \text{Lip}(\eta) \leq 1\}.$$

Our goal is to show

$$T \leq 2\mathfrak{R}_{\mathcal{D},n}(\mathcal{F}) + \frac{12}{\sqrt{n}}\int_0^1 \sqrt{\log N(u, \eta_0, \|\cdot\|_\infty)}\,du.$$

First, we will consider an $\epsilon$-net of the Lipschitz class, which is independent of data and $\sigma$. Fix geometric scales $\varepsilon_k := 2^{-k}$ for $k = 0, 1, 2, \ldots$. For each $k$, choose a *fixed* (data- and $\sigma$-independent) $\varepsilon_k$-net $\mathcal{N}_k \subset \eta_0$ under the uniform norm $\|\cdot\|_\infty$, with $|\mathcal{N}_k| = N(\varepsilon_k, \eta_0, \|\cdot\|_\infty)$. For every $\eta \in \eta_0$ pick a nearest projection $\pi_k(\eta) \in \mathcal{N}_k$ so that $\|\eta - \pi_k(\eta)\|_\infty \leq \varepsilon_k$. Then, by the triangle inequality,

$$\left\|\pi_k(\eta) - \pi_{k-1}(\eta)\right\|_\infty \leq \varepsilon_k + \varepsilon_{k-1} \leq 2\varepsilon_{k-1}.$$

Consequently,

$$\eta = \pi_0(\eta) + \sum_{k \geq 1} \big(\pi_k(\eta) - \pi_{k-1}(\eta)\big).$$

We set $A(h) = \sup_{f \in \mathcal{F}} \frac{1}{n} \sum_{i=1}^n \sigma_i \phi(f, h, X_i, Y_i)$. The discretization error by the $k$-th covering is given as

$$\mathbb{E}_S \mathbb{E}_\sigma \sup_{h \in \mathrm{Lip}} A(h) - \mathbb{E}_S \mathbb{E}_\sigma \sup_{h \in \mathcal{N}_k} A(h)$$

$$= \mathbb{E}_S \mathbb{E}_\sigma \sup_{h \in \mathrm{Lip}} \sup_{f \in \mathcal{F}} \frac{1}{n} \sum_{i=1}^n \sigma_i \phi(f, h, X_i, Y_i) - \mathbb{E}_S \mathbb{E}_\sigma \sup_{h \in \mathcal{N}_k} \sup_{f \in \mathcal{F}} \frac{1}{n} \sum_{i=1}^n \sigma_i \phi(f, h, X_i, Y_i)$$

$$\leq \mathbb{E}_S \mathbb{E}_\sigma \sup_{h \in \mathrm{Lip}} \frac{1}{n} \sum_{i=1}^n \sigma_i \phi(f', h, X_i, Y_i) - \sup_{h \in \mathcal{N}_k} \frac{1}{n} \sum_{i=1}^n \sigma_i \phi(f', h, X_i, Y_i) + 2\epsilon'$$

$$\leq \epsilon_k + 2\epsilon',$$

where $f'$ and $\epsilon'$ is coming from the definition of the supremum, that is, there exists some $f' \in \mathcal{F}$ such that $\sup_{f \in \mathcal{F}} \frac{1}{n} \sum_{i=1}^n \sigma_i \phi(f, h, X_i, Y_i) \leq \frac{1}{n} \sum_{i=1}^n \sigma_i \phi(f', h, X_i, Y_i) + \epsilon'$ for any $\epsilon' \in \mathbb{R}_+$. Since we can take $\epsilon'$ arbitrarily small, the discretization error is $\epsilon_k$.

Next we show that under the $k$-th covering, we can disentangle the complexity of the lipshicit function class by the covering number as follows:

$$\mathbb{E}_S \mathbb{E}_\sigma \sup_{h \in \mathcal{N}_k} A(h) - \sup_{h \in \mathcal{N}_k} \mathbb{E}_S \mathbb{E}_\sigma \left[ \sup_{f \in \mathcal{F}} \frac{1}{n} \sum_{i=1}^n \sigma_i \phi(f, h, X_i, Y_i) \right]$$

$$= \frac{1}{t} \log \exp t(\mathbb{E}_S \mathbb{E}_\sigma \sup_{h \in \mathcal{N}_k} A(h) - \sup_{h \in \mathcal{N}_k} \mathbb{E}_S \mathbb{E}_\sigma \left[ \sup_{f \in \mathcal{F}} \frac{1}{n} \sum_{i=1}^n \sigma_i \phi(f, h, X_i, Y_i) \right])$$

$$\leq \frac{1}{t} \log \mathbb{E}_S \mathbb{E}_\sigma \exp t(\sup_{h \in \mathcal{N}_k} A(h) - \sup_{h \in \mathcal{N}_k} \mathbb{E}_S \mathbb{E}_\sigma \left[ \sup_{f \in \mathcal{F}} \frac{1}{n} \sum_{i=1}^n \sigma_i \phi(f, h, X_i, Y_i) \right])$$

$$\leq \frac{1}{t} \log \mathbb{E}_S \mathbb{E}_\sigma \exp t(\sup_{h \in \mathcal{N}_k} A(h) - \mathbb{E}_S \mathbb{E}_\sigma \left[ \sup_{f \in \mathcal{F}} \frac{1}{n} \sum_{i=1}^n \sigma_i \phi(f, h, X_i, Y_i) \right])$$

$$\leq \frac{1}{t} \log \mathbb{E}_S \mathbb{E}_\sigma \sup_{h \in \mathcal{N}_k} \exp t(A(h) - \mathbb{E}_S \mathbb{E}_\sigma \left[ \sup_{f \in \mathcal{F}} \frac{1}{n} \sum_{i=1}^n \sigma_i \phi(f, h, X_i, Y_i) \right])$$

$$\leq \frac{1}{t} \log \mathbb{E}_S \mathbb{E}_\sigma \sum_{h \in \mathcal{N}_k} \exp t(A(h) - \mathbb{E}_S \mathbb{E}_\sigma \left[ \sup_{f \in \mathcal{F}} \frac{1}{n} \sum_{i=1}^n \sigma_i \phi(f, h, X_i, Y_i) \right]).$$

To evaluate this exponential moment, we use the Mcdiramid's inequality. The bounded difference coefficient is

$$\sup_{f \in \mathcal{F}} \frac{1}{n} \sum_{i=1}^n \sigma_i \phi(f, h, X_i, Y_i) - \left( \sup_{f' \in \mathcal{F}} \frac{1}{n} \sum_{i \neq j} \sigma_i \phi(f', h, X_i, Y_i) + \frac{1}{n} \sigma_j' \phi(f', h, X_j, Y_j) \right)$$

$$\leq \sup_{f \in \mathcal{F}} \frac{1}{n} (\sigma_j \phi(f', h, X_j, Y_j) - \sigma_j' \phi(f', h, X_j, Y_j)) \leq \frac{2}{n}.$$

Thus

$$\mathbb{E}_S\mathbb{E}_\sigma \sup_{h\in\mathcal{N}_k} A(h) - \sup_{h\in\mathcal{N}_k} \mathbb{E}_S\mathbb{E}_\sigma \left[\sup_{f\in\mathcal{F}} \frac{1}{n} \sum_{i=1}^n \sigma_i \phi(f,h,X_i,Y_i)\right] \le \frac{1}{t}\log \sum_{h\in\mathcal{N}_k} \exp t^2 n(\frac{2}{n})/8$$

$$= \frac{1}{t}\log|\mathcal{N}_k| + \frac{t}{2n},$$

Next, we analyze $\sup_{h\in\mathcal{N}_k} \mathbb{E}_S\mathbb{E}_\sigma \left[\sup_{f\in\mathcal{F}} \frac{1}{n}\sum_{i=1}^n \sigma_i\phi(f,h,X_i,Y_i)\right]$. $\mathbb{E}_S\mathbb{E}_\sigma \left[\sup_{f\in\mathcal{F}} \frac{1}{n}\sum_{i=1}^n \sigma_i\phi(f,h,X_i,Y_i)\right]$ is the Rademacher complexity of $\phi(f,h,X,Y)$. Moreover from Lemma 7.4 in Błasiok et al. (2023), $\phi(f,h,X,Y)$ is 2-Lipschitz with respect to $h$. Thus by Talagrand's contraction lemma, $\sup_{h\in\mathcal{N}_k} \mathbb{E}_S\mathbb{E}_\sigma \left[\sup_{f\in\mathcal{F}} \frac{1}{n}\sum_{i=1}^n \sigma_i\phi(f,h,X_i,Y_i)\right] \le 2\mathfrak{R}_{\mathcal{D},n}(\mathcal{F})$.

Thus, in conclusion, we have

$$\mathbb{E}_S\mathbb{E}_\sigma \left[\sup_{h\in\mathcal{N}_k}\sup_{f\in\mathcal{F}} \frac{1}{n}\sum_{i=1}^n \sigma_i\phi(f,h,X_i,Y_i)\right] \le 2\mathfrak{R}_{\mathcal{D},n}(\mathcal{F}) + \frac{1}{t}\log|\mathcal{N}_k| + \frac{t}{2n}.$$

Thus, combining the discretization result

$$\mathbb{E}_S\mathbb{E}_\sigma \left[\sup_{h\in\mathrm{Lip}_{L=1}([0,1],[-1,1])}\sup_{f\in\mathcal{F}} \frac{1}{n}\sum_{i=1}^n \sigma_i\phi(f,h,X_i,Y_i)\right] \le 2\mathfrak{R}_{\mathcal{D},n}(\mathcal{F}) + \sqrt{\frac{2\log|\mathcal{N}_k|}{n}} + \epsilon_k.$$

Then, following Dudley's integral approach as shown in Eq. (14), we have

$$\mathbb{E}_S\mathbb{E}_\sigma \left[\sup_{h\in\mathrm{Lip}_{L=1}([0,1],[-1,1])}\sup_{f\in\mathcal{F}} \frac{1}{n}\sum_{i=1}^n \sigma_i\phi(f,h,X_i,Y_i)\right]$$

$$\le 2\mathfrak{R}_{\mathcal{D},n}(\mathcal{F}) + \inf_{\epsilon\ge 0}\left[4\epsilon + 12\int_\epsilon^1 \frac{\sqrt{\ln N(\epsilon',\eta_0,\|\cdot\|_\infty)}}{\sqrt{n}}d\epsilon'\right]$$

$$\le 2\mathfrak{R}_{\mathcal{D},n}(\mathcal{F}) + \frac{C}{\sqrt{n}}, \tag{21}$$

where we used the covering number estimate of the 1-dimensional Lipschitz functions, see Wainwright (2019) for the details.

## D.2 SUB-OPTIMAL RESULT OF THE RADEMACHER COMPLEXITY

The evaluation of $|\mathrm{smCE}(f,S_{\mathrm{te}}) - \mathrm{smCE}(f,S_{\mathrm{tr}})|$ using Rademacher complexity can be carried out in the same manner as in Appendix D.1, except that randomness is now considered over the $2n$ data points from both the test and train sets. Apart from this difference, the argument is identical, and the resulting bound again depends—just as in Appendix D.1—on the complexity of Lipschitz functions through covering arguments, as well as on the complexity of $\mathcal{F}$. This reflects the inherent complexity of the composite function $h(f(X))$ appearing in the definition of smooth CE. Consequently, compared with Theorem 1, where the covering-based proof avoids dependence on the complexity of Lipschitz functions, this approach yields a suboptimal result.

On the other hand, as shown below, it is also possible to derive the Rademacher complexity bound from the covering-number result of Theorem 1. In this case, the dependence on the complexity of Lipschitz functions disappears, but at the cost of introducing an additional $(\log 2n)^2$ factor into the Rademacher complexity bound.

**Corollary 6.** *Under the same assumptions as in Theorem 1, there exist universal constants $\{C_i\}$s such that with probability at least $1-\delta$ over the draw of $S_{\mathrm{te}}$ and $S_{\mathrm{tr}}$, we have:*

$$\sup_{f\in\mathcal{F}} |\mathrm{smCE}(f,S_{\mathrm{te}}) - \mathrm{smCE}(f,S_{\mathrm{tr}})| \le C_1\mathfrak{R}_{\mathcal{D},2n}(\mathcal{F})(\log 2n)^2 + \frac{C_2}{\sqrt{n}} + 2\sqrt{\frac{\log\delta^{-1}}{n}}.$$

*Proof.* We begin by defining the empirical Gaussian complexity of $\mathcal{F}$ as:

$$\mathfrak{G}(\mathcal{F},S_n) = \mathbb{E}_{\mathbf{g}}\sup_{f\in\mathcal{F}} \frac{1}{n}\sum_{i=1}^n g_i f(Z_i),$$

where $\mathbf{g} = [g_1, \ldots, g_n]$ are independent standard normal random variables, i.e., $g_i \sim \mathcal{N}(0, 1)$ for all $i \in [n]$.

From the modified version of Sudakov's minoration inequality (Theorem 12.4 in Zhang (2023)), we have:

$$\sqrt{\ln \mathcal{N}(\epsilon, \mathcal{F}, L_2(S_{2n}))} \leq \sqrt{\ln M(\epsilon, \mathcal{F}, L_2(S_{2n}))} \leq \frac{2\sqrt{2n}\mathfrak{G}_{\mathcal{D}, 2n}(\mathcal{F})}{\epsilon} + 1.$$

This justifies the reason why we use the covering number based on the $L_2(S_{2n})$ pseudometric in the derivation of Theorem 1.

As a result, we obtain:

$$|\text{smCE}(f, S_{\text{te}}) - \text{smCE}(f, S_{\text{tr}})| \leq \inf_{\epsilon > 0} \left[ 8\epsilon + 24 \int_\epsilon^1 \left( \frac{2\sqrt{2}\mathfrak{G}_{\mathcal{D}, 2n}(\mathcal{F})}{\epsilon'} + \frac{1}{\sqrt{n}} \right) d\epsilon' \right] + \sqrt{\frac{\log(1/\delta)}{n}}$$

$$\leq \inf_{\epsilon > 0} \left[ 8\epsilon + 48\sqrt{2}\mathfrak{G}_{\mathcal{D}, 2n}(\mathcal{F}) \log\left( \frac{1}{\epsilon} \right) + \frac{24}{\sqrt{n}} \right] + \sqrt{\frac{\log(1/\delta)}{n}}$$

$$\leq \frac{8}{\sqrt{2n}} + 24\sqrt{2}\mathfrak{G}_{\mathcal{D}, 2n}(\mathcal{F}) \log(2n) + \frac{24}{\sqrt{2n}} + \sqrt{\frac{\log(1/\delta)}{n}},$$

where we have set $\epsilon = 1/\sqrt{2n}$ in the final inequality.

Furthermore, from Lemma 4 in Bartlett & Mendelson (2002), there exist universal constants $c$ and $C$ such that:

$$c\mathfrak{R}_{\mathcal{D}, 2n}(\mathcal{F}) \leq \mathfrak{G}_{\mathcal{D}, 2n}(\mathcal{F}) \leq C \log(2n)\mathfrak{R}_{\mathcal{D}, 2n}(\mathcal{F}).$$

Combining the above results, we conclude:

$$|\text{smCE}(f, S_{\text{te}}) - \text{smCE}(f, S_{\text{tr}})| \leq 24\sqrt{2}C\mathfrak{R}_{\mathcal{D}, 2n}(\mathcal{F})(\log 2n)^2 + \frac{32}{\sqrt{2n}} + 2\sqrt{\frac{\log(1/\delta)}{n}}.$$

$\square$

# E  ESTIMATION OF RADEMACHER COMPLEXITY OF THE LIPSHITZ FUNCTION CLASS AND UNIFORM CONVERGENCE BOUNDS

In the main paper, we evaluated the universal constant to express the complexity of Lipshitz functions following the standard way in the learning theory (Wainwright, 2019). In this section, we derive the explicit constant of the Lipshitz function class. This leads to the explicit upper bound of Eq. (3). Recall that the constant is based on Rademacher complexity of the following function class. We define the class of 1-Lipschitz functions on $[0, 1]$ with range in $[-1, 1]$:

$$\mathcal{F} := \left\{ f : [0, 1] \rightarrow [-1, 1] \ \middle| \ |f(x) - f(y)| \leq |x - y| \ \forall x, y \in [0, 1] \right\}.$$

Then we show that the corresponding complexity is

$$\widehat{\mathfrak{R}}_n(\mathcal{F}) \leq \frac{24\sqrt{5 \log 3}}{\sqrt{n}} + \frac{\sqrt{2 \log 2}}{\sqrt{n}} \leq \frac{58}{\sqrt{n}}.$$

From Eq. (20), we have that with probability at least $1 - \delta$ over the draw of the test dataset,

$$|\text{smCE}(f, \mathcal{D}) - \text{smCE}(f, S_{\text{te}})| \leq \frac{116}{\sqrt{n}} + \sqrt{\frac{\log \frac{2}{\delta}}{2n}}.$$

Then this leads to that the constant $C_1 = 116$ in Corollary 1. As for the constant $C_2$ in Theorem 2, from Eq. (21) and Eq. (22), we have $C_2 = 24\sqrt{5 \log 3} < 57$.

*Proof.* Note that we do not impose any anchoring condition such as $f(0) = 0$.

For any $f \in \mathcal{F}$, define

$$c := f(0) \in [-1, 1], \qquad g(x) := f(x) - f(0).$$

Then the following properties are immediate:

$$g(0) = 0, \qquad |g(x) - g(y)| = |f(x) - f(y)| \leq |x - y| \quad \forall x, y \in [0, 1].$$

Moreover, since $g(0) = 0$ and $g$ is 1-Lipschitz, we have

$$|g(x)| = |g(x) - g(0)| \leq |x - 0| \leq 1 \qquad \forall x \in [0, 1],$$

so $g(x) \in [-1, 1]$ for all $x$. Hence $g$ belongs to the anchored class

$$\mathcal{G} := \Big\{ g : [0, 1] \to [-1, 1] \ \Big| \ g(0) = 0, \ |g(x) - g(y)| \leq |x - y| \ \forall x, y \in [0, 1] \Big\}.$$

In addition, $f$ can be written as $f(x) = g(x) + c$ with $g \in \mathcal{G}$ and $c \in [-1, 1]$. If we define the class of constant functions

$$\mathcal{C} := \{ x \mapsto c \mid c \in [-1, 1] \},$$

we obtain the decomposition

$$\mathcal{F} = \mathcal{G} + \mathcal{C},$$

that is, every $f \in \mathcal{F}$ can be written as the sum of an anchored 1-Lipschitz function and a constant function.

Next, we relate the empirical Rademacher complexity of $\mathcal{F}$ to those of $\mathcal{G}$ and $\mathcal{C}$. Given a fixed sample $(x_1, \ldots, x_n)$ and independent Rademacher variables $(\sigma_1, \ldots, \sigma_n)$, the empirical Rademacher complexity of a function class $\mathcal{H}$ is defined as

$$\widehat{\mathfrak{R}}_n(\mathcal{H}) := \mathbb{E}_\sigma \left[ \sup_{h \in \mathcal{H}} \frac{1}{n} \sum_{i=1}^n \sigma_i h(x_i) \right].$$

Using the decomposition $f = g + c$ with $g \in \mathcal{G}$ and $c \in [-1, 1]$, we obtain

$$\widehat{\mathfrak{R}}_n(\mathcal{F}) = \mathbb{E}_\sigma \left[ \sup_{f \in \mathcal{F}} \frac{1}{n} \sum_{i=1}^n \sigma_i f(x_i) \right]$$

$$= \mathbb{E}_\sigma \left[ \sup_{g \in \mathcal{G}, \, c \in [-1,1]} \frac{1}{n} \sum_{i=1}^n \sigma_i \big( g(x_i) + c \big) \right]$$

$$= \mathbb{E}_\sigma \left[ \sup_{g \in \mathcal{G}, \, c \in [-1,1]} \left\{ \frac{1}{n} \sum_{i=1}^n \sigma_i g(x_i) + c \cdot \frac{1}{n} \sum_{i=1}^n \sigma_i \right\} \right].$$

The supremum over $(g, c)$ can be bounded by splitting the two terms:

$$\sup_{g \in \mathcal{G}, \, c \in [-1,1]} \left\{ \frac{1}{n} \sum_{i=1}^n \sigma_i g(x_i) + c \cdot \frac{1}{n} \sum_{i=1}^n \sigma_i \right\} \leq \sup_{g \in \mathcal{G}} \frac{1}{n} \sum_{i=1}^n \sigma_i g(x_i) + \sup_{c \in [-1,1]} c \cdot \frac{1}{n} \sum_{i=1}^n \sigma_i.$$

For the second term, by definition

$$\mathbb{E}_\sigma \left[ \sup_{c \in [-1,1]} c \cdot \frac{1}{n} \sum_{i=1}^n \sigma_i \right]$$

if $\sum_{i=1}^n \sigma_i < 0$, the maximum is achieved when $c = -1$ and if $\sum_{i=1}^n \sigma_i \geq 0$, the maximum is achieved when $c = 1$. We define $\mathbf{1} = (1, \ldots, 1) \in \mathbb{R}^n$, $A = \{-\mathbf{1}, \mathbf{1}\}$. By using Massart's lemma, we have

$$\mathbb{E}_\sigma \left[ \sup_{c \in [-1,1]} c \cdot \frac{1}{n} \sum_{i=1}^n \sigma_i \right] = \mathbb{E}_\sigma \left[ \sup_{z \in A} \frac{1}{n} \sum_{i=1}^n \sigma_i z_i \right] \leq \frac{\sqrt{2 \log 2}}{\sqrt{n}}.$$

Therefore, we only need to upper bound

$$\widehat{\mathfrak{R}}_n(\mathcal{G}) := \mathbb{E}_\sigma\left[\sup_{g\in\mathcal{G}} \frac{1}{n}\sum_{i=1}^n \sigma_i g(x_i)\right].$$

By using the discretization argument for the one dimensional Lipshitz function class (for example, see Eaxmple 5.10 in Wainwright (2019)), we have

$$\log N(\varepsilon, \mathcal{G}, \|\cdot\|_\infty) \leq \frac{5\log 3}{\varepsilon}.$$

Then the rademacher complexity can be bounded by the Dudley entropy integral (in empirical $L_2(P_n)$), we have

$$\widehat{\mathfrak{R}}_n(\mathcal{G}) \leq \frac{12}{\sqrt{n}}\int_0^1 \sqrt{\log N(u, \mathcal{G}, L_2(P_n))}\, du.$$

Since $\|h_1 - h_2\|_{L_2(P_n)} \leq \|h_1 - h_2\|_\infty$ for any $h_1, h_2$, the covering numbers in $L_2(P_n)$ are bounded by those in $\|\cdot\|_\infty$:

$$\log N(u, \mathcal{G}, L_2(P_n)) \leq \log N(u, \mathcal{G}, \|\cdot\|_\infty) \leq \frac{5\log 3}{u},$$

for all $u \in (0, 1]$, where we used the previous lemma with $\varepsilon = u$. Hence

$$
\begin{aligned}
\widehat{\mathfrak{R}}_n(\mathcal{G}) &\leq \frac{12}{\sqrt{n}}\int_0^1 \sqrt{\frac{5\log 3}{u}}\, du \\
&= \frac{12\sqrt{5\log 3}}{\sqrt{n}}\int_0^1 u^{-1/2}\, du \\
&= \frac{12\sqrt{5\log 3}}{\sqrt{n}}\cdot 2 = \frac{24\sqrt{5\log 3}}{\sqrt{n}}.
\end{aligned}
\tag{22}
$$

$\square$

# F   PROOFS OF THE GRADIENT BOOSTING TREE

## F.1   QUADRATIC UPPER BOUND OF THE BOOSTING

The iterative minimization problem is given as

$$\min_{\psi_\theta} \mathcal{L}_n\big(g^{(t)} - w_t\psi_\theta(x)\big),$$

and by considering the quadratic upper bound for this

$$\min_{\psi_\theta}\left(-\langle\nabla\mathcal{L}_n(g^{(t)}), w_t\psi_\theta\rangle + \frac{M}{2}\|w_t\psi_\theta\|^2\right) \Leftrightarrow \min_{\psi_\theta}\|Mw_t\psi_\theta - \nabla\|^2.$$

## F.2   FROM WEAK LEARNABILITY TO FUNCTIONAL GRADIENT

We use the empirical inner product and norms

$$\langle a, b\rangle_{L_2(S_n)} := \frac{1}{n}\sum_{i=1}^n a_i b_i, \quad \|f\|_{L_2(S_n)}^2 := \langle f, f\rangle_{L_2(S_n)}, \quad \|v\|_{L_1(S_n)} := \frac{1}{n}\sum_{i=1}^n |v_i|.$$

Consider the binary cross-entropy with logit $z = g(X)$, i.e.,

$$\ell_{\text{ent}}(\tilde{y}, z) = -\tilde{y}\log\sigma(z) - (1 - \tilde{y})\log(1 - \sigma(z)), \quad \sigma(z) = \frac{1}{1 + e^{-z}}.$$

Its per-sample gradient w.r.t. the logit is

$$\nabla_i := \frac{\partial}{\partial z}\ell_{\text{ent}}(\tilde{y}_i, z)\Big|_{z=g(X_i)} = \sigma\big(g(X_i)\big) - \tilde{y}_i.$$

Let $v = (\nabla_1, \ldots, \nabla_n)$.

**Lemma 4.** *Under the same assumptions in Theorem 3, the following holds:*

$$\mathrm{smCE}^\sigma\big(g^{(t)}, S_n\big) \le \|v\|_{L_1(S_n)} \le \frac{1}{\gamma B}\big\langle\psi_t, \nabla_t\big\rangle_{L_2(S_n)} + \frac{Mw_t}{2\gamma}B.$$

*Proof.* Introduce $y_i := 2\tilde{y}_i - 1 \in \{\pm 1\}$. Then

$$\nabla_i = \sigma\big(g(X_i)\big) - \tilde{y}_i = -y_i\sigma\big(-y_ig(X_i)\big) := -y_iq_i, \quad q_i \in (0,1),$$

hence $|\nabla_i| = q_i$. Let $w_i := \dfrac{|\nabla_i|}{\sum_{j=1}^n |\nabla_j|} \in \Delta_n$. By Assumption 2 there exists $\psi^\star \in \Psi$ such that $\sum_{i=1}^n w_iy_i\psi^\star(X_i) \ge \gamma B$. Therefore

$$\big\langle v, \psi^\star\big\rangle_{L_2(S_n)} = \frac{1}{n}\sum_{i=1}^n(-y_i|\nabla_i|)\psi^\star(X_i) = -\frac{1}{n}\Big(\sum_{i=1}^n |\nabla_i|\Big)\Big(\sum_{i=1}^n w_iy_i\psi^\star(X_i)\Big) \le -\gamma B\|v\|_{L_1(S_n)}.$$

Since $\Psi$ is sign-flip closed, $-\psi^\star \in \Psi$, and

$$\big\langle v, -\psi^\star\big\rangle_{L_2(S_n)} \ge \gamma B\|v\|_{L_1(S_n)}. \tag{23}$$

Note that

$$\|Mw_t\psi_{\theta_t} - \nabla\|_{L_2(S_n)}^2 \le \|Mw_t(-\psi^\star) - \nabla_t\|_{L_2(S_n)}^2,$$

which implies that

$$-2Mw_t\big\langle\psi_t, \nabla_t\big\rangle_{L_2(S_n)} \le -2Mw_t\big\langle(-\psi^\star), \nabla_t\big\rangle_{L_2(S_n)} + M^2w_t^2\|\psi^\star\|_{L_2(S_n)}^2,$$

and we obtain

$$\big\langle(-\psi^\star), \nabla_t\big\rangle_{L_2(S_n)} \le \big\langle\psi_t, \nabla_t\big\rangle_{L_2(S_n)} + \frac{Mw_t}{2}B^2. \tag{24}$$

Then combining Eq. (23) and (24) we obtain

$$\gamma B\|v\|_{L_1(S_n)} \le \big\langle\psi_t, \nabla_t\big\rangle_{L_2(S_n)} + \frac{Mw_t}{2}B^2.$$

By definition, we have

$$\mathrm{smCE}^\sigma\big(g^{(t)}, S_n\big) \le \|v\|_{L_1(S_n)}.$$

Since $M = 1/4$, combining the above two inequalities, we obtain the result. $\square$

## F.3 PROOF OF THEOREM 3

We consider the Taylor expansion of the objective

$$L_n(g^{(t+1)}) \le L_n(g^{(t)}) - w_t\langle\nabla, \psi_{\theta_t}\rangle_{L_2(S_n)} + \frac{M}{2}w_t^2\|\psi_{\theta_t}\|_{L_2(S_n)}^2.$$

Here we use Lemma 5 in Appendix F.4. The lemma is the Pythagorean relation of the projection of the function gradient. In the lemma, $w_tM \to w$ and $\nabla \to x$, and $p \to \psi_{\theta_t}$ corresponding to the GBT step. From Eq. (25), setting $q = 0$ and , we get

$$\langle\psi_{\theta_t}, w_tM\psi_{\theta_t} - \nabla_t\rangle_{L_2(S_n)} \le 0,$$

and this leads to

$$Mw_t\|\psi_{\theta_t}\|_n^2 \le \langle\psi_{\theta_t}, \nabla_t\rangle_{L_2(S_n)}.$$

Finally we have

$$L_n(g^{(t+1)}) \le L_n(g^{(t)}) - \frac{w_t}{2}\langle\nabla_t, \psi_{\theta_t}\rangle_{L_2(S_n)}.$$

By summing up from $0$ to $T-1$, we have

$$\frac{1}{2}\sum_{t=0}^{T-1} w_t \langle \nabla_t, \psi_{\theta_t}\rangle_{L_2(S_n)} \le L_n(g^{(0)}) - L_n(g^{(T)}).$$

If we use the constant stepsize $w_t = w$, we have

$$\frac{1}{T}\sum_{t=0}^{T-1} \langle \nabla_t, \psi_{\theta_t}\rangle_{L_2(S_n)} \le \frac{2\big(L_n(g^{(0)}) - L_n(g^{(T)})\big)}{wT}.$$

Combining Lemma 4, we have

$$\frac{1}{T}\sum_{t=0}^{T-1} \text{smCE}^\sigma\big(g^{(t)}, S_n\big) \le \frac{1}{T}\sum_{t=0}^{T-1} \|\nabla_t\|_{L_1(S_n)} \le \frac{1}{2\gamma B} \cdot \frac{1}{T}\sum_{t=0}^{T-1} \langle \nabla_t, \psi_{\theta_t}\rangle_{L_2(S_n)} + \frac{wMB}{2\gamma}$$

$$\le \frac{1}{2\gamma B} \cdot \frac{2\big(L_n(g^{(0)}) - L_n(g^{(T)})\big)}{wT} + \frac{wMB}{2\gamma}.$$

Since the cross entropy loss is always positive, we drop $L_n(g^{(T)})$ and obtain the result.

## F.4 AUXILIARY LEMMAS

**Setting.** Let $X \subset \mathbb{R}^d$ be a nonempty closed convex set and fix a scalar weight $w > 0$. Consider

$$f(z) := \tfrac{1}{2}\|wz - x\|_2^2 = \tfrac{w^2}{2}\big\|z - \tfrac{x}{w}\big\|_2^2 + \text{const}, \qquad x \in \mathbb{R}^d,$$

and let

$$p \in \arg\min_{z \in X} f(z).$$

Equivalently, with $y_0 := x/w$, $p$ is the Euclidean projection $p = \Pi_X(y_0)$.

Following is the alternative lemma of Lemma 3.1 in Bubeck et al. (2015),

**Lemma 5.** *For all $q \in X$, the following relation holds.*

$$\langle p - q, w^2 p - wx\rangle \le 0.$$

*Equivalently, since $w^2 p - wx = w^2(p - y_0)$, we have*

$$\langle p - q, p - y_0\rangle \le 0. \tag{25}$$

*Proof.* The objective $f$ is convex and differentiable with gradient

$$\nabla f(z) = w^2 z - wx.$$

For convex differentiable optimization over a convex set $X$, the necessary and sufficient first-order condition at a minimizer $p$ is

$$\langle \nabla f(p), q - p\rangle \ge 0 \qquad (\forall q \in X).$$

Rewriting gives $\langle \nabla f(p), p - q\rangle \le 0$, i.e. $\langle p - q, w^2 p - wx\rangle \le 0$. $\square$

## F.5 SUMMARY OF THE GRADIENT BOOSTING TREE ALGORITHM

We remark that the regions of a binary tree are defined as follows: Given a node associated with some region $\tilde{R} \subset \mathcal{X} \subset \mathbb{R}^d$, we split the region using a splitting threshold $s \in \mathbb{R}$ and a splitting dimension $j \in [d]$ into two child nodes, $\tilde{R}_L$ and $\tilde{R}_R$, which respectively correspond to the regions $\tilde{R}_L := \{x \in \tilde{R} \mid x_j \le s\}$ and $\tilde{R}_R := \{x \in \tilde{R} \mid x_j > s\}$. By recursively applying this procedure, we obtain partitions $\{R_j\}_{j=1}^J$, where each region corresponds to a hyperrectangular region in $\mathbb{R}^d$.

The set of all such trees defines the following function class:

$$\mathcal{T} = \left\{ x \mapsto \sum_{j=1}^J c_j \cdot \mathbb{1}_{\{x \in R_j\}} \;\middle|\; J \le 2^m, R_j \text{ disjoint}, c_j \in \mathbb{R} \right\}.$$

and $|C_j|$ is bounded by $B$

Let $\nabla_{t,i} = \nabla_g \ell(\sigma(g^{(t)}(X_i)), Y_i)$ denote the functional gradient at the $i$-th training point.

---

**Algorithm 1** Gradient Boosting Tree with Cross-Entropy Loss

---

**Require:** Training data $S_{\mathrm{tr}} = \{(X_i, Y_i)\}_{i=1}^n$, base learner $\psi_\theta \in \mathcal{T}$, loss function $\ell_{\mathrm{ent}}$

**Ensure:** Final logit function $g^{(T)}(x)$

1: Initialize $g^{(0)}(x) = 0$

2: **for** $t = 0, 1, \ldots, T-1$ **do**

3:     Compute functional gradients:

$$\nabla_{t,i} = \nabla_g \ell_{\mathrm{ent}}(\sigma(g^{(t)}(X_i)), Y_i), \quad i = 1, \ldots, n$$

4:     (Optionally set step size $w_t$)

5:     Solve: $\theta_t = \arg\min_{\theta \in \Theta} \frac{1}{n} \sum_{i=1}^n |M w_t \psi_\theta(X_i) - \nabla_{t,i}|^2$

6:     Update the model:

$$g^{(t+1)}(x) = g^{(t)}(x) - w_t \psi_{\theta_t}(x)$$

7: **end for**

8: **return** $g^{(T)}(x)$ or $\bar{g}^{(T)}(x)$

---

### F.6   PROOF OF COROLLARY 2

We first remark that the following relationship holds:

$$\mathrm{smCE}(f, S_{\mathrm{tr}}) \leq \mathrm{smCE}^\sigma(g, S_{\mathrm{tr}}).$$

The proof is nearly identical to that of Lemma 4.7 in Błasiok et al. (2023). By replacing the population expectation over $\mathcal{D}$ with the empirical expectation over the sample $S_{\mathrm{tr}}$, we obtain the result.

Therefore, an upper bound on the dual smooth CE over the training dataset directly implies an upper bound on the standard smooth CE. By combining this with our developed generalization bounds, we obtain the desired result.

We now evaluate the Rademacher complexity. We express the function class of $\bar{g}^{(T)} = \frac{1}{n} \sum_{t=0}^{T-1} g^t$ as $\mathcal{G}$.

Since $\mathcal{F} = \sigma \circ \mathcal{G}$ and $\sigma$ is $1/4$-Lipschitz, Talagrand's contraction lemma yields:

$$\mathfrak{R}_{\mathcal{D},n}(\mathcal{F}) = \mathfrak{R}_{\mathcal{D},n}(\sigma \circ \mathcal{G}) \leq \frac{1}{4} \mathfrak{R}_{\mathcal{D},n}(\mathcal{G}).$$

We now derive an upper bound on the Rademacher complexity of $\mathcal{G}$. Using its sub-additive property, we have:

$$\mathfrak{R}_{\mathcal{D},n}(\mathcal{G}) \leq \frac{1}{T} \sum_{t=0}^{T-1} \mathfrak{R}_{\mathcal{D},n}(\{g(t)\}).$$

where $\{g(t)\}$ is the function class that after $t$-step GBT algorithms.

Then

$$\mathfrak{R}_{\mathcal{D},n}(\{g^{(t)}\}) \leq wt \mathfrak{R}_{\mathcal{D},n}(\mathcal{T}),$$

where $\mathcal{T}$ denotes the class of regression trees.

Furthermore, by Theorem 6.25 in Zhang (2023), the Rademacher complexity is upper bounded by the covering number:

$$\mathfrak{R}_{\mathcal{D},n}(\mathcal{T}) \leq \inf_{\epsilon > 0} \left[ 4\epsilon + 12 \int_\epsilon^1 \sqrt{\frac{\ln N(\epsilon', \mathcal{T}, L_2(S_n))}{n}} d\epsilon' \right].$$

To proceed, we upper bound the covering number for binary regression trees. From Appendix B in Klusowski & Tian (2024), we have:

$$\ln N(\epsilon, \mathcal{T}, L_2(S_n)) \leq \ln(nd)^{2^m} \left( \frac{C}{\epsilon^2} \right)^{2^{m+1}}.$$

Therefore, there exists a universal constant $C > 0$ such that:

$$\mathfrak{R}_{\mathcal{D},n}(\mathcal{T}) \leq C\sqrt{\frac{2^m \log(nd)}{n}}.$$

In conclusion, we have that

$$\mathfrak{R}_{\mathcal{D},n}(\mathcal{F}) \leq CwT\sqrt{\frac{2^m \log(nd)}{n}}.$$

Finally, substituting this into Theorem 2, we obtain the result.

$$\text{smCE}^\sigma\Big(\frac{1}{T}\sum_{t=0}^{T-1} g^{(t)}, S_n\Big) \leq \frac{L_n(g^{(0)})}{\gamma BwT} + \frac{wMB}{2\gamma} + wTC\sqrt{\frac{2^m \log(nd)}{n}} + \sqrt{\frac{\log\frac{2}{\delta}}{n}}.$$

### F.7 MISCLASSIFICATION RATE

Let $S_n$ denote the empirical distribution of the training dataset. From standard margin-based generalization theory, for any $\rho > 0$, with probability at least $1 - \delta$, the following inequality holds:

$$P_{(X,Y)\sim\mathcal{D}}[(2Y-1)\bar{g}^{(T)}(X) \leq 0] \leq P_{(X,Y)\sim S_n}[(2Y-1)\bar{g}^{(T)}(X) \leq \rho] + \frac{2}{\rho}\mathfrak{R}_{\mathcal{D},n}(\mathcal{G}) + \sqrt{\frac{\log\frac{1}{\delta}}{2n}}.$$

According to Nitanda & Suzuki (2018); Nitanda et al. (2019), the empirical margin error can be upper-bounded by the functional gradient as follows:

$$P_{(X,Y)\sim S_n}[(2Y-1)\bar{g}^{(T)}(X) \leq \rho] \leq (1 + e^\rho)\|\nabla_g \ell_{\text{ent}}(\bar{g}^{(T)}(X), Y)\|_{L_1(S_n)}.$$

Combining the above results, we obtain:

$$P_{(X,Y)\sim\mathcal{D}}[(2Y-1)\bar{g}^{(T)}(X) \leq 0] \leq (1 + e^\rho)\left(\frac{L_n(g^{(0)})}{\gamma BwT} + \frac{wB}{8\gamma}\right) + \frac{CwT}{\rho}\sqrt{\frac{2^J \log(nd)}{n}} + \sqrt{\frac{\log\frac{1}{\delta}}{2n}}.$$

This result illustrates a trade-off with respect to $wT$, which balances the training misclassification rate and the complexity of the hypothesis class.

## G PROOFS FOR THE KERNEL BOOSTING

### G.1 PROOF OF EQ. (4.2)

From the margin assumption, the following property holds:

$$\phi(x_i) > \gamma \quad \text{if } y_i = 1, \quad -\phi(x_i) > \gamma \quad \text{if } y_i = 0, \quad \text{for all } (x_i, y_i) \in S_{\text{tr}}.$$

Since $\phi \in \mathcal{H}$ implies $-\phi \in \mathcal{H}$, we define $\tilde{\phi} := -\phi$. Then, the margin assumption implies:

$$\tilde{\phi}(x_i) < -\gamma \quad \text{if } y_i = 1, \quad \tilde{\phi}(x_i) > \gamma \quad \text{if } y_i = 0, \quad \text{for all } (x_i, y_i) \in S_{\text{tr}}.$$

We now analyze the gradient in the RKHS under this construction:

$$\begin{aligned}
\|\mathcal{T}_k\nabla_g L_n(g)\|_{\mathcal{H}} &= \left\langle \mathcal{T}_k\nabla_g L_n(g), \frac{\mathcal{T}_k\nabla_g L_n(g)}{\|\mathcal{T}_k\nabla_g L_n(g)\|_{\mathcal{H}}} \right\rangle_{\mathcal{H}} \qquad (26)\\
&= \sup_{\phi\in\mathcal{H},\|\phi\|_{\mathcal{H}}\leq 1} \langle \mathcal{T}_k\nabla_g L_n(g), \phi\rangle_{\mathcal{H}}\\
&\geq \left\langle \nabla_g L_n(g), \tilde{\phi}\right\rangle_{L_2(S_n)}\\
&= \frac{1}{n}\sum_{i=1}^n \nabla_g\ell(g(X_i), Y_i)\cdot\tilde{\phi}(X_i)\\
&= \frac{1}{n}\sum_{i=1}^n (\sigma(g(X_i)) - Y_i)\cdot\tilde{\phi}(X_i),
\end{aligned}$$

where the last equality follows from the functional gradient.

Using the margin assumption, we obtain:

$$\frac{1}{n}\sum_{i=1}^{n}(Y_i - \sigma(g(X_i))) \cdot (-\tilde{\phi}(X_i)) \geq \frac{\gamma}{n}\sum_{i=1}^{n}|\nabla_g\ell(g(X_i), Y_i)|.$$

This completes the proof.

### G.2 PROOF OF THEOREM 4

Assume that the loss function $L(g)$ is $M$-Lipschitz smooth with respect to $g$, meaning that for all $g, g'$,

$$L(g') \leq L(g) + \langle \nabla L(g), g' - g \rangle_{L_2(S_n)} + \frac{M}{2}\|g' - g\|_{L_2(S_n)}^2.$$

Plugging in $g = g^{(t)}$ and $g' = g^{(t+1)} = g^{(t)} - w_t \mathcal{T}_k \nabla_g L(g^{(t)})$, we obtain:

$$\begin{aligned} L(g^{(t+1)}) &\leq L(g^{(t)}) - w_t \langle \nabla L(g^{(t)}), \mathcal{T}_k \nabla_g L(g^{(t)}) \rangle_{L_2(S_n)} \\ &\quad + \frac{M}{2}w_t^2\|\mathcal{T}_k \nabla_g L(g^{(t)})\|_{L_2(S_n)}^2 \\ &\leq L(g^{(t)}) - w_t\|\mathcal{T}_k \nabla_g L_n(g^{(t)})\|_{\mathcal{H}}^2 + \frac{M\Lambda}{2}w_t^2\|\mathcal{T}_k \nabla_g L_n(g^{(t)})\|_{\mathcal{H}}^2 \\ &= L(g^{(t)}) - w_t\left(1 - \frac{w_t M\Lambda}{2}\right)\|\mathcal{T}_k \nabla_g L_n(g^{(t)})\|_{\mathcal{H}}^2 \\ &\leq L(g^{(t)}) - \frac{w_t}{2}\|\mathcal{T}_k \nabla_g L_n(g^{(t)})\|_{\mathcal{H}}^2, \end{aligned}$$

where the last inequality uses $w_t \leq \frac{1}{M\Lambda}$.

Summing over $t = 0, \ldots, T - 1$, we get:

$$\sum_{t=0}^{T-1}\frac{1}{2}w_t\|\mathcal{T}_k \nabla_g L_n(g^{(t)})\|_{\mathcal{H}}^2 \leq L_n(g^{(0)}).$$

In particular, using constant step size $w_t = w$, we obtain:

$$\frac{1}{T}\sum_{t=0}^{T-1}\|\mathcal{T}_k \nabla_g L_n(g^{(t)})\|_{\mathcal{H}}^2 \leq \frac{2}{wT}L_n(g^{(0)}).$$

From Eq. (26), we have:

$$\frac{1}{T}\sum_{t=0}^{T-1}\left\|\nabla_g\ell(g^{(t)}(X), Y)\right\|_{L_1(S_n)}^2 \leq \frac{2}{wT}L_n(g^{(0)}).$$

Then, using Jensen's inequality and the definition of $\bar{g}^{(T)} := \frac{1}{T}\sum_{t=0}^{T-1}g^{(t)}$, we get:

$$\left\|\nabla_g\ell(\bar{g}^{(T)}(X), Y)\right\|_{L_1(S_n)}^2 \leq \frac{1}{T}\sum_{t=0}^{T-1}\left\|\nabla_g\ell(g^{(t)}(X), Y)\right\|_{L_1(S_n)}^2.$$

Combining these gives:

$$\left\|\nabla_g\ell(\bar{g}^{(T)}(X), Y)\right\|_{L_1(S_n)} \leq \sqrt{\frac{2}{wT}L_n(g^{(0)})}.$$

**Generalization Bound via Rademacher Complexity.**

We now evaluate the Rademacher complexity. We express the function class of $\bar{g}^{(T)} = \frac{1}{n}\sum_{t=0}^{T-1} g^t$ as $\mathcal{G}$. Since $\mathcal{F} = \sigma \circ \mathcal{G}$ and $\sigma$ is $1/4$-Lipschitz, Talagrand's contraction lemma yields:

$$\mathfrak{R}_{\mathcal{D},n}(\mathcal{F}) = \mathfrak{R}_{\mathcal{D},n}(\sigma \circ \mathcal{G}) \leq \frac{1}{4}\mathfrak{R}_{\mathcal{D},n}(\mathcal{G}) \leq \frac{\alpha}{4}\sqrt{\frac{\Lambda}{n}},$$

where $\alpha$ is an upper bound on the RKHS norm of $\mathcal{G}$, which we estimate below. The final inequality follows from the standard Rademacher complexity estimate of the RKHS, see Mohri et al. (2018) for the details.

Recall the recursion:

$$g^{(T)} = g^{(0)} - w\sum_{t=0}^{T-1} \mathcal{T}_k \nabla_g L_n(g^{(t)}).$$

Then:

$$\|g^{(T)}\|_{\mathcal{H}} \leq \|g^{(0)}\|_{\mathcal{H}} + \left\|w\sum_{t=0}^{T-1} \mathcal{T}_k \nabla_g L_n(g^{(t)})\right\|_{\mathcal{H}}.$$

Applying Jensen's inequality:

$$\|w\sum_{t=0}^{T-1} \mathcal{T}_k \nabla_g L_n(g^{(t)})\|_{\mathcal{H}}^2 \leq \frac{1}{T}\sum_{t=0}^{T-1} \|wT\mathcal{T}_k \nabla_g L_n(g^{(t)})\|_{\mathcal{H}}^2 \leq w^2 T^2 \frac{1}{T}\sum_{t=0}^{T-1} \|\mathcal{T}_k \nabla_g L_n(g^{(t)})\|_{\mathcal{H}}^2$$

$$\leq w^2 T^2 \frac{2}{wT} L_n(g^{(0)})$$

$$\leq 2wT L_n(g^{(0)})$$

Assuming $\|g^{(0)}\|_{\mathcal{H}} \leq \Lambda'$, we obtain:

$$\|g^{(T)}\|_{\mathcal{H}} \leq \Lambda' + \sqrt{2wT L_n(g^{(0)})}.$$

Absorbing $\sqrt{2L_n(g^{(0)})}$ into a universal constant. When considering the norm of $\bar{g}^{(T)}$, it is also $\mathcal{O}(\Lambda' + \sqrt{wT})$. Thus, we conclude $\alpha = \mathcal{O}(\Lambda' + \sqrt{wT})$. Substituting this into the Rademacher bound yields the final generalization result.

### G.3 PROOF OF MISCLASSIFICATION RATE

Let $S_n$ denote the empirical distribution of the training dataset. From standard margin-based generalization theory, for any $\rho > 0$, with probability at least $1 - \delta$, the following inequality holds:

$$P_{(X,Y)\sim\mathcal{D}}[(2Y-1)\bar{g}^{(T)}(X) \leq 0] \leq P_{(X,Y)\sim S_n}[(2Y-1)\bar{g}^{(T)}(X) \leq \rho] + \frac{2}{\rho}\mathfrak{R}_{\mathcal{D},n}(\mathcal{G}) + \sqrt{\frac{\log\frac{1}{\delta}}{2n}}.$$

According to Nitanda & Suzuki (2018); Nitanda et al. (2019), the empirical margin error can be upper-bounded by the functional gradient as follows:

$$P_{(X,Y)\sim S_n}[(2Y-1)\bar{g}^{(T)}(X) \leq \rho] \leq (1 + e^\rho)\|\nabla_g \ell_{\text{ent}}(\bar{g}^{(T)}(X), Y)\|_{L_1(S_n)}.$$

Combining the above results, we obtain:

$$P_{(X,Y)\sim\mathcal{D}}[(2Y-1)\bar{g}^{(T)}(X) \leq 0] \leq \frac{(1+e^\rho)}{\gamma}\sqrt{\frac{L_n(g^{(0)})}{wT}} + \frac{2(\Lambda' + \sqrt{2wTL_n(g^{(0)})})}{\rho}\sqrt{\frac{\Lambda}{n}} + \sqrt{\frac{\log\frac{1}{\delta}}{2n}}.$$

This result illustrates a trade-off with respect to $wT$, which balances the training misclassification rate and the complexity of the hypothesis class.

### G.4 PROOF OF COROLLARY 4

We first remark that both the smooth calibration error and the misclassification rate exhibit similar asymptotic behavior with respect to $n$ and $wT$. By substituting the given hyperparameters into the bound presented in Theorem 4, we obtain the desired result.

## H FORMAL SETTINGS OF THE TWO-LAYER NEURAL NETWORK

### H.1 FORMAL RESULTS FOR THE SMOOTH CE

Here we provide the formal setting of the results from Nitanda et al. (2019). We assume $Y = \{\pm 1\}$ and denote by $\nu$ the true probability measure on $\mathcal{X} \times \mathcal{Y}$, and by $\nu_n$ the empirical measure based on samples $\{(X_i, Y_i)\}_{i=1}^n$ drawn independently from $\nu$, i.e.,

$$d\nu_n(X, Y) = \frac{1}{n} \sum_{i=1}^n \delta_{(X_i, Y_i)}(X, Y) dX dY,$$

where $\delta$ denotes the Dirac delta function. In the main paper, we consider labels in $\tilde{Y} = \{0, 1\}$; here, we convert them to $Y = 2\tilde{Y} - 1$, so the distribution $\mathcal{D}$ over $\mathcal{X} \times \tilde{Y}$ becomes $\mu$ over $\mathcal{X} \times Y$.

The marginal distributions of $\nu$ and $\nu_n$ over $\mathcal{X}$ are denoted by $\nu_{\mathcal{X}}$ and $\nu_{\mathcal{X},n}$, respectively. For $s \in \mathbb{R}$ and $y \in \mathcal{Y}$, let $\ell(s, y)$ denote the logistic loss:

$$\ell(s, y) = \log(1 + \exp(-ys)).$$

We remark that given a logit value $s$, the predicted probability of $Y = 1$ is $p = \sigma(s)$, and the loss can also be written as $\ell(s, y) = \frac{y+1}{2} \log p + \frac{1-y}{2} \log(1 - p)$. Replacing $Y$ with $\tilde{Y}$ recovers the standard cross-entropy loss: $\ell(s, \tilde{y}) = \tilde{y} \log p + (1 - \tilde{y}) \log(1 - p)$.

The empirical objective to be minimized is defined as

$$L_n(\theta) \coloneqq \mathbb{E}_{(X,Y)\sim\nu_n} [\ell(g_\theta(X), Y)] = \frac{1}{n} \sum_{i=1}^n \ell(g_\theta(X_i), Y_i),$$

where $g_\theta : \mathcal{X} \to \mathbb{R}$ is a two-layer neural network parameterized by $\theta = (\theta_r)_{r=1}^m$. When treating the function $g_\theta$ as the variable of the objective, we also write $L_n(g_\theta) \coloneqq L_n(\theta)$.

We now introduce the following formal assumptions.

**Assumption 1.**

(A1) Assume that $\operatorname{supp}(\nu_{\mathcal{X}}) \subset \{x \in \mathcal{X} \mid \|x\|_2 \le 1\}$. Let $\sigma$ be a $C^2$-class function, and there exist constants $K_1, K_2 > 0$ such that

$$\|\sigma'\|_\infty \le K_1 \quad \text{and} \quad \|\sigma''\|_\infty \le K_2.$$

(A2) A distribution $\mu_0$ on $\mathbb{R}^d$, used for the initialization of $\theta_r$, has a sub-Gaussian tail bound: there exist constants $A, b > 0$ such that

$$\mathbb{P}_{\theta^{(0)}\sim\mu_0} \left[ \|\theta^{(0)}\|_2 \ge t \right] \le A \exp(-bt^2).$$

(A3) Assume that the number of hidden units $m \in \mathbb{Z}_+$ is even. The constant parameters $(a_r)_{r=1}^m$ and parameters $\theta^{(0)} = (\theta_r^{(0)})_{r=1}^m$ are initialized symmetrically as follows:

$$a_r = 1 \quad \text{for } r \in \left\{1, \dots, \frac{m}{2}\right\}, \quad a_r = -1 \quad \text{for } r \in \left\{\frac{m}{2} + 1, \dots, m\right\},$$

and

$$\theta_r^{(0)} = \theta_{r+\frac{m}{2}}^{(0)} \quad \text{for } r \in \left\{1, \dots, \frac{m}{2}\right\},$$

where the initial parameters $(\theta_r^{(0)})_{r=1}^{m/2}$ are independently drawn from the distribution $\mu_0$.

(A4) Assume that there exist $\gamma > 0$ and a measurable function $v : \mathbb{R}^d \to \{w \in \mathbb{R}^d \mid \|w\|_2 \le 1\}$ such that the following inequality holds for all $(x, y) \in \operatorname{supp}(\nu) \subset \mathcal{X} \times \mathcal{Y}$:

$$y \left\langle \partial_\theta \sigma(\theta^{(0)} \cdot x), v(\theta^{(0)}) \right\rangle_{L^2(\mu_0)} = y \mathbb{E}_{\theta^{(0)}\sim\mu_0} \left[ \partial_\theta \sigma(\theta^{(0)\top} x) \cdot v(\theta^{(0)}) \right] \ge \gamma.$$

**Remark.** Clearly, many activation functions (sigmoid, tanh, and smooth approximations of ReLU such as swish) satisfy assumption (A1). Typical distributions, including the Gaussian distribution, satisfy (A2). The purpose of the symmetrized initialization (A3) is to uniformly bound the initial value of the loss function $L_n(\theta^{(0)})$ over the number of hidden units $m$. This initialization leads to $f_{\theta^{(0)}}(x) = 0$, resulting in $L_n(\theta^{(0)}) = \log(2)$. Assumption (A4) implies the separability of a dataset using the neural tangent kernel. We next discuss the validity of this assumption.

**Theorem 6** (Global Convergence, Theorem 2 in Nitanda et al. (2019)). *Suppose Assumption 1 holds. We set constant $K$ as*

$$K = K_1^4 + 2K_1^2 K_2 + K_1^4 K_2^2.$$

*For all $\beta \in [0, 1)$, $\delta \in (0, 1)$, and $m \in \mathbb{Z}_+$ such that*

$$m \geq \frac{16K_1^2}{\gamma^2} \log \frac{2n}{\delta},$$

*consider gradient descent* (6) *with learning rate*

$$0 < w \leq \min \left\{ \frac{1}{m^\beta}, \frac{4m^{2\beta-1}}{K_1^2 + K_2} \right\},$$

*and the number of iterations*

$$T \leq \left\lfloor \frac{m\gamma^2}{32w K_2^2 \log(2)} \right\rfloor.$$

*Then, with probability at least $1 - \delta$ over the random initialization, we have:*

$$\frac{1}{T} \sum_{t=0}^{T-1} \|\nabla_g L_n(g_{\theta^{(t)}})\|_{L^1(\nu_{\mathcal{X},n})}^2 \leq \frac{16 \log(2)}{\gamma^2 T} \left( \frac{m^{2\beta-1}}{w} + K \right).$$

*where we define the $L^1$-norm of the functional gradient as*

$$\|\nabla_g L_n(g_\theta)\|_{L^1(\nu_{\mathcal{X},n})} := \frac{1}{n} \sum_{i=1}^n |\partial_g \ell(g_\theta(X_i), Y_i)| = \frac{1}{2n} \sum_{i=1}^n |Y_i - 2p_\theta(Y = 1 \mid x_i) + 1|.$$

The result above corresponds to Eq. (6). This bound provides a guarantee on the training smooth CE. Proposition 8 in Nitanda et al. (2019) yields estimates for the Rademacher complexity and covering numbers.

**Theorem 7** (Proposition 8 in Nitanda et al. (2019)). *Suppose Assumptions **(A1)** and **(A2)** hold. Let $\forall w > 0$, $\forall m \in \mathbb{Z}_+$, $\forall T \in \mathbb{Z}_+$, $\forall \delta \in (0, 1)$, and $\forall S$ of size $n$. Then, there exists a universal constant $C > 0$ such that with probability at least $1 - \delta$ with respect to the initialization of $\Theta^{(0)}$, the empirical Rademacher complexity satisfies:*

$$\hat{\mathfrak{R}}_S(\mathcal{F}) \leq C m^{1/2-\beta} D_{w,m,T} (1 + K_1 + K_2)$$
$$\cdot \sqrt{\frac{d}{n} \log \left( n(1 + K_1 + K_2) \left( \log(m/\delta) + D_{w,m,T}^2 \right) \right)}. \tag{29}$$

*where*

$$D_{w,m,T} = \sqrt{wT}$$

*Moreover, when $\sigma$ is convex and $\sigma(0) = 0$, with probability at least $1 - \delta$ over a random initialization of $\theta^{(0)}$, we have:*

$$\hat{\mathfrak{R}}_S(\mathcal{F}) \leq \frac{8K_1 m^{1/2-\beta}}{\sqrt{n}} \left( D_{w,m,T} + \sqrt{\frac{\log(Am/\delta)}{b}} \right). \tag{30}$$

By substituting Theorems 6 and 7 into the uniform convergence bound in Theorem 2, we obtain the following guarantee on the smooth CE for two-layer neural networks.

**Corollary 7.** *Suppose Assumption 1 and the settings of Theorem 6 hold. Then with probability $1 - \delta$ over the random draw of $S_{\mathrm{tr}}$ and initial parameter $\theta_0$, we have*

$$\min_{t \in \{0,\dots,T-1\}} \mathrm{smCE}^\sigma(g_{\theta^{(t)}}, \mu) \le C_1 \sqrt{\frac{1}{\gamma^2 T} \left( \frac{m^{2\beta-1}}{w} + K \right)} + \tag{31}$$

$$\frac{C_2}{\sqrt{n}} + C_3 m^{1/2-\beta} D_{w,m,T}(1 + K_1 + K_2) \cdot$$

$$\cdot \sqrt{\frac{d}{n} \log\left( n(1 + K_1 + K_2) \left( \log(2m/\delta) + D_{w,m,T}^2 \right) \right)} + 6\sqrt{\frac{\log \frac{6}{\delta}}{n}}.$$

*where $C_1, \dots, C_3$ are universal constants.*

*Moreover, when $\sigma$ is convex and $\sigma(0) = 0$, with probability at least $1 - \delta$ over the random draw of $S_{\mathrm{tr}}$ and initial parameter $\theta_0$, such that we have:*

$$\min_{t \in \{0,\dots,T-1\}} \mathrm{smCE}^\sigma(g_{\theta^{(t)}}, \mu) \le C_1 \sqrt{\frac{1}{\gamma^2 T} \left( \frac{m^{2\beta-1}}{w} + K \right)} +$$

$$\frac{C_2}{\sqrt{n}} + \frac{C_4 K_1 m^{1/2-\beta}}{\sqrt{n}} \left( D_{w,m,T} + \sqrt{\frac{\log(2Am/\delta)}{b}} \right) + 6\sqrt{\frac{\log \frac{6}{\delta}}{n}}.$$

*where $C_4$ is a universal constant.*

*Proof.* We use the following upper bound from Mohri et al. (2018): with probability $1 - \delta/2$, we have

$$\mathfrak{R}_{\mathcal{D},n}(\mathcal{F}) \le \hat{\mathfrak{R}}_S(\mathcal{F}) + \sqrt{\frac{\log \frac{2}{\delta}}{2n}}.$$

Then from Theorem 2 with probability $1 - \delta$, we have

$$\sup_{f \in \mathcal{F}} |\mathrm{smCE}(f, \mathcal{D}) - \mathrm{smCE}(f, S_{\mathrm{tr}})| \le \frac{C_2}{\sqrt{n}} + 4\hat{\mathfrak{R}}_S(\mathcal{F}) + 6\sqrt{\frac{\log \frac{3}{\delta}}{n}}.$$

We then substitute Eq. (29) and (30) and taking the union bound, we obtain the result. $\square$

For completeness, we include the result on the classification error, adapted from Theorem 4 in Nitanda & Suzuki (2018):

**Theorem 8** (Theorem 4 in Nitanda & Suzuki (2018)). *Suppose Assumption 1 and the settings of Theorem 6 hold. Fix $\forall \epsilon > 0$. Then, with probability at least $1 - 3\delta$ over a random initialization and random choice of training dataset $S_{\mathrm{tr}}$, we have*

$$\min_{t \in \{0,\dots,T-1\}} \mathbb{P}_{(X,Y)\sim\nu} \left[ Y g_{\theta^{(t)}}(X) \le 0 \right] \le C_5 (1 + \exp(\epsilon)) C_{w,m,T} + 3\sqrt{\frac{\log(2/\delta)}{2n}}$$

$$+ C_6 m^{1/2-\beta} D_{w,m,T}(1 + K_1 + K_2) \cdot \sqrt{\frac{d}{n} \log\left( n(1 + K_1 + K_2) \left( \log(m/\delta) + D_{w,m,T}^2 \right) \right)},$$

*where*

$$C_{w,m,T} = \gamma^{-1} T^{-1/2} \left( m^{\beta-1/2} w^{-1/2} + \sqrt{K} \right)$$

*and $C_5$ and $C_6$ are universal constants.*

*Moreover, when $\sigma$ is convex and $\sigma(0) = 0$, we can avoid the dependence on the dimension $d$. With probability at least $1 - 3\delta$ over a random initialization and random choice of training dataset $S_{\mathrm{tr}}$,*

$$\min_{t \in \{0,\dots,T-1\}} \mathbb{P}_{(X,Y)\sim\nu} \left[ Y g_{\theta^{(t)}}(X) \le 0 \right] \le C_5 (1 + \exp(\epsilon)) C_{w,m,T} + 3\sqrt{\frac{\log(2/\delta)}{2n}}$$

$$+ C_7 K_1 m^{1/2-\beta} \frac{1}{\epsilon\sqrt{n}} \left( D_{w,m,T} + \sqrt{\frac{\log(Am/\delta)}{b}} \right).$$

*where $C_7$ is a universal constant.*

## H.2 Formal statement for Corollary 5

We restate Corollary 5 with formal assumptions

**Corollary 8.** *Suppose Assumption 1 and the settings of Theorem 6 hold. If for any $\epsilon > 0$, the hyperparameters satisfy one of the following:*
*(i) $\beta \in [0,1)$, $m = \Omega(\gamma^{\frac{-2}{1-\beta}}\epsilon^{\frac{-1}{1-\beta}})$, $T = \Omega(\gamma^{-2}\epsilon^{-2})$, $w = \Theta(\gamma^{-2}\epsilon^{-2}T^{-1}m^{2\beta-1})$, $n = \tilde{\Omega}(\gamma^{-2}\epsilon^{-4})$,*
*(ii) $\beta = 0$, $m = \Theta\left(\gamma^{-2}\epsilon^{-3/2}\log(1/\epsilon)\right)$, $T = \Theta\left(\gamma^{-2}\epsilon^{-1}\log^2(1/\epsilon)\right)$, $w = \Theta(m^{-1})$, $n = \tilde{\Omega}(\epsilon^{-2})$,*
*then with probability at least $1 - \delta$, gradient descent with the stepsize $w$ finds a parameter $\theta^{(t)}$ satisfying $\mathrm{smCE}(g_{\theta^{(t)}}, S_{\mathrm{tr}}) \leq \epsilon$ and $\mathbb{P}_{(X,Y)\sim\nu}\left[Y g_{\theta^{(t)}}(X) \leq 0\right] \leq \epsilon$ within $T$ iterations.*

*Proof.* The guarantee for the misclassification rate is adapted from Corollary 3 in Nitanda & Suzuki (2018). As for the smooth CE, since the complexity terms in the misclassification bound (Theorem 8) and the smooth CE bound (Corollary 7) are of the same order, we also obtain a corresponding guarantee for the smooth CE. □

## I Relationships of the proper scoring rule and functional gradient

In the main paper, we discussed the relationships involving the functional gradient only over the training dataset. Here, we provide the corresponding results for the population smooth CE.

We begin by connecting the post-processing gap with the functional gradient. First, we recall the definition of the functional gradient:

**Definition 11.** *Let $\mathcal{H}$ be a Hilbert space and $h$ be a function on $\mathcal{H}$. For $\xi \in \mathcal{H}$, we say that $h$ is Fréchet differentiable at $\xi$ in $\mathcal{H}$ if there exists an element $\nabla_\xi h(\xi) \in \mathcal{H}$ such that*

$$h(\zeta) = h(\xi) + \langle \nabla_\xi h(\xi), \zeta - \xi \rangle_\mathcal{H} + o(\|\xi - \zeta\|_\mathcal{H}).$$

*Moreover, for simplicity, we refer to $\nabla_\xi h(\xi)$ as the functional gradient.*

See Luenberger (1997) for more details.

Next, we focus on the case of the squared loss. To simplify the notation, we express $r(f(x)) = f(x) + h(f(x))$ where $h$ is the post-processing function in the definition of the smooth CE. Then we set $h \to \ell_{\mathrm{sq}}$, $\zeta \to f(x)$, and $\xi \to r(f(x))$ in the definition of the functional gradient. This results in the following relation.

$$\begin{aligned}
&\ell_{\mathrm{sq}}(f(x), y) \\
&= \ell_{\mathrm{sq}}(r(f(x)), y) - \nabla_f \ell_{\mathrm{sq}}(f(x), y) \cdot (r(f(x)) - f(x)) - \frac{1}{2}\|r(f(x)) - f(x)\|^2,
\end{aligned} \tag{32}$$

where $\nabla_f \ell_{\mathrm{sq}}(f(x), y)$ denotes the partial derivative of $\ell_{\mathrm{sq}}(f, y)$ with respect to $f$. Thus, we observe that $\nabla_f \ell_{\mathrm{sq}}(f(x), y)$ corresponds to the functional gradient. Therefore,

$$\begin{aligned}
\mathrm{smCE}(f, \mathcal{D})^2 \leq \mathrm{pGap}(f, \mathcal{D}) &= \sup_h \mathbb{E}\left[-\nabla_f \ell_{\mathrm{sq}}(f(X), Y) \cdot h(f(X)) - \frac{1}{2}\|h(f(X))\|^2\right] \\
&\leq \sup_h \mathbb{E}\left[-\nabla_f \ell_{\mathrm{sq}}(f(X), Y) \cdot h(f(X))\right],
\end{aligned}$$

where the supremum is taken over all 1-Lipschitz functions.

Next, for the cross-entropy loss, recall that $\nabla_s^2 \ell^\psi(s, y) = p(1-p)$, where $p = \mathrm{pred}_\psi(s) = e^s/(1 + e^s)$. This implies that the cross-entropy loss is $1/4$-Lipschitz with respect to $s$. Therefore, by Taylor's theorem, there exists $\tilde{p} \in (0, 1/4]$ such that

$$\ell^\psi(g(x), y) = \ell^\psi(r(g(x)), y) - \nabla_g \ell^\psi(g(x), y) \cdot (r(g(x)) - g(x)) - \frac{1}{2}\tilde{p}(1-\tilde{p})\|r(g(x)) - g(x)\|^2. \tag{33}$$

Thus, we obtain

$$
\begin{aligned}
2\text{smCE}^{(\sigma)}(g, \mathcal{D})^2 &\leq \text{pGap}^{(\psi, 1/4)}(g, \mathcal{D}) \\
&= \sup_h \mathbb{E}\left[ -\nabla_g \ell^\psi(g(X), Y) \cdot h(g(X)) - \frac{1}{2}\tilde{p}(1-\tilde{p})\|h(g(X))\|^2 \right] \\
&\leq \sup_h \mathbb{E}\left[ -\nabla_g \ell^\psi(g(X), Y) \cdot h(g(X)) \right],
\end{aligned}
$$

where the supremum is taken over all $1/4$-Lipschitz functions.

From Eqs. (32) and (33), we obtain upper bounds on the training smooth CE in terms of the functional gradient evaluated on the training dataset:

$$
\text{smCE}(f, S_{\text{tr}})^2 \leq \sup_\eta \frac{1}{n} \sum_{i=1}^n \left[ -\nabla_f \ell_{\text{sq}}(f(X_i), Y_i) \cdot \eta(f(X_i)) \right],
$$

and

$$
2\text{smCE}^{(\psi, 1/4)}(g, S_{\text{tr}})^2 \leq \sup_\eta \frac{1}{n} \sum_{i=1}^n \left[ -\nabla_g \ell^\psi(g(X_i), Y_i) \cdot \eta(g(X_i)) \right].
$$

Finally, we remark on the case of general proper losses. As discussed in Appendix A, the post-processing gap can be defined more generally. If the loss is Fréchet differentiable, then similar relationships to those derived above for the cross-entropy loss can be obtained.

## J  DISCUSSION

### J.1  IMPLICATIONS OF THE BOUNDS IN SECTION 4

Section 4 develops an algorithm-dependent, finite-sample analysis of smooth CE for three representative learning procedures: Gradient Boosting trees (GBTs), Kernel Boosting in RKHS, and Two-layer Neural Networks. In contrast to most existing calibration results, which focus on population-level predictors, these bounds are stated directly for concrete algorithms and make explicit how optimization and generalization jointly control the accuracy and smooth CE.

Technically, the analysis proceeds in three steps for each algorithm. The first is a functional gradient analysis to evaluate the training smooth CE $\text{smCE}(f, S_{\text{tr}})$, which quantifies how optimization (number of iterations, step size, regularization) reduces $\text{smCE}$ on the training sample. The second step is a control of the discrepancy $|\text{smCE}(f, \mathcal{D}) - \text{smCE}(f, S_{\text{tr}})|$ using the uniform convergence bounds for smooth CE derived in Section 3, specialized to each hypothesis class. Combining these two ingredients yields explicit conditions on the number of iterations, regularization strength, and model complexity under which test accuracy and $\text{smCE}(f, \mathcal{D})$ are simultaneously controlled with the same order.

For each algorithm, we summarize concisely which parts are new and how they are connected to the above objectives as follows.

**Section 4.1: Gradient Boosting Trees (GBT).** Under a margin-type condition, we establish convergence of the $L^1$-norm of the functional gradient associated with $\text{smCE}(f, S_{\text{tr}})$, formalized in Theorem 3. This provides an explicit relationship between the number of boosting iterations $T$ and the decrease of the training smooth calibration error.

For the generalization step, we invoke the uniform convergence bound for smooth CE from Section 3 and relate the reduction in $\text{smCE}(f, S_{\text{tr}})$ to the reduction in $\text{smCE}(f, \mathcal{D})$ as a function of $n$ and the complexity of the underlying tree class (Corollary 2). This makes explicit how the generalization gap in smooth CE depends on the number of boosting iterations $T$.

The accuracy bounds themselves rely on existing GBT analyses, but Corollary 3 combines them with the smooth CE bounds to yield a simultaneous guarantee for test accuracy and $\text{smCE}(f, \mathcal{D})$. This formalizes the common heuristic that, for GBT, suitable early stopping can yield models that are both accurate and well-calibrated.

**Section 4.2: Kernel Boosting in RKHS.** For kernel boosting, we relate the functional-gradient convergence analysis to the RKHS norm and establish convergence under conditions tailored to smooth CE (Theorem 4). This is closely related to previous analyses of kernel boosting (Wei et al., 2017; Nitanda & Suzuki, 2018), but the assumptions and objective are adapted to the smooth calibration setting. The result shows that, for RKHS-based boosting, the decay of the functional gradient along iterations translates into an improvement of the training $\mathrm{smCE}(f, S_{\mathrm{tr}})$.

By combining Theorem 4 with the uniform convergence bound from Section 3, we obtain explicit test bounds in which the RKHS norm, the kernel choice, and the number of boosting iterations jointly determine both accuracy and the generalization of $\mathrm{smCE}(f, \mathcal{D})$.

As in the GBT case, the accuracy analysis builds on known results, while Corollary 4 provides a simultaneous guarantee for accuracy and smooth CE. This yields a concrete guideline on how to choose regularization and the number of iterations so that kernel methods achieve both good predictive performance and calibration.

**Section 4.3.: Two-layer Neural Networks.** For two-layer neural networks, we directly use the functional gradient norm bounds from Nitanda et al. (2019); The novelty of this section lies in combining these optimization bounds with the smooth CE generalization bound from Section 3. This leads to Corollary 5, which provides a simultaneous guarantee on test accuracy and $\mathrm{smCE}(f, \mathcal{D})$ for two-layer networks, under appropriate choices of iteration number and regularization.

In this way, the analysis offers a theoretical lens on the empirical observation that neural networks can be miscalibrated: it identifies a training regime (in terms of functional-gradient control and uniform convergence of smooth CE) under which smooth calibration can be quantitatively guaranteed alongside accuracy.

### J.2 LIMITATION OF THE MARGIN ASSUMPTION

In the main analysis, the margin assumption is used to control the functional gradient and to derive an upper bound on the smooth calibration error $\mathrm{smCE}$ over the training dataset. In the RKHS setting, however, there is an alternative route to controlling $\mathrm{smCE}$. Consider an RKHS $\mathcal{H}$ that is closed under composition with Lipschitz functions, in the sense that for any $f \in \mathcal{H}$ and any $h \in \mathrm{Lip}$ we have $h \circ f \in \mathcal{H}$. In this case, the composite function $h \circ f$ that appears in the inner product defining the smooth CE is again an element of $\mathcal{H}$. Around Eq. (26), one can therefore take $\tilde{\phi} = h \circ f$, and it is then natural to expect that $smCE$ can be controlled directly via the RKHS norm.

A limitation of this argument is that RKHSs $\mathcal{H}$ enjoying such a closure property are rather restricted (for example, the RKHS associated with the Laplace kernel on $\mathcal{X} = \mathbb{R}$). Clarifying in which more general situations this type of argument holds, and identifying kernel families beyond this restricted Laplace-kernel example, is an interesting direction for future work.

### J.3 COMPARISON WITH EXISTING WORK

To the best of our knowledge, no prior work provides generalization bounds specifically for smooth CE; our results are therefore new in this respect. Fujisawa & Futami (2025) derives generalization bounds for ECE, not for smooth CE. Since $\mathrm{smCE} \leq \mathrm{ECE}$ holds in general, their bound can be transferred to smooth CE, but our results are strictly stronger in several ways: First, the bound in Fujisawa & Futami (2025) requires a smoothness assumption on the conditional probability, whose validity on real data is unclear. Our bounds hold without such additional assumptions. Second, their bound has rate $\mathcal{O}(n^{-1/3})$, whereas our bound chieves $\mathcal{O}(n^{-1/2})$, which is strictly tighter. Finally, Fujisawa & Futami (2025) does not provide conditions under which the training ECE becomes small, and therefore does not identify when $\mathrm{smCE} \leq \varepsilon$ can be guaranteed. In contrast, we show explicit conditions under which smooth CE is small for concrete algorithms.

We next clarify how our analysis relates to the post-processing gap of Błasiok et al. (2023). As shown in Section 2.2, Błasiok et al. (2023)establish that the evaluation of the post-processing gap is equivalent to the smooth calibration error. Moreover, post-processing can be interpreted as measuring how much the loss decreases when the predictor is updated; in our setting, this corresponds to updating the function along its functional gradient. This connection is the main motivation behind the analysis developed in Section 4.

While Błasiok et al. (2023) use the post-processing gap primarily to discuss, in a qualitative rather than fully theoretical way, how much the smooth CE can be reduced, Section 4 provides a rigorous characterization of this phenomenon for concrete algorithms. In particular, in situations where post-processing is implemented via functional-gradient-based updates, we prove that the smooth CE decreases for specific learning procedures. In this sense, our results can be viewed as strengthening and formalizing the insights of Błasiok et al. (2023) within an algorithm-dependent, finite-sample framework.

### J.4 GUIDANCE FOR PRACTICAL ALGORITHMS

The results in Section 4 provide concrete guidance for practical training of GBT, / Kernel Boosting / Two-layer NN models. For example, in Section 4.1 (Corollary 2), the first term in the upper bound decreases with the number of iterations $T$, reflecting the improvement of the training smooth CE. In contrast, the fourth term corresponds to the Rademacher complexity of the tree model and increases with $T$, reflecting the growth of the generalization gap (i.e., overfitting). The same qualitative behavior appears in Section 4.2 and 4.3.

Hence, our analysis formally justifies the following practical tuning rules: i) excessive iterations can worsen smooth CE due to overfitting and ii) appropriate early stopping (and regularization) can jointly optimize accuracy and smooth CE. This shows that, from the perspective of calibration as well, controlling hypothesis class complexity via regularization and early stopping is as important as it is for accuracy.

Finally, our framework and analyses also applies to post-hoc calibration methods. Suppose a model $f$ is learned using the training data $S_{\mathrm{tr}}$, and an independent recalibration dataset $S_{\mathrm{re}}$ is used to learn a post-processing map $h$, so that predictions are given by $h(f(X))$.

In this setting, the uniform convergence bounds of Section 3 can be applied to bound $\big|\mathrm{smCE}(h \circ f, \mathcal{D}) - \mathrm{smCE}(h \circ f, S_{\mathrm{re}})\big|$ i.e., the generalization gap of the post-processed model with respect to the recalibration sample. For binary classification, this allows one to analyze post-hoc calibration schemes such as histogram binning or beta calibration within the same framework. Moreover, the results in Section 4.2 provide theoretical guarantees for post-hoc methods based on kernel techniques, which have attracted attention in recent years (Wenger et al., 2020; Gruber & Bach, 2025).

The main focus of the paper is the control of smooth CE when learning $f$ on $S_{\mathrm{tr}}$, but Section 3's analysis extends directly to the generalization behavior of post-hoc calibrated models based on $S_{\mathrm{re}}$.

## K EXPERIMENT

In this section, we numerically validate our theoretical findings regarding smooth CE for GBTs and two-layer neural networks presented in Section 4.

We evaluate the behavior of several metrics on both training and test datasets, including cross-entropy loss, accuracy, functional gradient norm (denoted as "Func Grad" in the experiments), and binning ECE, MMCE, and smooth CE. These evaluations are conducted by varying the number of iterations $T$ and the training dataset size $n$. Each experiment is repeated 10 times with different random seeds, and we report the mean and standard deviation.

For the binning ECE, we use 10 equally spaced bins. For MMCE, we use the Laplacian kernel with a bandwidth set to 1. Definitions and additional details of the calibration metrics are provided in Appendix B.

### K.1 GRADIENT BOOSTING TREES

We conduct numerical experiments on the Gradient Boosting Tree with cross-entropy loss, as described in Algorithm 1. The main objective is to investigate how the training and test smooth CE behaves on both toy and real datasets, as predicted by Theorem 3 and Corollary 2.

The toy dataset consists of two classes with equal probability, i.e.,

$$P(Y = 0) = P(Y = 1) = \frac{1}{2}.$$

Given the class label $Y \in \{0, 1\}$, the conditional distribution of $X \in \mathbb{R}^2$ is:

$$X \mid Y = 0 \sim N\left(\begin{bmatrix} -1.3 \\ -1 \end{bmatrix}, \begin{bmatrix} (1.2 \cdot 1.3)^2 & 0 \\ 0 & 1.2^2 \end{bmatrix}\right),$$

$$X \mid Y = 1 \sim N\left(\begin{bmatrix} 1 \\ 1.3 \end{bmatrix}, \begin{bmatrix} 1.2^2 & 0 \\ 0 & (1.2 \cdot 1.3)^2 \end{bmatrix}\right). \tag{34}$$

First, we set $m = 3$ and present the results in Figure 1. In Figure 1(a), we fix the training dataset size ($n = 200$) and increase the number of boosting iterations $T$. The left panel shows the behavior of the metrics on the training dataset, the middle panel shows the same metrics on the test dataset, and the right panel reports the generalization gaps for both log loss and smooth CE. From the left panel, we observe that both the cross-entropy loss and smooth CE decrease as $T$ increases, consistent with Theorem 3. In contrast, the middle panel shows that neither metric necessarily decreases on the test dataset as $T$ increases. This is consistent with the well-known overfitting behavior of boosting, and demonstrates that smooth CE exhibits overfitting trends similar to those of the cross-entropy loss, as predicted by Corollary 2. The right panel confirms this overfitting phenomenon for both cross-entropy loss and smooth CE.

Next, we investigate how these metrics behave as the training sample size $n$ increases. The results are shown in Figure 1(b). We observe that the test calibration metrics decrease monotonically as $n$ increases, indicating improved generalization. These results show that GBTs, when trained with sufficient samples, can achieve both high accuracy and low smooth CE.

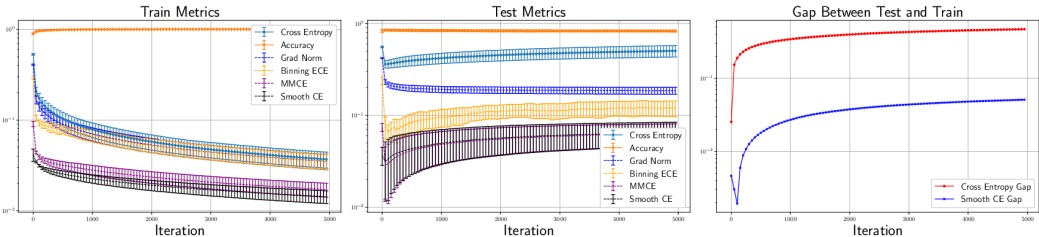

(a) Increasing the number of iterations $T$

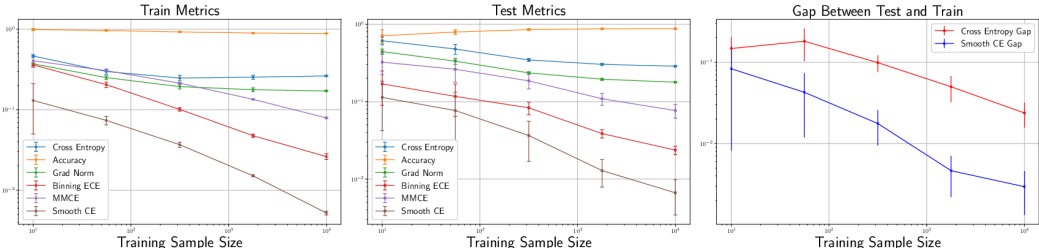

(b) Increasing the number of training samples $n$

Figure 1: GBT experiments on the toy dataset defined in Eq. (34) with $m \geq d$

Next, we set $m = 1$, which is shallow since $m \leq d$. The results are shown in Figure 2. We can see that the behavior of the calibration metrics seems similar to that of $m = 3 \geq d$.

Next, using the UCI Breast Cancer dataset, we observed how the metrics behave on the training and test datasets by increasing the number of iterations $T$ while keeping the training dataset size fixed. The results are shown in Figure 3. Since $d = 30$ in this dataset, we considered two settings: (i) $m = 30$, which is not shallow $m \geq d$, and (ii) $m = 3$, which is shallow since $m < d$.

We found that the results closely resemble those of the toy dataset in Figure 1. This implies that both shallow and not-so-shallow trees behave similarly in this setting.

Finally, we also remark that the binning ECE, smooth CE, and MMCE exhibit similar behavior across all experimental settings. This observation is consistent with the discussion in Appendix B.

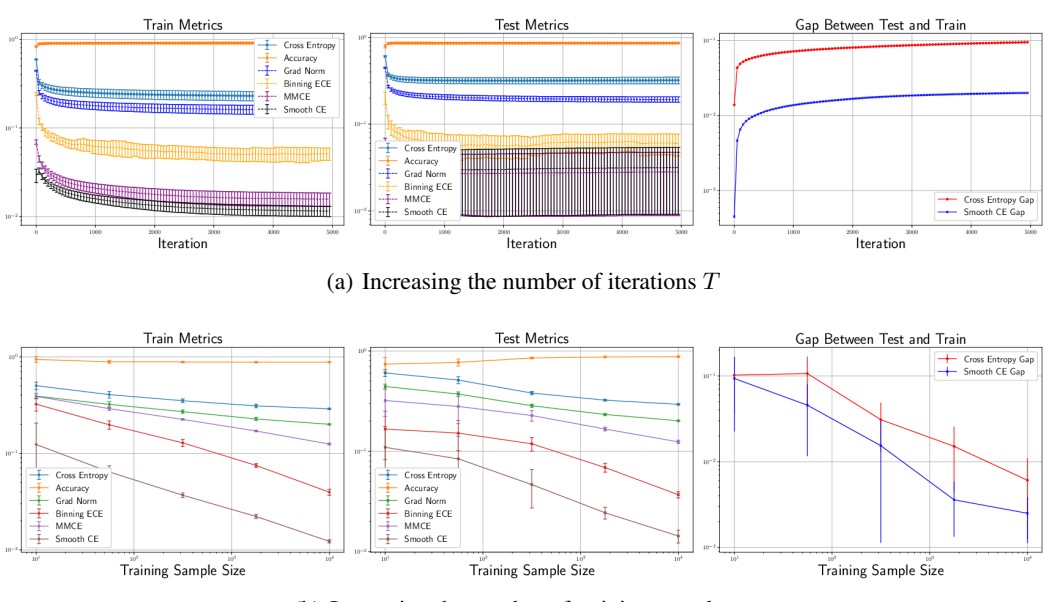

(a) Increasing the number of iterations $T$

(b) Increasing the number of training samples $n$

Figure 2: GBT experiments on the toy dataset defined in Eq. (34) with $m < d$

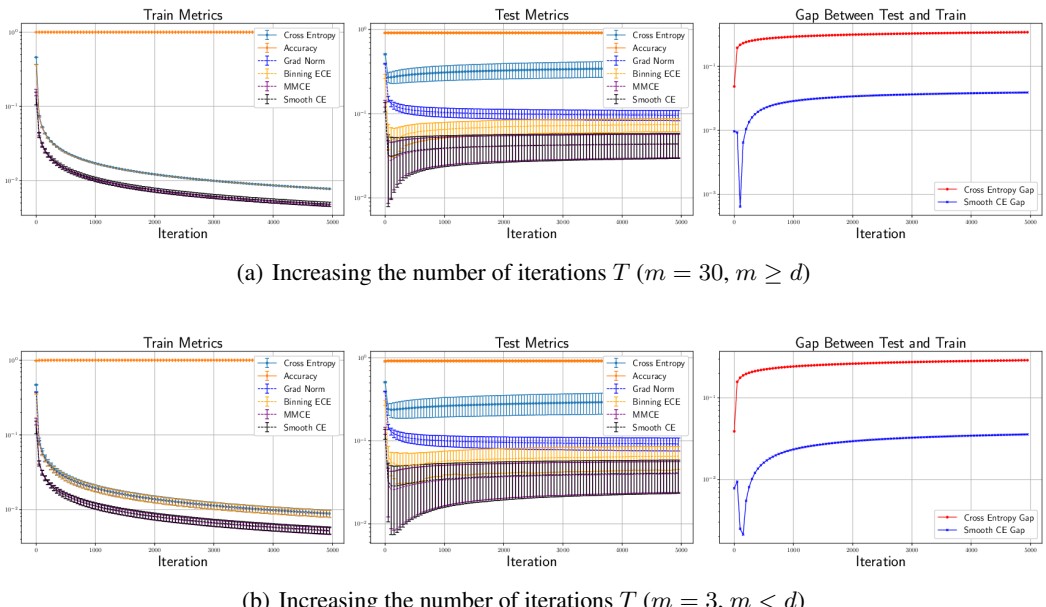

(a) Increasing the number of iterations $T$ ($m = 30, m \geq d$)

(b) Increasing the number of iterations $T$ ($m = 3, m < d$)

Figure 3: GBT experiments for UCI breast cancer dataset $d = 30$

## K.2 NUMERICAL EXPERIMENTS OVER TWO-LAYER NEURAL NETWORK

We conducted similar experiments using a two-layer neural network corresponding to Section 4.3. We prepared the toy dataset and designed the two-layer neural network to satisfy Assumption 1 in Appendix H. We used the sigmoid activation function, initialized the parameters independently from the standard Gaussian distribution, and set the number of hidden units to be even to satisfy Assumption (A3). The coefficients $a_r$ were also set according to the specification in Assumption (A3). Unless otherwise stated, we used 300 hidden units. We used full-batch gradient descent (GD) for optimization with a fixed step size of $w = 0.01$.

First, we performed experiments on the same toy dataset used for GBTs. We evaluated the training and test metrics by increasing the number of GD iterations while fixing the training dataset size ($n = 200$). Figure 4(a) reports the same set of metrics as in the GBT experiments (Figure 1). We observed that the smooth CE decreases monotonically, consistent with the upper bound behavior predicted by Eq. (6). However, from the middle and right panels of Figure 4(a), we observe that the test smooth CE does not decrease as the number of training iterations increases. We also confirmed that the generalization gap of the cross-entropy loss increases as $T$ increases. According to Eq. (31), the complexity term grows with the number of iterations, which may explain this phenomenon. On the other hand, we did not observe an apparent increase in the generalization gap of the smooth CE as $T$ increased.

Next, we varied the training dataset size, and the results are shown in Figure 4(b). From the middle and right panels, we observe that both the test smooth CE and the generalization gap decrease monotonically as $n$ increases. This is consistent with the theoretical discussion provided in Appendix H.

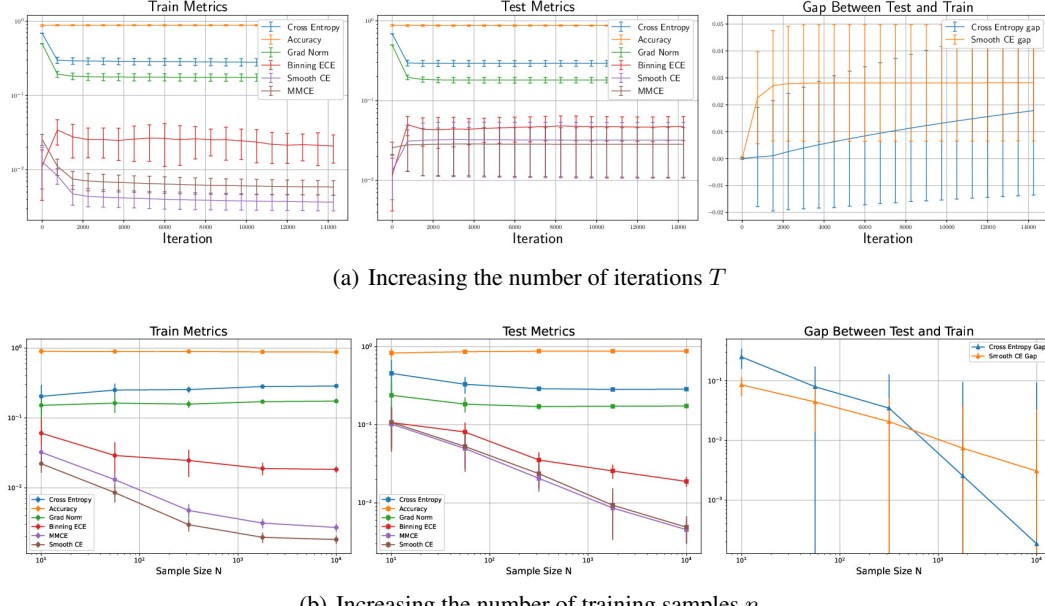

(a) Increasing the number of iterations $T$

(b) Increasing the number of training samples $n$

Figure 4: Two-layer neural network using toydata defined in Eq. (34)

The experiments in Figure 4 are based on a toy dataset where the separability condition is easily satisfied. Next, we consider a toy dataset in which the separability condition is more difficult to satisfy, constructed as follows. Assume that $n$ is even and define $n_0 = n_1 = \frac{n}{2}$. Let $\{Z_i\}_{i=1}^{n_0} \sim \mathcal{N}(0, \sigma^2 I_2)$ with $\sigma = 0.05$ be shared random base vectors. Define two classes:

$$X_i^{(0)} = Z_i + \begin{bmatrix} 0.1 \\ 0.1 \end{bmatrix} + \varepsilon_i^{(0)}, \quad \varepsilon_i^{(0)} \sim \mathcal{N}(0, \tau^2 I_2), \tag{35}$$

$$X_i^{(1)} = -Z_i + \begin{bmatrix} -0.1 \\ -0.1 \end{bmatrix} + \varepsilon_i^{(1)}, \quad \varepsilon_i^{(1)} \sim \mathcal{N}(0, \tau^2 I_2),$$

with $\tau = 0.01$. Each $X_i^{(0)}$ is labeled $Y = 0$, and each $X_i^{(1)}$ is labeled $Y = 1$. Due to the symmetric structure of the data, this setting makes the margin assumption more difficult to satisfy than in the previous toy dataset. The results are shown in Figure 5, where we increase the training dataset size and evaluate the relevant statistics on both training and test datasets. Even under this more challenging setting, we observe that the test smooth CE decreases monotonically.

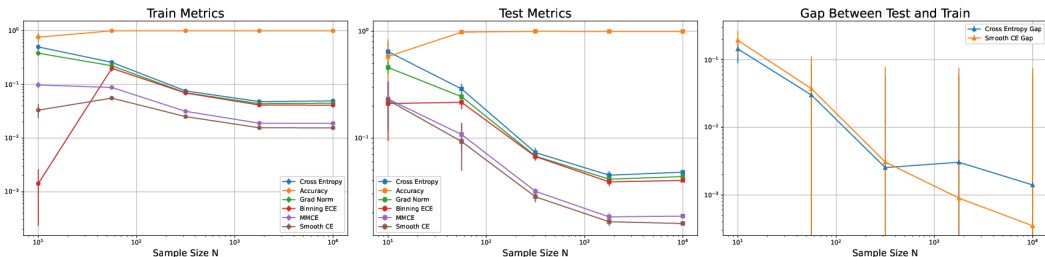

Figure 5: Two-layer neural network using toy dataset defined in Eq. (35)

Next, we used the UCI Breast Cancer dataset and evaluated various statistics on both the training and test datasets, following the same procedure as in Figure 4(a). Here, we varied the number of hidden units from a relatively small size (10) to a larger value (100). The results are shown in Figure 6.

Consistent with the results in Figure 4(a), we observed that the training statistics decrease monotonically in both small and large hidden unit settings. However, the test statistics did not show any corresponding improvement.

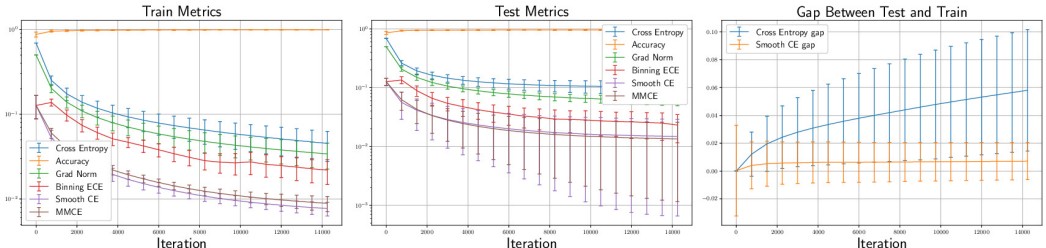

(a) Increasing the number of iterations $T$ when the hidden unit size is 10

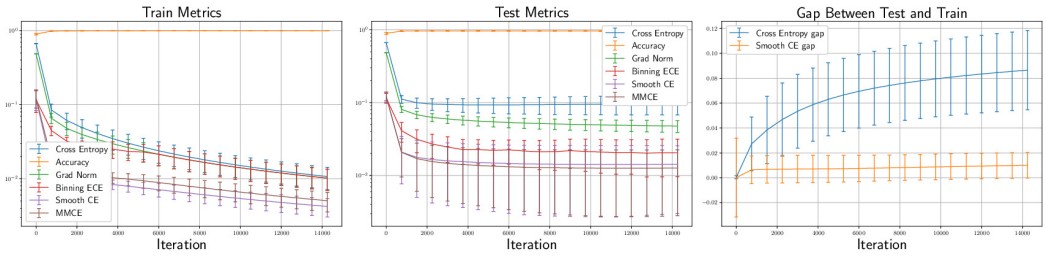

(b) Increasing the number of iterations $T$ when the hidden unit size is 100

Figure 6: UCI breast cancer dataset $d = 30$

