# OpenReview forum: "Smooth Calibration Error: Uniform Convergence and Functional Gradient Analysis"
_ICLR.cc/2026/Conference — ICLR 2026 Poster_

### Official Review · Reviewer_hJEU · 2025-11-02

**Soundness:** 3
**Presentation:** 3
**Contribution:** 3
**Rating:** 8
**Confidence:** 4

**Summary:**

The paper adds on to the accumulating evidence on the connection between loss minimisation and calibration. They adopt smooth calibration error and derive the uniform convergence result to relate the training set calibration error to test time error, and then show that training set smooth calibration error can be studied in connection to functional gradient. While the insight is simple, in hindsight, this helps answering some questions as to when one can get better predictors (accuracy wise) while also being calibrated, and the paper develops results for boosting (gradient and kernel) and two-layer neural networks. There is an extensive amount of work and details in the appendix as well (which I didn't carefully check) but also the experimental demonstration.

**Strengths:**

1.This is a strong paper, that helped answer some of my own questions on the connections between loss minimisation and calibration. My favourite part is the connection to functional gradient.

Question wise: is it fair to conclude that loss minimisation will yield calibration together with good performance if the training resembles that of the functional gradient descent or some approximation of it? In that sense, an earlier paper by (Błasiok et al., 2023; when does minimising proper losses yield calibration) argued for the emergent calibration behaviour in modern large neural networks, and how do the current functional gradient analysis add on to that?


Overall, I think this is a good paper, with not much concerns, and I think this should be accepted.

**Weaknesses:**

1. The paper misses to mention [1]. While the focus of this paper is re-calibration, I think they also study the generalisation of calibration error, and might be worth to look at.
2. While the focus of the paper is theoretical, could the study give some concrete practical guidance to design predictors to achieve both calibration and better accurate predictors.



[1] Masahiro Fujisawa, Futoshi Futami. PAC-Bayes Analysis for Recalibration in Classification. ICML 2025

**Questions:**

see above.

---

> ### Author Response · Authors · 2025-11-20
> **Response to review**
>
> We sincerely appreciate your careful reading of our paper and your positive evaluation. Below we respond to the points you raised. Also we have revised the paper to address the points you raised and have highlighted the corresponding parts in red.
>
> ## **(Q) Discussion in relation to Błasiok et al., 2023**
> As stated in Line 124 and Eq. (1) of our paper, Błasiok et al. (2023) show that the evaluation of the post-processing gap is equivalent to the smooth CE. Moreover, as we mention in Lines 220–221, post-processing measures how much the loss decreases when the predictor is updated, which in our setting corresponds to updating the function along its functional gradient. This is the main motivation behind our Section 4.
>
> In other words, while Błasiok et al. (2023) use the post-processing gap to discuss, in a qualitative rather than fully theoretical way, how much the smooth CE can be reduced, our contribution is to rigorously prove that, in situations where post-processing is implemented via functional-gradient-based updates, the smooth CE decreases for specific algorithms. In this sense, we view our work as providing a stronger theoretical guarantee for the insights of Błasiok et al. (2023).
>
> On the other hand, Section 5 of Błasiok et al. (2023) shows that the post-processing gap can also be reduced by regularization, not only via the functional-gradient-based updates. Our current analysis does not cover such regularization-based mechanisms, so extending our framework in that direction is an interesting topic for future work.
>
> This discussion is added to the Appendix J.3 in the revised version.
>
> ## **(Q) Discussion in relation to [1]**
> The paper [1] derives generalization bounds for **ECE**, not for smooth CE. Since $smCE \le ECE$ holds in general, their bound can be transferred to smooth CE, but our results are strictly stronger in several ways:
>
> (1) The bound in [1] requires a smoothness assumption on the conditional probability, whose validity on real data is unclear. Our bounds in Section 3 hold **without** such additional assumptions.
> (2) The ECE bound in [1] has rate $O(n^{-1/3})$, whereas our bound in Section 3 achieves $O(n^{-1/2})$, which is strictly tighter.
> (3) [1] does not provide conditions under which the training ECE becomes small, and therefore does not identify when $smCE \le \varepsilon$ can be guaranteed. In contrast, Section 4 (Corollary 2, Theorem 4, and Corollary 5) gives, for concrete algorithms, explicit conditions under which smooth CE is small.
>
> However, [1] proposes concrete recalibration algorithms based on generalization error analysis, which offers a perspective different from ours that is primarily focused on theoretical analysis. In future work, we would like to investigate whether our results can be connected to such recalibration algorithms.
>
> This discussion is added to the Appendix J.3 in the revised version.
>
>
> ## **(Q) Practical guidance to design predictors to achieve both calibration and better accurate predictors.**
> The results in Section 4 also provide concrete guidance for practical training of Boosting / Kernel Boosting / Two-layer NN models.
>
> For example, in Sec.4.1 (Corollary 2), the first term in the upper bound decreases with the number of iterations $T$, reflecting the improvement of the *training* smooth CE. In contrast, the fourth term corresponds to the Rademacher complexity of the tree model and *increases* with $T$, reflecting the growth of the generalization gap (i.e., overfitting). The same qualitative behavior appears in Sec.4.2 and Sec.4.3.
>
> Hence, our analysis formally justifies the following practical tuning rules:
> - excessive iterations can worsen smooth CE due to overfitting;
> - appropriate early stopping (and regularization) can jointly optimize accuracy and smooth CE.
>
> This shows that, from the perspective of calibration as well, controlling hypothesis class complexity via regularization and early stopping is as important as it is for accuracy. We have added a short practical “take-away” paragraph in the Appendix J.4 to emphasize this point.

---

### Official Review · Reviewer_J6Qt · 2025-11-06

**Soundness:** 3
**Presentation:** 3
**Contribution:** 3
**Rating:** 8
**Confidence:** 3

**Summary:**

This paper presents a two part framework for smooth calibration error or smCE (for the binary case). It first provides uniform convergence bounds that control the population level smCE by training smCE plus a generalization gap that depends on the hypothesis class. The arguments involve standard chaining number, Dudley entrop-integral type techniques (standard Kolmogorov-Tikhomirov fare) for showing a Rademacher bound. Second, it presents a functional gradient view that provides an upper bound on the training smCE by norms (of functional gradients) for proper losses. This can result in concrete guarantees for gradient boosting machines, kernel boosting, and two layer NNs (NTK regime), Under clearly articulated margin assumptions, appropriate values for T, w, and n, we get $\epsilon$-smCE and $\epsilon$ misclassification simultaneously. This teases out a clear optimization v capacity trade off.

To expand in more detail: The paper focuses on a proper loss setup, in particular on squared loss and cross entropy. smCE is defined as a supermum over 1-lipschitz post-processing of the prediction. This is connected to the post-processing gap under squared loss (Biasiok et al). For cross entropy, the dual smCE is defined on the logit with 1/4-lipschitz post-processing and the relation showing the gap is explained. Then a covering number argument is used to bound sup_f |smCE(f,Ste) − smCE(f,Str)|, avoiding explicit composition class Lip \odot f. Theorem 1 provides a tail bound with an integral over log covering numbers of F (only F, not of Lip \odot f), plus an O(1\(sqrt(n))) term reflecting the Lipschitz part. Theorem 2 provides an alternative Rademacher bound. Again, the key point is that the bound depends on F and not on Lip \odot F. The main observation is that training smCE equals a dual pairing between h(f(X)) and negative functional gradients. This means that small functional gradients imply small training smCE. After this three cases are explicated (GBT, kernel boosting, 2 layer NNs).

**Strengths:**

I like the conceptual clarity of the paper, although I can't say I am extremely familiar with the theoretical literature here. It links calibration to functional gradients, turning calibration control into optimization-style quantities.

The uniform convergence bounds only depend on  F (plus a small fixed term).

The results for the three cases are interesting and non-trivial and make the framework concrete, with explicit T, w, n, choices and accuracy-calibration trade-off.

**Weaknesses:**

The margin assumptions (classification-style) are strong for a calibration question. Many realistic miscalibration settings lack separability. (pages.4–8)

I found the kernel dependence opaque. The population bounds hide kernel spectrum dependence.

Another weakness has more to do with NTK than the framework itself.: Finite-width and feature-learning behaviors would have been interesting to discuss.

The dual smCE <> post-processing gap constants are not discussed for tightness. The $C/\sqrt(n)$ term's origin and size is not specified enough (unless I missed it).

**Questions:**

In the dual post-processing gap relation (p.2), can the (2,4) constants be improved? Would tighter constants significantly change the optimization/complexity trade-offs later?

The first $C/\sqrt(n)$  term is attributed to the Lip class. Is it possible to expose the exact constant (and L dependence) and show whether it is improvable by structure in h (e.g. by monotonicity)?

If $\gamma$ is small/zero (heavy overlap, label noise), do functional-gradient controls still yield non-trivial smCE bounds (perhaps with slower rates) via early stopping or other regularization?

Is it possible to restate Theorem 4’s population term explicitly in terms of eigen-decay (like effective dimension $N(\gamma)$. It might be useful to help pick kernels.

What breaks in the analysis if we move to softmax?

For the Dudley chaining analysis, and some of the other standard arguments, please provide in-line citations.

---

> ### Author Response · Authors · 2025-11-20
> **Response to review (1)**
>
> We sincerely appreciate your careful reading of our paper and your positive evaluation. Below we respond to the points you raised. Also we have revised the paper to address the points you raised and have highlighted the corresponding parts in red.
>
> ## **(Q) In the dual post-processing gap relation (Ep.2), can the (2,4) constants be improved?**
> This relation and constants are taken from the results of paper [1]. In Appendix D of that paper, the tightness of these constants is discussed. On the other hand, in our work, we use the relation $smCE \leq \text{(dual) } smCE$ stated at the end of Line 151, so changes in the constants in Eq. (2) do not directly affect our bounds.
>
> We have added this discussion to Appendix A as well.
>
> ## **(Q) Clarifying the universal constant and Lipschitz dependence in the first bound**
> The constant you point to, such as $C_1$ in Corollary 1, comes from Eq. (3), which is taken from Błasiok et al., 2023 ([5]). In their work, Eq. (3) is obtained by evaluating the Rademacher complexity of Lipschitz functions, and only the order is provided  omitting detailed constants. In general, bounds for Lipschitz functions are often expressed using a universal constant, but in view of your comment, we have newly derived an explicit constant in Appendix E using Dudley integral. For example, we have shown that the constant $C_1$ in for Corollary 1 in Section 3 satisfies $C_1 \leq 116$.
>
> As for the dependence on the Lipschitz constant, the smooth CE is defined over the class of 1-Lipschitz functions, so the Lipschitz constant does not explicitly appear in the universal constant. While the universal constant could be improved by tightening the evaluation of Dudley integral bound, the class of $h$ is fixed to 1-Lipschitz functions in the smooth CE setting, and such improvements would not change the essential order of the bound.
>
> We have added these derivations and the discussion of the universal constant to the Appendix E.
>
> ## **(Q) Discuss about the smCE bounds under small/zero margin ($\gamma$) and the role of regularization**
> In our analysis, we used a margin condition to evaluate upper bounds on the functional gradient norm for the Gradient Boosting and Kernel Boosting algorithms. This is used to bound $smCE(f,S_{tr})$, i.e., the functional gradient norm on the training data. When $\gamma$ is too small, our upper bound on $smCE(f,S_{tr})$ becomes large (see Theorems 3 and 4). On the other hand, the uniform convergence results in Section 3 do not depend on $\gamma$. The uniform convergence term plays the role of a regularization term and therefore the effect of regularization does not depend on $\gamma$.
>
> In this sense, our current results do not cover situations in which the margin assumption is violated. Therefore an important direction for future work is to develop a theory that does not rely on margin assumptions. In the answer to the “Weakness” comment below, we also discuss, specifically for the kernel case, a possible approach to removing the margin assumption, and we would appreciate it if you could refer to that part as well.
>
> ## **(Q) Is it possible to restate Theorem 4’s population term explicitly in terms of eigen-decay ?**
>
> In principle, it is possible to incorporate eigenvalues into the analysis. Concretely, one would use assumptions on the decay of the kernel eigenvalues when evaluating the Rademacher complexity or covering numbers used in the uniform coveregence bounds in Section 3. In our work, however, we chose to use the simpler assumption $k(x,x)\leq \Lambda$ for all $x$ to derive the upper bounds of Rademacher complexity.
>
> The reason is that an eigenvalue-based analysis requires additional assumptions defining the decay of eigenvalues of the kernel operator, which leads to a more complicated mathematical setup. Since the main focus of this work is the analysis of smooth CE via functional gradients, and the setting is already mathematically involved even without additional kernel-specific structure, we decided not to include eigenvalue-based arguments so as not to obscure the main contribution.
>
> That said, as you point out, kernel eigenvalues can provide more informative insight into learning behavior. We plan in future work to introduce concrete assumptions on the eigen-decay and analyze how this affects the overall order of the bounds.

---

> > ### Author Response · Authors · 2025-11-20
> > **Response to review (2)**
> >
> > ## **(Q) What breaks in the analysis if we move to softmax?**
> > We interpret this as a question about the difficulties in extending our results to the multi-class setting. The possibility of extending smooth CE to multi-class problems has recently been studied in [3]. However, in the multi-class case, it has been shown that when the number of classes is large, an exponential number of samples is required for the estimation. In this sense, the extension of smooth CE to the multi-class setting is itself still an ongoing research question.
> >
> > If we temporarily ignore this sample complexity issue, the multiclass smooth CE in [3] contains the term $y - f(x) \in \mathbb{R}^K$ for the $K$-class setting. We can replace this term with the functional gradient for the softmax, similarly to the binary setting, by using the result in [4]. Therefore, by applying the theorem in [4], we expect that our framework can in principle be extended to the softmax-based setting. This is an important future direction.
> >
> > ## **(Q) For the Dudley chaining analysis, and some of the other standard arguments, please provide in-line citations**
> > We apologize for the missing citations. In the revised version, we have added the appropriate references. For example, please see Lines 177, 199, 977, 1217 in the updated manuscript.
> >
> > Next, we respond to the Weaknesses you pointed out.
> >
> > ## **(W) The margin assumptions are strong for a calibration question.**
> >
> > We agree that the margin assumption is strong. In our analysis, this assumption is used specifically to control the functional gradient and to derive an upper bound on the smooth CE over the training dataset. However, in the RKHS setting there is an alternative way to control $smCE$. Suppose we work with an RKHS $\mathcal{H}$ that is closed under composition with Lipschitz functions, i.e., that contains the Lipschitz function class $\mathrm{Lip}$ in the sense that for any $f \in \mathcal{H}$ and $h \in \mathrm{Lip}$ we have $h \circ f \in \mathcal{H}$. Then the composite function $h \circ f$ that appears in the inner product defining the smooth CE is again an element of $\mathcal{H}$. Around Line 1937, we can therefore take $\tilde{\phi} = h \circ f$, and it is natural to expect that $smCE$ can be controlled directly via the RKHS norm.
> >
> > The limitation of this argument is that RKHSs $\mathcal{H}$ that enjoy this closure property are rather restricted (for example, the Laplace kernel with $\mathcal{X} = \mathbb{R}$ [5]). Clarifying in which general situations this type of argument holds, and identifying kernel families beyond this restricted Laplace-kernel example, is an important direction for future work.
> >
> > Nevertheless, we decided to retain the margin condition for the following reasons: (i) in kernel-based classification problems, this condition is typically assumed in existing work when analyzing the misclassification risk, and (ii) in the NN example in Sec.~4.3, the margin condition is essential, so including it here allows a smoother transition of the discussion to that section.
> >
> > We have added the above discussion to Appendix J.2.
> >
> > ## **(W) Limitation of the NTK framework**
> > We use the NTK framework in order to leverage the relationship between derivatives in parameter space and functional gradients, based on [2]. Proposition 8 in that paper plays a crucial role in deriving this relationship, and its proof relies on the NTK framework and the separability condition (see [2] for details).
> >
> > Therefore, at present, these assumptions are important for carrying over the discussion from parameter-space derivatives to functional gradients. Whether the Lipschitz properties inherent in smooth CE can be directly connected to feature learning behavior, without relying on NTK and separability, remains an open question and is a topic we would like to explore in future work.

---

> > > ### Comment · Reviewer_J6Qt · 2025-11-25
> > > **Ack**
> > >
> > > Thanks for the response. This is very useful. I hope the authors are able to incorporate the changes that they have promised in the various responses -- which will improve the paper.

---

### Official Review · Reviewer_oi3k · 2025-11-09

**Soundness:** 2
**Presentation:** 2
**Contribution:** 2
**Rating:** 4
**Confidence:** 4

**Summary:**

1.	The authors derive a uniform convergence bound for smooth CE.
2.	They prove that such training smooth CE can be bounded by the norm of the functional gradient of the loss evaluated on training data
3.	They apply the theoretical framework to three representative algorithms for the binary classification closely tied to functional gradients.

**Strengths:**

The paper provides a solid theoretical guarantee for the calibration performance and the classification error for three standard binary classification algorithms.

**Weaknesses:**

1. The technical novelty of deriving the results in Section 4 is not clear. In Section 4, the authors present the connection between functional gradients and the smooth CE and derive the convergence rates for three example models. These results heavily rely on the existing conclusion and analysis, but it should be clarified which contents are novel and important contributions, and why they are.

2. Lack of comparison for the uniform convergence results with the previous ones.

**Questions:**

1. Can you compare with existing works that analyze the test accuracy or smooth CE to validate your results in Section 4?

2. What guidance or improvement do the theoretical results (for example, Corollaries 3,4) bring to the algorithms in practice? Or could you design better algorithms to achieve both calibration and accuracy?

3. Neural networks are known as non-calibrated, so calibration methods, such as temperature scaling and vector scaling, are proposed. Can the theoretical framework be applied to these post-hoc techniques, and what results can be derived?

**Details Of Ethics Concerns:**

Nan

---

> ### Author Response · Authors · 2025-11-20
> **Response to review (1)**
>
> We sincerely appreciate your careful reading of our paper and the detailed comments. Below we respond to the points you raised. Also we have revised the paper to address the points you raised and have highlighted the corresponding parts in red.
>
> ## **(Q) Clarification of the technical novelty in Sec.4**
> We apologize that our technical contributions were not clearly conveyed. To clarify what they are and why they matter, we first summarize the overall contribution of Sec.4 and then explain how the technical developments support these goals.
>
> ### Overall contribution of Sec.4
> Most existing calibration works only discuss or analyze *population-level* quantities. To the best of our knowledge, there are no *algorithm-dependent* results for concrete learning methods such as Gradient Boosting trees (GBT), Kernel Boosting, or Two-layer Neural Networks (NN) that
> - quantify how fast $smCE$ decreases in finite samples, and
> - explain how optimization error and generalization error jointly control $smCE$.
>
> Against this, the main message of Sec.4 is twofold:
> 1. **First finite-sample analysis for $smCE$ for concrete algorithms.**
>    Building on the uniform convergence results of Sec.3, we derive explicit finite-sample rates for $smCE$ for three representative algorithms: GBTs, kernel methods, and NNs. This allows calibration to be discussed not only as an ideal population property, but also *quantitatively* in terms of sample size, model complexity, and number of iterations.
>
> 2. **Simultaneous guarantees for accuracy and calibration.**
>    For all three algorithms, we derive conditions under which *test accuracy* and *$smCE$* can be controlled with the **same order** in $n$ and in the algorithmic parameters. This shows how early stopping and regularization can be chosen so that *calibration is improved without sacrificing accuracy*, which is important in practical applications.
>
> ### Technical novelty and significance of Sec.4
> Then, to establish these results, the technical contributions of Sec. 4 are as follows:
> (1) a functional gradient analysis to evaluate $smCE(f, S_{tr})$,
>
> (2) a control of $|smCE(f,D) - smCE(f,S_{tr})|$ using the bounds in Sec. 3.
>
> By combining these,
> (3) we explicitly provide the model and sample complexity under which the test accuracy and $smCE$ can be guaranteed simultaneously.
>
> Below, we explain what is new about each of these three points for each algorithm.
> - **Sec.4.1: GBT**
>   - (1) Under a margin condition, we show convergence of the $L^1$ functional gradient for $smCE(f, S_{tr})$ (Theorem 3). This new technical result links the number of boosting iterations $T$ to the decrease of training $smCE$, i.e., how many iterations are needed to make the model well calibrated on the training data.
>   - (2) For the generalization error, we use the new uniform convergence bound for $smCE$ from Sec. 3 and make explicit how the improvement in $smCE(f, S_{tr})$ translates into improvement of $smCE(f, D)$ as a function of $n$ and the hypothesis class complexity (Corollary 2). This is the first analysis that characterizes the $smCE$ generalization gap as a function of the boosting iterations $T$.
>   - (3) The accuracy analysis itself relies on existing results, but the simultaneous control of accuracy and $smCE$ is new (Corollary 3). This formally shows that, for GBT, appropriate early stopping can guarantee both accuracy and calibration.
> - **Sec.4.2: Kernel methods**
>   - (1) We relate the functional gradient to the RKHS norm and analyze its convergence, which is a new technical contribution (Theorem 4). This is related to  Wei (2017) and Nitanda (2019), but we are the first to carry out a convergence analysis tailored specifically to smooth CE. It shows that, for kernel boosting, decreasing the functional gradient along iterations improves the training $smCE$.
>   - (2) Combining the uniform convergence bound from Sec. 3 with Theorem 4, we make explicit how the RKHS norm and the choice of kernel affect $smCE$. This is, as in Sec. 4.1, technically new  analysis.
>   - (3) The accuracy analysis itself is based on existing results, but the simultaneous guarantee together with $smCE$ is new (Corollary 4). This yields concrete guidance on how much regularization and how many iterations are needed for kernel methods to achieve both accuracy and calibration.
>
> - **Sec.4.3: NNs**
>   - (1) The norm of the functional gradient itself is taken directly from the result of Nitanda (2019), and since this part is not new, it is simply cited as equation (6).
>   - (2) and (3) are new in that we combine the $smCE$ bound from Sec. 3 with the optimization analysis of Nitanda (2019) to obtain a simultaneous guarantee of accuracy and $smCE$ for two-layer NNs (Corollary 5). This provides a theoretical framework that explains, in light of the empirical fact that neural networks are often miscalibrated, which training regimes can offer guarantees in terms of smooth CE.
>
> The above discussion have added in Appendix J.1.

---

> > ### Author Response · Authors · 2025-11-20
> > **Response to review (2)**
> >
> > ## **(Q) Comparison with existing bounds**
> > To the best of our knowledge, no prior work provides generalization bounds for **smooth CE**; our results are therefore new in this respect. The paper [1] mentioned by Reviewer hJEU derives generalization bounds for **ECE**, not for $smCE$. Since $smCE \le ECE$ holds in general, their bound can be transferred to $smCE$, but our results are strictly stronger in several ways:
> >
> > (1) The bound in [1] requires a smoothness assumption on the conditional probability, whose validity on real data is unclear. Our bounds in Sec.3 hold **without** such additional assumptions.
> > (2) The ECE bound in [1] has rate $O(n^{-1/3})$, whereas our bound in Sec.3 achieves $O(n^{-1/2})$, which is strictly tighter.
> > (3) [1] does not provide conditions under which the training ECE becomes small, and therefore does not identify when $smCE \le \varepsilon$ can be guaranteed. In contrast, Sec.4 (Corollary 2, Theorem 4, and Corollary 5) gives, for concrete algorithms, explicit conditions under which smooth CE is small.
> >
> > [1] Masahiro Fujisawa, Futoshi Futami. PAC-Bayes Analysis for Recalibration in Classification, 2025.
> >
> > Moreover, the uniform convergence bounds in Sec.3 (Theorems 1 and 2) are not obtained by a direct application of standard uniform convergence theory; they are technically adapted to smooth CE:
> >
> > - In Theorem 1, the bound depends only on the complexity of the hypothesis class $F$, and **not on the complexity of the Lipschitz function class** appearing in the definition of $smCE$. Standard covering-number based arguments typically introduce an extra dependence on that Lipschitz class, which is suboptimal because our primary interest is generalization over $F$. Our proof avoids this and is therefore a nontrivial technical improvement. This is discussed in Line 188.
> > - In Theorem 2, naively applying Rademacher complexity would require handling the composition of $F$ with the Lipschitz functions, again introducing unnecessary complexity. We provide a new argument that reduces the bound to the Rademacher complexity of $F$ alone. This is discussed in Line 209.
> >
> > For accuracy, we primarily rely on existing accuracy bounds (e.g., Nitanda 2018, 2019), which also yield an $O(n^{-1/2})$ rate. Our main objective is not to improve these accuracy rates, but to **align** them with our new $smCE$ bounds and thereby establish *simultaneous* control of accuracy and smooth CE.
> >
> > The above discussion is added to the Appendix J.3 in the revised version.
> >
> > ## **(Q) Guidance for practical algorithms**
> > The results in Section 4 also provide concrete guidance for practical training of Boosting / Kernel Boosting / Two-layer NN models.
> >
> > For example, in Sec.4.1 (Corollary 2), the first term in the upper bound decreases with the number of iterations $T$, reflecting the improvement of the *training* smooth CE. In contrast, the fourth term corresponds to the Rademacher complexity of the tree model and *increases* with $T$, reflecting the growth of the generalization gap (i.e., overfitting). The same qualitative behavior appears in Sec.4.2 and Sec.4.3.
> >
> > Hence, our analysis formally justifies the following practical tuning rules:
> > - excessive iterations can worsen smooth CE due to overfitting;
> > - appropriate early stopping (and regularization) can jointly optimize accuracy and smooth CE.
> >
> > This shows that, from the perspective of calibration as well, controlling hypothesis class complexity via regularization and early stopping is as important as it is for accuracy. We have added a short practical “take-away” paragraph in the Appendix J.4 to emphasize this point. We also remark that the following discussion about the post-processing is also insightful for the practitioners.
> >
> > ## **(Q) Implication for post-processing methods**
> > Our framework also applies to post-hoc calibration methods. Suppose a model $f$ is learned on the training data $S_{tr}$, and an independent recalibration dataset $S_{re}$ is used to learn a post-processing map $h$, so that predictions are $h(f(X))$.
> >
> > In this setting, the uniform convergence bounds of Section 3 control
> > $\bigl| smCE(h \circ f, D) - smCE(h \circ f, S_{re}) \bigr|,$
> > i.e., the generalization gap of the post-processed model with respect to the recalibration sample. For binary classification, this covers post-hoc calibration schemes such as histogram binning and beta calibration. In addition, the results in Sec. 4.2 yield theoretical guarantees for kernel-based post-hoc methods, which have received increasing attention [2, 3].
> >
> > The main focus of our paper is the control of $smCE$ when learning $f$ on $S_{tr}$, but the analysis in Sec.3 extends directly to the post-hoc calibrated models based on $S_{re}$. We briefly summarize this discussion in Appendix J.4.
> >
> > [2] Sebastian G. Gruber and and Francis Bach. Optimizing Estimators of Squared Calibration Errors in Classification, 2025.
> >
> > [3] J. Wenger, H. Kjellström, and R. Triebel. Non-parametric calibration for classification, 2020.

---

### Official Review · Reviewer_Y9y6 · 2025-11-10

**Soundness:** 1
**Presentation:** 3
**Contribution:** 1
**Rating:** 4
**Confidence:** 3

**Summary:**

This paper presents a theoretical understanding of learning algorithms that can produce accurate and well calibrated results. The paper mainly focuses on Smooth Calibratin Error(smCE) and provides a uniform convergence bound to that. Also, this paper analyses classifiation and calibration performance of three representative algorithms gradient boosting trees, kernel boosting and two-layer neural networks.

**Strengths:**

1. The paper attempts to provie a theoretical understanding between the relation of calinbration and classification and provided bounds of smCE.
2. The paper analyses the conditions an algorithm needs for achieving both classification and calibration gurrantees. This is extremely crucial as calibration is very important in safety critical tasks.

**Weaknesses:**

1. There are issues in the proof of theorem 1. In line 999, the author has refered to Lemma 7.6 of  Blasiok et al. (2023) and justified the existence of an optimal soultion to the convex problem in equation 8. But the assocaiation with lemma 7.6 is not obvious. In the lemma 7.6 of Blasiok et al. (2023), $\Pi(u,v,y)$ is a distribution function, hence $\Pi(u,v,y) \in [0,1]$. But, in line 997 the author has stated that $\omega_i \in [-1,1]  $.

2. The inequality in line 1343 is not clear. Means how the inequality is being derived from line 1339.

3.Same problem with the inequality in line 1357.

Overall, the theoretical section is excessively long; however, the three issues mentioned above are particularly critical, as they undermine the validity of Theorem 1. Consequently, the soundness of Corollary 1 and Theorem 2 also comes into question.

**Questions:**

1. Explain the relation with lemma 7.6.

2. Explain the inequality of both 2 and 3 in the weaknesses.

---

> ### Author Response · Authors · 2025-11-20
> **Response to review**
>
> We sincerely appreciate your careful reading of our paper and the detailed comments. Below we respond to the points you raised. Also we have revised the paper to address the points you raised and have highlighted the corresponding parts in red.
>
> ## **(Q) Response regarding Lemma 7.6**
>
> We apologize for the lack of clarity in the connection to Lemma 7.6 of Blasiok et al. (2023).
>
> First, note that the optimization problem in Eq. (8) is a linear optimization problem in $n$ variables, and its feasible region is bounded (and closed) in $\\mathbb{R}^n$. Since the objective function is linear (hence continuous) and the feasible set is bounded, the existence of an optimal solution follows directly from standard results in analysis, stating that a continuous function on a compact set attains its maximum and minimum.
>
> Thus, the existence of an optimal solution for the convex optimization problem in Eq. (8) does not rely on Lemma 7.6 of Blasiok et al. (2023). Referring to Lemma 7.6, which is formulated in terms of distribution functions, only obscured this simpler argument, and we apologize for the confusion. We have revised the manuscript:
> - we explicitly argued that the boundedness (and closedness) of the feasible region together with continuity of the objective guarantees the existence of an optimizer, and
> - we removed the unnecessary dependence on Lemma 7.6 of Blasiok et al. (2023).
>
> In addition, we clarified at Line 990 that the reformulation of the smooth CE is based on Theorem 7.14 of Blasiok et al. (2023), so that the precise role of their result in our proof is clearly identified.
>
> ## **(Q) Inequality regarding Line 1339**
>
> We also apologize for not providing a detailed explanation of the inequality at Line 1339. Here we explain the argument.
>
> By definition, for a dataset $S_n$, we write
>
> $$
> \\mathrm{smCE}(f, S_n)
> = \\max_{\\omega_1,\\dots,\\omega_n} \\frac{1}{n} \\sum_{i=1}^{n} (y_i - f(x_i)) \\omega_i
> = \\frac{1}{n} \\sum_{j=1}^{n} (y_j - f(x_j)) \\omega_j^{*}.
> $$
>
> where $(\\omega_j^{*})_{j=1}^n$ is an optimal solution given $f$ and $S_n$.
>
> Similarly, for the dataset $S_n'$ in which the $i$-th sample $(x_i,y_i)$ is replaced by $(x'_i,y'_i)$, we define
>
> $$
> \mathrm{smCE}(f, S_n')
> = \max_{\omega_1,\dots,\omega_n} \frac{1}{n} \sum_{j \neq i} (y_j-f(x_j)) \omega_j +
>   \frac{1}{n} (y_i^{\prime}-f(x_i^{\prime})) \omega_i
> = \frac{1}{n} \sum_{j \neq i} (y_j-f(x_j)) \omega_j^{\prime\star} +
>   \frac{1}{n} (y_i^{\prime}-f(x_i^{\prime})) \omega_i^{\prime\star}.
> $$
>
>
>
> where $(\\omega_j^{\\prime *})_{j=1}^n$ is an optimal solution given $f$ and $S_n'$.
>
> Since $(\omega_j^{\prime\star})_{j=1}^n$ is by definition the optimizer for $\mathrm{smCE}(f, S_n')$, we have
>
> $$
> \begin{aligned}
> \mathrm{smCE}(f, S_n')
> &= \frac{1}{n} \sum_{j \ne i} (y_j-f(x_j)) \omega_j^{\prime\star}+\frac{1}{n} (y_i^{\prime}-f(x_i^{\prime})) \omega_i^{\prime\star} \ge \frac{1}{n} \sum_{j \ne i} (y_j-f(x_j)) \omega_j^\star+\frac{1}{n} (y_i^{\prime}-f(x_i^{\prime})) \omega_i^\star.
> \end{aligned}
> $$
>
> Thus, $(\omega_j^\star)_{j=1}^n$ is just another feasible choice for the optimization, and the optimal value must be at least as large as the value attained at any feasible point.
>
> This inequality leads directly to the inequality used at Line 1339. We have added this explanation in the revised manuscript to make the argument transparent.
>
>
> ## **(Q) Inequality regarding Line 1357**
>
> As in Line 1339 above, we also use the definition of $\mathrm{smCE}$ here. For the original dataset $S_n$, we have
> $$
> \mathrm{smCE}(f, S_n)
> = \max_{\omega_1,\dots,\omega_n} \frac{1}{n} \sum_{i=1}^{n} (y_i - v_i) \omega_i
> = \frac{1}{n} \sum_{j=1}^{n} (y_j - f(x_j)) \omega_j^\star
> \geq \frac{1}{n} \sum_{j=1}^{n} (y_j - f(x_j)) \omega_j^{\prime\star}.
> $$
> In other words, this inequality comes from using a value different from the optimal solution $(\omega_j^\star)_{j=1}^n$. This yields the first inequality in the equation on Line 1357. We have added an explanation of this point in the revised manuscript as well.

---

### Author Response · Authors · 2025-12-02
**Summary of the discussion**

To summarize the discussion:

- We have responded to all questions raised by the reviewers and revised the manuscript accordingly (highlighted in red). We believe these changes address concerns.

- Reviewer Y9y6 asked (i) how the minimization problem for training the smooth CE relates to Lemma 7.6, and (ii) for more detail on the derivations of the inequalities on Lines 1339 and 1357. For (i), we clarified that the existence of a minimizer follows directly from a classical result in analysis stating that every continuous function on a compact set attains its minimum. We also provided step-by-step derivations of the relevant inequalities. Both clarifications have been added to the manuscript in red.

- Reviewer oi3k raised questions about (i) the technical novelty and significance of Section 4, (ii) comparison with existing bounds, (iii) guidance for practical algorithms, and (iv) implications for post-processing methods. To address these points and to clarify the message for future readers with similar concerns, we added a new Appendix J that explicitly discusses these aspects.

- Reviewer J6Qt requested a more detailed discussion of the constants in our bounds, as well as derivations that track explicit constants rather than unspecified universal ones. In response, we added new results in Appendix E. The reviewer indicated that these clarifications were satisfactory, so we believe their concerns have been fully resolved.

- Reviewer hJEU asked for additional comparisons with related work and further guidance for practical algorithm design. We addressed these points in the newly added Appendix J.

- Although not all reviewers replied during the discussion phase, we believe that our rebuttal and the corresponding revisions to the manuscript fully resolve the raised concerns.

---

### Meta-Review · Area_Chair_Y8mq · 2026-01-07

**Summary:**

The submission provides a smooth calibration error analysis of several learning methods: gradient boosting trees, kernel boosting, and 2 layer neural networks.  This gives some generalization bounds for proper losses with respect to calibration error, which is a nice theoretical result.

**Reviewer Concerns:**

Reviewers were initially split on the submission, with two giving clear acceptance recommendations and two indicating that there were issues putting it marginally below acceptance.  The technical issues were clearly rebutted, with improvements/clarifications in mathematical arguments as a result.  Concerns about e.g. neural networks are not entirely resolved, as the results are limited to 2 layer networks, which do not fully encompass the complexity of deep networks.

**Reviewer Scores:**

The technical issues are essentially successfully rebutted, so I estimate that at least one reviewer who gave a score of 4 would have increased their evaluation, yielding a majority vote for acceptance.  I think this is a nice contribution, of course with its limitations.  It would be a welcome paper at ICLR.

---

### Decision · Program_Chairs · 2026-01-26

Accept (Poster)